# ON THE UNIVERSALITY OF THE DOUBLE DESCENT PEAK IN RIDGELESS REGRESSION

**David Holzmüller**
University of Stuttgart
Faculty of Mathematics and Physics
Institute for Stochastics and Applications
`david.holzmueller@mathematik.uni-stuttgart.de`

## ABSTRACT

We prove a non-asymptotic distribution-independent lower bound for the expected mean squared generalization error caused by label noise in ridgeless linear regression. Our lower bound generalizes a similar known result to the overparameterized (interpolating) regime. In contrast to most previous works, our analysis applies to a broad class of input distributions with almost surely full-rank feature matrices, which allows us to cover various types of deterministic or random feature maps. Our lower bound is asymptotically sharp and implies that in the presence of label noise, ridgeless linear regression does not perform well around the interpolation threshold for any of these feature maps. We analyze the imposed assumptions in detail and provide a theory for analytic (random) feature maps. Using this theory, we can show that our assumptions are satisfied for input distributions with a (Lebesgue) density and feature maps given by random deep neural networks with analytic activation functions like sigmoid, tanh, softplus or GELU. As further examples, we show that feature maps from random Fourier features and polynomial kernels also satisfy our assumptions. We complement our theory with further experimental and analytic results.

## 1 INTRODUCTION

Seeking for a better understanding of the successes of deep learning, Zhang et al. (2016) pointed out that deep neural networks can achieve very good performance despite being able to fit random noise, which sparked the interest of many researchers in studying the performance of interpolating learning methods. Belkin et al. (2018) made a similar observation for kernel methods and showed that classical generalization bounds are unable to explain this phenomenon. Belkin et al. (2019a) observed a "double descent" phenomenon in various learning models, where the test error first decreases with increasing model complexity, then increases towards the "interpolation threshold" where the model is first able to fit the training data perfectly, and then decreases again in the "overparameterized" regime where the model capacity is larger than the training set. This phenomenon has also been discovered in several other works (Bös & Opper, 1997; Advani & Saxe, 2017; Neal et al., 2018; Spigler et al., 2019). Nakkiran et al. (2019) performed a large empirical study on deep neural networks and found that double descent can not only occur as a function of model capacity, but also as a function of the number of training epochs or as a function of the number of training samples.

Theoretical investigations of the double descent phenomenon have mostly focused on specific unregularized ("ridgeless") or weakly regularized linear regression models. These linear models can be described via i.i.d. samples $(\boldsymbol{x}_1, y_1), \ldots, (\boldsymbol{x}_n, y_n) \in \mathbb{R}^d$, where the covariates $\boldsymbol{x}_i$ are mapped to feature representations $\boldsymbol{z}_i = \phi(\boldsymbol{x}_i) \in \mathbb{R}^p$ via a (potentially random) feature map $\phi$, and (ridgeless) linear regression is then performed on the transformed samples $(\boldsymbol{z}_i, y_i)$. While linear regression with random features can be understood as a simplified model of fully trained neural networks, it is also interesting in its own right: For example, random Fourier features (Rahimi & Recht, 2008) and random neural network features (see e.g. Cao et al., 2018; Scardapane & Wang, 2017) have gained a notable amount of attention.

Unfortunately, existing theoretical investigations of double descent are usually limited in one or more of the following ways:

(1) They assume that the $z_i$ (or a linear transformation thereof) have (centered) i.i.d. components. This assumption is made by Hastie et al. (2019), while Advani & Saxe (2017) and Belkin et al. (2019b) even assume that the $z_i$ follow a Gaussian distribution. While the assumption of i.i.d. components facilitates the application of some random matrix theory results, it excludes most feature maps: For feature maps $\phi$ with $d < p$, the $z_i$ will usually be concentrated on a $d$-dimensional submanifold of $\mathbb{R}^p$, and will therefore usually *not* have i.i.d. components.

(2) They assume a (shallow) random feature model with fixed distribution of the $x_i$, e.g. an isotropic Gaussian distribution or a uniform distribution on a sphere. Examples for this are the single-layer random neural network feature models by Hastie et al. (2019) in the un-regularized case and by Mei & Montanari (2019); d'Ascoli et al. (2020a) in the regularized case. A simple Fourier model with $d = 1$ has been studied by Belkin et al. (2019b). While these analyses provide insights for some practically relevant random feature models, the assumptions on the input distribution prevent them from applying to real-world data.

(3) Their analysis only applies in a high-dimensional limit where $n, p \to \infty$ and $n/p \to \gamma$, where $\gamma \in (0, \infty)$ is a constant. This applies to all works mentioned in (1) and (2) except the model by Belkin et al. (2019b) where the $z_i$ follow a standard Gaussian distribution.

In this paper, we provide an analysis under significantly weaker assumptions. We introduce the basic setting of our paper in Section 2 and Section 3. Our main contributions are:

- In Section 4, we show a non-asymptotic distribution-independent lower bound for the expected excess risk of ridgeless linear regression with (random) features. While the underparameterized bound is adapted from a minimax lower bound in Mourtada (2019), the overparameterized bound is new and perfectly complements the underparameterized version. The obtained general lower bound relies on significantly weaker assumptions than most previous works and shows that there is only limited potential to reduce the sensitivity of unregularized linear models to label noise via engineering better feature maps.

- In Section 5, we show that our lower bound applies to a large class of input distributions and feature maps including random deep neural networks, random Fourier features and polynomial kernels. This analysis is also relevant for related work where similar assumptions are not investigated (e.g. Mourtada, 2019; Muthukumar et al., 2020). For random deep neural networks, our result requires weaker assumptions than a related result by Nguyen & Hein (2017).

- In Section 6 and Appendix C, we compare our lower bound to new theoretical and experimental results for specific examples, including random neural network feature maps as well as finite-width Neural Tangent Kernels (Jacot et al., 2018). We also show that our lower bound is asymptotically sharp in the limit $n, p \to \infty$.

Similar to this paper, Muthukumar et al. (2020) study the "fundamental price of interpolation" in the overparameterized regime, providing a probabilistic lower bound for the generalization error under the assumption of subgaussian features or (suitably) bounded features. We explain the difference to our lower bound in detail in Appendix L, showing that our overparameterized lower bound for the expected generalization error requires significantly weaker assumptions, that it is uniform across feature maps and that it yields a more extreme interpolation peak.

Our lower bound also applies to a large class of kernels if they can be represented using a feature map with finite-dimensional feature space, i.e. $p < \infty$. For ridgeless regression with certain classes of kernels, lower or upper bounds have been derived (Liang & Rakhlin, 2020; Rakhlin & Zhai, 2019; Liang et al., 2019). However, as explained in more detail in Appendix K, these analyses impose restrictions on the kernels that allow them to ignore "double descent" type phenomena in the feature space dimension $p$.

Beyond Double Descent, a series of papers have studied "Multiple Descent" phenomena theoretically and empirically, both with respect to the number of parameters $p$ and the input dimension $d$. Adlam & Pennington (2020) and d'Ascoli et al. (2020b) theoretically investigate Triple Descent phenomena. Nakkiran et al. (2020) argue that Double Descent can be mitigated by optimal regularization. They also empirically observe a form of Triple Descent in an unregularized model. Liang

et al. (2019) prove an upper bound exhibiting infinitely many peaks and empirically observe Multiple Descent. Chen et al. (2020) show that in ridgeless linear regression, the feature distributions can be designed to control the locations of ascents and descents in the double descent curve for a "dimension-normalized" noise-induced generalization error. Our lower bound provides a fundamental limit to this "designability" of the generalization curve for methods that can interpolate with probability one in the overparameterized regime.

Proofs for our statements can be found in the appendix. We provide code to reproduce all of our experimental results at

$$\texttt{https://github.com/dholzmueller/universal\_double\_descent}$$

and we will provide the computed data at $\texttt{https://doi.org/10.18419/darus-1771}$.

## 2 BASIC SETTING AND NOTATION

Following Györfi et al. (2002), we consider the scenario where the samples $(\boldsymbol{x}_i, y_i)$ of a data set $D = ((\boldsymbol{x}_1, y_1), \ldots, (\boldsymbol{x}_n, y_n)) \in (\mathbb{R}^d \times \mathbb{R})^n$ are sampled independently from a probability distribution $P$ on $\mathbb{R}^d \times \mathbb{R}$, i.e. $D \sim P^n$.[1] We define

$$\boldsymbol{X} := \begin{pmatrix} \boldsymbol{x}_1^\top \\ \vdots \\ \boldsymbol{x}_n^\top \end{pmatrix} \in \mathbb{R}^{n \times d}, \qquad \boldsymbol{y} := \begin{pmatrix} y_1 \\ \vdots \\ y_n \end{pmatrix} \in \mathbb{R}^n .$$

We also consider random variables $(\boldsymbol{x}, y) \sim P$ that are independent of $D$ and denote the distribution of $\boldsymbol{x}$ by $P_X$. The (least squares) *population risk* of a function $f : \mathbb{R}^d \to \mathbb{R}$ is defined as

$$R_P(f) := \mathbb{E}_{\boldsymbol{x}, y}(y - f(\boldsymbol{x}))^2 .$$

We assume $\mathbb{E}y^2 < \infty$. Then, $R_P$ is minimized by the target function $f_P^*$ given by

$$f_P^*(\boldsymbol{x}) = \mathbb{E}(y|\boldsymbol{x}) ,$$

we have $R_P(f_P^*) < \infty$, and the *excess risk* (a.k.a. generalization error) of a function $f$ is

$$R_P(f) - R_P(f_P^*) = \mathbb{E}_{\boldsymbol{x}}(f(\boldsymbol{x}) - f_P^*(\boldsymbol{x}))^2 .$$

**Notation** For two symmetric matrices, we write $\boldsymbol{A} \succeq \boldsymbol{B}$ if $\boldsymbol{A} - \boldsymbol{B}$ is positive semidefinite and $\boldsymbol{A} \succ \boldsymbol{B}$ if $\boldsymbol{A} - \boldsymbol{B}$ is positive definite. For a symmetric matrix $\boldsymbol{S} \in \mathbb{R}^{n \times n}$, we let $\lambda_1(\boldsymbol{S}) \geq \ldots \geq \lambda_n(\boldsymbol{S})$ be its eigenvalues in descending order. We denote the trace of $\boldsymbol{A}$ by $\mathrm{tr}(\boldsymbol{A})$ and the Moore-Penrose pseudoinverse of $\boldsymbol{A}$ by $\boldsymbol{A}^+$. For $\phi : \mathbb{R}^d \to \mathbb{R}^p$ and $\boldsymbol{X} \in \mathbb{R}^{n \times d}$, we let $\phi(\boldsymbol{X}) \in \mathbb{R}^{n \times p}$ be the matrix with $\phi$ applied to each of the *rows* of $\boldsymbol{X}$ individually. For a set $\mathcal{A}$, we denote its indicator function by $\mathbb{1}_{\mathcal{A}}$. For a random variable $\boldsymbol{x}$, we say that $\boldsymbol{x}$ has a Lebesgue density if $P_X$ can be represented by a probability density function (w.r.t. the Lebesgue measure). We say that $\boldsymbol{x}$ is nonatomic if for all possible values $\widetilde{\boldsymbol{x}}$, $P(\boldsymbol{x} = \widetilde{\boldsymbol{x}}) = 0$. We denote the uniform distribution on a set $\mathcal{A}$, e.g. the unit sphere $\mathbb{S}^{p-1} \subseteq \mathbb{R}^p$, by $\mathcal{U}(\mathcal{A})$. We denote the normal (Gaussian) distribution with mean $\boldsymbol{\mu}$ and covariance $\boldsymbol{\Sigma}$ by $\mathcal{N}(\boldsymbol{\mu}, \boldsymbol{\Sigma})$. For $n \in \mathbb{N}$, we define $[n] := \{1, \ldots, n\}$.

We review relevant matrix facts, e.g. concerning the Moore-Penrose pseudoinverse, in Appendix B.

## 3 LINEAR REGRESSION WITH (RANDOM) FEATURES

The most general setting that we will consider in this paper is ridgeless linear regression in random features: Given a random variable $\boldsymbol{\theta}$ that is independent from the data set $D$ and an associated random feature map $\phi_{\boldsymbol{\theta}} : \mathbb{R}^d \to \mathbb{R}^p$, we define the estimator

$$f_{\boldsymbol{X}, \boldsymbol{y}, \boldsymbol{\theta}}(\boldsymbol{x}) := \phi_{\boldsymbol{\theta}}(\boldsymbol{x})^\top \phi_{\boldsymbol{\theta}}(\boldsymbol{X})^+ \boldsymbol{y} ,$$

---

[1]Although many of our theorems apply to general domains $\boldsymbol{x}_i \in \mathcal{X}$ and not just $\mathcal{X} = \mathbb{R}^d$, we set $\mathcal{X} = \mathbb{R}^d$ for notational simplicity. We require $\mathcal{X} = \mathbb{R}^d$ whenever we assume that the distribution of the $\boldsymbol{x}_i$ has a Lebesgue density or work with analytic feature maps.

which simply performs unregularized linear regression with random features. As a special case, the feature map $\phi_{\boldsymbol{\theta}}$ may be deterministic, in which case we drop the index $\boldsymbol{\theta}$. An even more specialized case is ordinary linear regression, where $d = p$ and $\phi_{\boldsymbol{\theta}} = \mathrm{id}$, yielding $f_{\boldsymbol{X},\boldsymbol{y}}(\boldsymbol{x}) = \boldsymbol{x}^\top \boldsymbol{X}^+ \boldsymbol{y}$.

As described in Hastie et al. (2019), the ridgeless linear regression parameter $\widehat{\boldsymbol{\beta}} := \phi_{\boldsymbol{\theta}}(\boldsymbol{X})^+ \boldsymbol{y}$

- has minimal Euclidean norm among all parameters $\boldsymbol{\beta}$ minimizing $\|\phi_{\boldsymbol{\theta}}(\boldsymbol{X})\boldsymbol{\beta} - \boldsymbol{y}\|_2^2$,
- is the limit of gradient descent with sufficiently small step size on $L(\boldsymbol{\beta}) := \|\phi_{\boldsymbol{\theta}}(\boldsymbol{X})\boldsymbol{\beta} - \boldsymbol{y}\|_2^2$ with initialization $\boldsymbol{\beta}^{(0)} := \boldsymbol{0}$, and
- is the limit of ridge regression with regularization $\lambda > 0$ for $\lambda \searrow 0$:

$$\widehat{\boldsymbol{\beta}} = \lim_{\lambda \searrow 0} \phi_{\boldsymbol{\theta}}(\boldsymbol{X})^\top (\phi_{\boldsymbol{\theta}}(\boldsymbol{X})\phi_{\boldsymbol{\theta}}(\boldsymbol{X})^\top + \lambda \boldsymbol{I}_n)^{-1} \boldsymbol{y} \,. \tag{1}$$

For a fixed feature map $\phi$, the kernel trick provides a correspondence between ridgeless linear regression with $\phi$ and ridgeless kernel regression with the kernel $k(\boldsymbol{x}, \tilde{\boldsymbol{x}}) := \phi(\boldsymbol{x})^\top \phi(\tilde{\boldsymbol{x}})$ via

$$f_{\boldsymbol{X},\boldsymbol{y}}(\boldsymbol{x}) = \phi(\boldsymbol{x})^\top \phi(\boldsymbol{X})^+ \boldsymbol{y} = \phi(\boldsymbol{x})^\top \phi(\boldsymbol{X})^\top (\phi(\boldsymbol{X})\phi(\boldsymbol{X})^\top)^+ \boldsymbol{y} = k(\boldsymbol{x}, \boldsymbol{X})k(\boldsymbol{X}, \boldsymbol{X})^+ \boldsymbol{y} \,, \tag{2}$$

where

$$k(\boldsymbol{x}, \boldsymbol{X}) := \begin{pmatrix} k(\boldsymbol{x}, \boldsymbol{x}_1) \\ \vdots \\ k(\boldsymbol{x}, \boldsymbol{x}_n) \end{pmatrix}, \qquad k(\boldsymbol{X}, \boldsymbol{X}) := \begin{pmatrix} k(\boldsymbol{x}_1, \boldsymbol{x}_1) & \dots & k(\boldsymbol{x}_1, \boldsymbol{x}_n) \\ \vdots & \ddots & \vdots \\ k(\boldsymbol{x}_n, \boldsymbol{x}_1) & \dots & k(\boldsymbol{x}_n, \boldsymbol{x}_n) \end{pmatrix} \,.$$

## 4   A LOWER BOUND

In this section, we prove our main theorem, which provides a non-asymptotic distribution-independent lower bound on the expected excess risk.

The expected excess risk $\mathbb{E}_{\boldsymbol{X},\boldsymbol{y},\boldsymbol{\theta}} R_P(f_{\boldsymbol{X},\boldsymbol{y},\boldsymbol{\theta}}) - R_P(f_P^*)$ can be decomposed into several different contributions (see e.g. d'Ascoli et al., 2020a). In the following, we will focus on the contribution of label noise to the expected excess risk for the estimators $f_{\boldsymbol{X},\boldsymbol{y},\boldsymbol{\theta}}$ considered in Section 3. Using a bias-variance decomposition with respect to $\boldsymbol{y}$, it is not hard to show that the label-noise-induced error provides a lower bound for the expected excess risk:

$$\begin{aligned}
\mathcal{E}_{\mathrm{Noise}} &:= \mathbb{E}_{\boldsymbol{X},\boldsymbol{y},\boldsymbol{\theta},\boldsymbol{x}} \left( f_{\boldsymbol{X},\boldsymbol{y},\boldsymbol{\theta}}(\boldsymbol{x}) - \mathbb{E}_{\boldsymbol{y}|\boldsymbol{X}} f_{\boldsymbol{X},\boldsymbol{y},\boldsymbol{\theta}}(\boldsymbol{x}) \right)^2 \\
&\leq \mathbb{E}_{\boldsymbol{X},\boldsymbol{\theta},\boldsymbol{x}} \left[ \mathbb{E}_{\boldsymbol{y}|\boldsymbol{X}} \left( f_{\boldsymbol{X},\boldsymbol{y},\boldsymbol{\theta}}(\boldsymbol{x}) - \mathbb{E}_{\boldsymbol{y}|\boldsymbol{X}} f_{\boldsymbol{X},\boldsymbol{y},\boldsymbol{\theta}}(\boldsymbol{x}) \right)^2 + \left( \mathbb{E}_{\boldsymbol{y}|\boldsymbol{X}} f_{\boldsymbol{X},\boldsymbol{y},\boldsymbol{\theta}}(\boldsymbol{x}) - f_P^*(\boldsymbol{x}) \right)^2 \right] \\
&= \mathbb{E}_{\boldsymbol{X},\boldsymbol{y},\boldsymbol{\theta},\boldsymbol{x}} \left( f_{\boldsymbol{X},\boldsymbol{y},\boldsymbol{\theta}}(\boldsymbol{x}) - f_P^*(\boldsymbol{x}) \right)^2 \\
&= \mathbb{E}_{\boldsymbol{X},\boldsymbol{y},\boldsymbol{\theta}}(R_P(f_{\boldsymbol{X},\boldsymbol{y},\boldsymbol{\theta}}) - R_P(f_P^*)) \,.
\end{aligned}$$

For linear models as considered here, it is not hard to see that $\mathcal{E}_{\mathrm{Noise}}$ does not depend on $f_P^*$ and is equal to the expected excess risk in the special case $f_P^* \equiv 0$.

In the following, we first consider the setting where the feature map $\phi$ is deterministic. We will consider linear regression on $\boldsymbol{z} := \phi(\boldsymbol{x})$ and $\boldsymbol{Z} := \phi(\boldsymbol{X})$ and formulate our assumptions directly w.r.t. the distribution $P_Z$ of $\boldsymbol{z}$, hiding the dependence on the feature map $\phi$. While the distribution $P_X$ of $\boldsymbol{x}$ is usually fixed and determined by the problem, the distribution $P_Z$ can be actively influenced by choosing a suitable feature map $\phi$. We will analyze in Section 5 how the assumptions on $P_Z$ can be translated back to assumptions on $P_X$ and assumptions on $\phi$.

**Remark 1.** For typical feature maps, we have $p > d$, $P_Z$ is concentrated on a $d$-dimensional submanifold of $\mathbb{R}^p$ and the components of $\boldsymbol{z}$ are not independent. A simple example (cf. Proposition 8) is a polynomial feature map $\phi : \mathbb{R}^1 \to \mathbb{R}^p, x \mapsto (1, x, x^2, \dots, x^{p-1})$. The imposed assumptions on $P_Z$ should hence allow for such distributions on submanifolds and *not* require independent components. ◀

**Definition 2.** Assuming that $\mathbb{E}\|\boldsymbol{z}\|_2^2 < \infty$, i.e. (MOM) in Theorem 3 holds, we can define the (positive semidefinite) second moment matrix

$$\boldsymbol{\Sigma} := \mathbb{E}_{\boldsymbol{z} \sim P_Z} \left[ \boldsymbol{z}\boldsymbol{z}^\top \right] \in \mathbb{R}^{p \times p} \,.$$

If $\mathbb{E}\boldsymbol{z} = 0$, $\boldsymbol{\Sigma}$ is also the covariance matrix of $\boldsymbol{z}$. If $\boldsymbol{\Sigma}$ is invertible, i.e. (COV) in Theorem 3 holds, the rows $\boldsymbol{w}_i := \boldsymbol{\Sigma}^{-1/2}\boldsymbol{x}_i$ of the "whitened" data matrix $\boldsymbol{W} := \boldsymbol{Z}\boldsymbol{\Sigma}^{-1/2}$ satisfy $\mathbb{E}\boldsymbol{w}_i\boldsymbol{w}_i^\top = \boldsymbol{I}_p$. ◀

With these preparations, we can now state our main theorem. Its assumptions and the obtained lower bound will be discussed in Section 5 and Section 6, respectively.

**Theorem 3** (Main result). *Let $n, p \geq 1$. Assume that $P$ and $\phi$ satisfy:*

*(INT)* $\mathbb{E}y^2 < \infty$ and hence $R_P(f_P^*) < \infty$,
*(NOI)* $\mathrm{Var}(y|\mathbf{z}) \geq \sigma^2$ almost surely over $\mathbf{z}$,
*(MOM)* $\mathbb{E}\|\mathbf{z}\|_2^2 < \infty$, i.e. $\mathbf{\Sigma}$ exists and is finite,
*(COV)* $\mathbf{\Sigma}$ is invertible,
*(FRK)* $\mathbf{Z} \in \mathbb{R}^{n \times p}$ almost surely has full rank, i.e. $\mathrm{rank}\,\mathbf{Z} = \min\{n, p\}$.

*Then, for the ridgeless linear regression estimator $f_{\mathbf{Z},\mathbf{y}}(\mathbf{z}) = \mathbf{z}^\top \mathbf{Z}^+ \mathbf{y}$, the following holds:*

$$\text{If } p \geq n, \qquad \mathcal{E}_{\mathrm{Noise}} \overset{(I)}{\geq} \sigma^2 \mathbb{E}_{\mathbf{Z}} \, \mathrm{tr}((\mathbf{Z}^+)^\top \mathbf{\Sigma} \mathbf{Z}^+) \overset{(II)}{\geq} \sigma^2 \mathbb{E}_{\mathbf{Z}} \, \mathrm{tr}((\mathbf{W}\mathbf{W}^\top)^{-1}) \overset{(IV)}{\geq} \sigma^2 \frac{n}{p + 1 - n} \, .$$

$$\text{If } p \leq n, \qquad \mathcal{E}_{\mathrm{Noise}} \overset{(I)}{\geq} \sigma^2 \mathbb{E}_{\mathbf{Z}} \, \mathrm{tr}((\mathbf{Z}^+)^\top \mathbf{\Sigma} \mathbf{Z}^+) \overset{(III)}{=} \sigma^2 \mathbb{E}_{\mathbf{Z}} \, \mathrm{tr}((\mathbf{W}^\top \mathbf{W})^{-1}) \overset{(V)}{\geq} \sigma^2 \frac{p}{n + 1 - p} \, .$$

*Here, the matrix inverses exist almost surely in the considered cases. Moreover, we have:*

- *If (NOI) holds with equality, then* (I) *holds with equality.*
- *If $n = p$ or $\mathbf{\Sigma} = \lambda \mathbf{I}_p$ for some $\lambda > 0$, then* (II) *holds with equality.*

For a discussion on how $\mathbf{\Sigma}$ influences $\mathcal{E}_{\mathrm{Noise}}$, we refer to Remark G.1. We can extend Theorem 3 to random features if it holds for almost all of the random feature maps:

**Corollary 4** (Random features). *Let $\boldsymbol{\theta} \sim P_\Theta$ be a random variable such that $\phi_{\boldsymbol{\theta}} : \mathbb{R}^d \to \mathbb{R}^p$ is a random feature map. Consider the random features regression estimator $f_{\mathbf{X},\mathbf{y},\boldsymbol{\theta}}(\mathbf{x}) = \mathbf{z}_{\boldsymbol{\theta}}^\top \mathbf{Z}_{\boldsymbol{\theta}}^+ \mathbf{y}$ with $\mathbf{z}_{\boldsymbol{\theta}} := \phi_{\boldsymbol{\theta}}(\mathbf{x})$ and $\mathbf{Z}_{\boldsymbol{\theta}} := \phi_{\boldsymbol{\theta}}(\mathbf{X})$. If for $P_\Theta$-almost all $\widetilde{\boldsymbol{\theta}}$, the assumptions of Theorem 3 are satisfied for $\mathbf{z} = \mathbf{z}_{\widetilde{\boldsymbol{\theta}}}$ and $\mathbf{Z} = \mathbf{Z}_{\widetilde{\boldsymbol{\theta}}}$ (with the corresponding matrix $\mathbf{\Sigma} = \mathbf{\Sigma}_{\widetilde{\boldsymbol{\theta}}}$), then*

$$\mathcal{E}_{\mathrm{Noise}} \geq \begin{cases} \sigma^2 \frac{n}{p+1-n} & \text{if } p \geq n, \\ \sigma^2 \frac{p}{n+1-p} & \text{if } p \leq n. \end{cases}$$

The main novelty in Theorem 3 is the explicit uniform lower bound (IV) for $p \geq n$: The lower bound (V) for $p \leq n$ follows by adapting Corollary 1 in Mourtada (2019). Statements similar to (I), (II) and (III) have also been proven, see e.g. Hastie et al. (2019) and Theorem 1 in Muthukumar et al. (2020). However, as discussed in Section 1, Hastie et al. (2019) make sificiantly stronger assumptions for computing the expectation. In Appendix L, we explain in more detail that the probabilistic overparameterized lower bound of Muthukumar et al. (2020) is not distribution-independent and only applies to a smaller class of distributions than our lower bound. For a discussion on how Theorem 3 applies to kernels with finite-dimensional feature space, we refer to Appendix K.

## 5 WHEN ARE THE ASSUMPTIONS SATISFIED?

In this section, we want to discuss the assumptions of Theorem 3 and provide different results helping to verify these assumptions for various input distributions and feature maps. The theory will be particularly nice for analytic feature maps, which we define now:

**Definition 5** (Analytic function). *A function $f : \mathbb{R}^d \to \mathbb{R}$ is called (real) analytic if for all $\mathbf{z} \in \mathbb{R}^d$, the Taylor series of $f$ around $\mathbf{z}$ converges to $f$ in a neighborhood of $\mathbf{z}$. A function $f : \mathbb{R}^d \to \mathbb{R}^p, \mathbf{z} \mapsto (f_1(\mathbf{z}), \ldots, f_p(\mathbf{z}))$ is called (real) analytic if $f_1, \ldots, f_p$ are analytic.* ◀

Sums, products and compositions of analytic functions are analytic, cf. e.g. Section 2.2 in Krantz & Parks (2002). We will discuss examples of analytic functions later in this section.

**Proposition 6** (Characterization of (COV) and (FRK)). *Consider the setting of Theorem 3 and let $\mathrm{FRK}(n)$ be the statement that (FRK) holds for $n$. Then,*

*(i) Let $n \geq 1$. Then, $\mathrm{FRK}(n)$ iff $P(\mathbf{z} \in U) = 0$ for all linear subspaces $U \subseteq \mathbb{R}^p$ of dimension $\min\{n, p\} - 1$.*

*(ii) Let (MOM) hold. Then, (COV) holds iff $P(\mathbf{z} \in U) < 1$ for all linear subspaces $U \subseteq \mathbb{R}^p$ of dimension $p - 1$.*

*Assuming that (MOM) holds such that (COV) is well-defined, consider the following statements:*

    *(a)* $\mathrm{FRK}(p)$ *holds.*
    *(b)* $\mathrm{FRK}(n)$ *holds for all* $n \geq 1$.
    *(c)* *(COV) holds.*
    *(d) There exists a fixed deterministic matrix* $\widetilde{\boldsymbol{X}} \in \mathbb{R}^{p \times d}$ *such that* $\det(\phi(\widetilde{\boldsymbol{X}})) \neq 0$.

*We have (a)* $\Leftrightarrow$ *(b)* $\Rightarrow$ *(c)* $\Rightarrow$ *(d). Furthermore, if* $\boldsymbol{x} \in \mathbb{R}^d$ *has a Lebesgue density and* $\phi$ *is analytic, then (a) – (d) are equivalent.*

With this in mind, we can characterize the assumptions now:

- The assumption (INT) is standard (see e.g. Section 1.6 in Györfi et al., 2002) and guarantees $R_P(f_P^*) < \infty$, such that the excess risk is well-defined.
- The assumption (NOI) is required to ensure the existence of sufficient label noise. Importantly, Lemma H.1 shows that (NOI), i.e. $\mathrm{Var}(y|\boldsymbol{z}) \geq \sigma^2$ almost surely over $\boldsymbol{z}$, holds if $\mathrm{Var}(y|\boldsymbol{x}) \geq \sigma^2$ almost surely over $\boldsymbol{x}$. All Double Descent papers from Section 1 make the stronger assumption that the distribution of $y - \mathbb{E}(y|\boldsymbol{x})$ is independent of $\boldsymbol{x}$ or even a fixed Gaussian.
- The assumption (MOM) can be reformulated as $\mathbb{E}\|\boldsymbol{z}\|_2^2 = \mathbb{E}\|\phi(\boldsymbol{x})\|_2^2 = \mathbb{E}k(\boldsymbol{x}, \boldsymbol{x}) < \infty$. For example, if $k$ or equivalently $\phi$ are bounded, or if $\phi$ is continuous and $\|\boldsymbol{x}\|_2$ is bounded, then (MOM) is satisfied. Such assumptions are frequently imposed (see e.g. Chapters 6, 7, 8, 10 in Györfi et al., 2002). In this sense, (MOM) is a standard assumption.
- The assumptions (COV) and $\mathrm{FRK}(n)$ are implied by $\mathrm{FRK}(p)$ and are even equivalent to $\mathrm{FRK}(p)$ in the underparameterized case $p \leq n$. In the following, we will therefore focus on proving $\mathrm{FRK}(p)$ for various $\phi$ and $P_X$. In the case $p = n$, $\mathrm{FRK}(p)$ ensures that $f_{\boldsymbol{X}, \boldsymbol{y}}$ almost surely interpolates the data, or equivalently that the kernel matrix $k(\boldsymbol{X}, \boldsymbol{X})$ almost surely has full rank. Importantly, assuming $\mathrm{FRK}(p)$ is weaker than assuming a strictly positive definite kernel, since strictly positive definite kernels require $p = \infty$. Example D.1 shows that the assumption (FRK) in Theorem 3 cannot be removed.

For the analytic function $\phi = \mathrm{id}$ with $d = p$, Proposition 6 yields a simple sufficient criterion: If $\boldsymbol{z}$ has a Lebesgue density, then (FRK) holds for all $n$. This assumption is already more realistic than assuming i.i.d. components. However, Proposition 6 is also very useful for other analytic feature maps, as we will see in the remainder of this section.

**Remark 7.** Suppose that $\phi \not\equiv 0$ is analytic, $\boldsymbol{x}$ has a Lebesgue density and (INT), (MOM) and (NOI) are satisfied. If (d) in Proposition 6 does not hold, there exists $\tilde{p} < p$ such that the lower bound from Theorem 3 holds with $p$ replaced by $\tilde{p}$: Let $U := \mathrm{Span}\{\phi(\boldsymbol{x}) \mid \boldsymbol{x} \in \mathbb{R}^d\}$. Since $\phi \not\equiv 0$, $\tilde{p} := \dim U \geq 1$. Moreover, (d) holds iff $\dim U = p$. Take any isometric isomorphism $\psi : U \to \mathbb{R}^{\tilde{p}}$ and define the feature map $\tilde{\phi} : \mathbb{R}^d \to \mathbb{R}^{\tilde{p}}, \boldsymbol{x} \mapsto \psi(\phi(\boldsymbol{x}))$. Then, $\tilde{\phi}$ is analytic since $\psi$ is linear, and $\tilde{\phi}$ satisfies (d), hence Theorem 3 can be applied to $\tilde{\phi}$. However, $\phi$ and $\tilde{\phi}$ lead to the same kernel $k$ since $\psi$ is isometric, hence to the same estimator $f_{\boldsymbol{X}, \boldsymbol{y}}$ by Eq. (2) and hence to the same $\mathcal{E}_{\mathrm{Noise}}$. ◀

**Proposition 8** (Polynomial kernel). *Let* $m, d \geq 1$ *and* $c > 0$. *For* $\boldsymbol{x}, \tilde{\boldsymbol{x}} \in \mathbb{R}^d$, *define the polynomial kernel* $k(\boldsymbol{x}, \tilde{\boldsymbol{x}}) := (\boldsymbol{x}^\top \tilde{\boldsymbol{x}} + c)^m$. *Then, there exists a feature map* $\phi : \mathbb{R}^d \to \mathbb{R}^p$, $p := \binom{m+d}{m}$, *such that:*

    *(a)* $k(\boldsymbol{x}, \tilde{\boldsymbol{x}}) = \phi(\boldsymbol{x})^\top \phi(\tilde{\boldsymbol{x}})$ *for all* $\boldsymbol{x}, \tilde{\boldsymbol{x}} \in \mathbb{R}^d$, *and*
    *(b) if* $\boldsymbol{x} \in \mathbb{R}^d$ *has a Lebesgue density and we use* $\boldsymbol{z} = \phi(\boldsymbol{x})$, *then (FRK) is satisfied for all* $n$.

Proposition 8 says that the lower bound from Theorem 3 holds for ridgeless kernel regression with the polynomial kernel with $p := \binom{m+d}{m}$ if $\boldsymbol{x}$ has a Lebesgue density and $\mathbb{E}\|\boldsymbol{z}\|_2^2 = \mathbb{E}k(\boldsymbol{x}, \boldsymbol{x}) = \mathbb{E}(\|\boldsymbol{x}\|_2^2 + c)^m < \infty$. The proof of Proposition 8 can be extended to the case $c = 0$, where one needs to choose $p = \binom{m+d-1}{m}$. In general, we discuss in Appendix K that Theorem 3 can often be applied to ridgeless kernel regression, where $p$ needs to be chosen as the minimal feature space dimension for which $k$ can still be represented.

We can also extend our theory to analytic random feature maps:

**Proposition 9** (Random feature maps). *Consider feature maps* $\phi_{\boldsymbol{\theta}} : \mathbb{R}^d \to \mathbb{R}^p$ *with (random) parameter* $\boldsymbol{\theta} \in \mathbb{R}^q$. *Suppose the map* $(\boldsymbol{\theta}, \boldsymbol{x}) \mapsto \phi_{\boldsymbol{\theta}}(\boldsymbol{x})$ *is analytic and that* $\boldsymbol{\theta}$ *and* $\boldsymbol{x}$ *are independent*

*and have Lebesgue densities. If there exist fixed $\widetilde{\boldsymbol{\theta}} \in \mathbb{R}^q, \widetilde{\boldsymbol{X}} \in \mathbb{R}^{p \times d}$ with $\det(\phi_{\widetilde{\boldsymbol{\theta}}}(\widetilde{\boldsymbol{X}})) \neq 0$, then almost surely over $\boldsymbol{\theta}$, (FRK) holds for all $n$ for $\boldsymbol{z} = \phi_{\boldsymbol{\theta}}(\boldsymbol{x})$.*

In Appendix C, we demonstrate that Proposition 9 can be used to computationally verify (FRK) for analytic random feature maps.

Up until now, we have assumed that $\boldsymbol{x}$ has a Lebesgue density. It is desirable to weaken this assumption, such that $\boldsymbol{x}$ can, for example, be concentrated on a submanifold of $\mathbb{R}^d$. It is necessary for $\mathrm{FRK}(p)$ with $p \geq 2$ that $\boldsymbol{x}$ is nonatomic, such that the $\boldsymbol{x}_i$ are distinct with probability one. In general, this is not sufficient: For example, if $\phi = \mathrm{id}$ and $\boldsymbol{x}$ lives on a proper linear subspace of $\mathbb{R}^d$, $\mathrm{FRK}(p)$ is not satisfied. Perhaps surprisingly, we will show next that for random neural network feature maps, it is indeed sufficient that $\boldsymbol{x}$ is nonatomic.[2] Especially, our lower bound in Corollary 4 thus applies to a large class of feedforward neural networks where only the last layer is trained (and initialized to zero, such that gradient descent converges to the Moore-Penrose pseudoinverse).

**Theorem 10** (Random neural networks). *Let $d, p, L \geq 1$, let $\sigma : \mathbb{R} \to \mathbb{R}$ be analytic and let the layer sizes be $d_0 = d$, $d_1, \ldots, d_{L-1} \geq 1$ and $d_L = p$. Let $\boldsymbol{W}^{(l)} \in \mathbb{R}^{d_{l+1} \times d_l}$ for $l \in \{0, \ldots, L-1\}$ be random variables and consider the two cases where*

*(a) $\sigma$ is not a polynomial with less than $p$ nonzero coefficients, $\boldsymbol{\theta} := (\boldsymbol{W}^{(0)}, \ldots, \boldsymbol{W}^{(L-1)})$ and the random feature map $\phi_{\boldsymbol{\theta}} : \mathbb{R}^d \to \mathbb{R}^p$ is recursively defined by*

$$\phi(\boldsymbol{x}^{(0)}) := \boldsymbol{x}^{(L)}, \quad \boldsymbol{x}^{(l+1)} := \sigma(\boldsymbol{W}^{(l)}\boldsymbol{x}^{(l)}) .$$

*(b) $\sigma$ is not a polynomial of degree $< p - 1$, $\boldsymbol{\theta} := (\boldsymbol{W}^{(0)}, \ldots, \boldsymbol{W}^{(L-1)}, \boldsymbol{b}^{(0)}, \ldots, \boldsymbol{b}^{(L-1)})$ with random variables $\boldsymbol{b}^{(l)} \in \mathbb{R}^{d_{l+1}}$ for $l \in \{0, \ldots, L-1\}$, and the random feature map $\phi_{\boldsymbol{\theta}} : \mathbb{R}^d \to \mathbb{R}^p$ is recursively defined by*

$$\phi(\boldsymbol{x}^{(0)}) := \boldsymbol{x}^{(L)}, \quad \boldsymbol{x}^{(l+1)} := \sigma(\boldsymbol{W}^{(l)}\boldsymbol{x}^{(l)} + \boldsymbol{b}^{(l)}) .$$

*In both cases, if $\boldsymbol{\theta}$ has a Lebesgue density and $\boldsymbol{x}$ is nonatomic, then (FRK) holds for all $n$ and almost surely over $\boldsymbol{\theta}$.*

The assumptions of Theorem 10 on $\boldsymbol{\theta}$ are satisfied by the classical initialization methods of He et al. (2015) and Glorot & Bengio (2010). Possible choices of $\sigma$ are presented in Table 1. A statement similar to Theorem 10 has been proven in Lemma 4.4 in Nguyen & Hein (2017). However, their statement only applies to networks with bias and to a more restricted class of activation functions: For example, the activation functions RBF, GELU, SiLU/Swish, Mish, sin and cos are not covered by their assumptions. In Appendix E, we explain that the proofs of many other theorems in the literature similar to Theorem 10 for single-layer networks are incorrect.

Table 1: Some examples of analytic activation functions that are not polynomials. The CDF of $\mathcal{N}(0, 1)$ is denoted by $\Phi$. Other non-polynomial analytic activation functions are $\sin$, $\cos$ or $\mathrm{erf}$.

| Activation function | Equation |
|---|---|
| Sigmoid | $\mathrm{sigmoid}(x) = 1/(1 + e^{-x})$ |
| Hyperbolic Tangent | $\tanh(x) = (e^x - e^{-x})/(e^x + e^{-x})$ |
| Softplus | $\mathrm{softplus}(x) = \log(1 + e^x)$ |
| RBF | $\mathrm{RBF}(x) = \exp(-\beta x^2)$ |
| GELU (Hendrycks & Gimpel, 2016) | $\mathrm{GELU}(x) = x\Phi(x)$ |
| SiLU (Elfwing et al., 2018) | $\mathrm{SiLU}(x) = x\,\mathrm{sigmoid}(x)$ |
| Swish (Ramachandran et al., 2017) | $\mathrm{Swish}(x) = x\,\mathrm{sigmoid}(\beta x)$ |
| Mish (Misra, 2019) | $\mathrm{Mish}(x) = x\tanh(\mathrm{softplus}(x))$ |

---

[2]This appears to be a convenient consequence of randomizing the feature map: For each fixed feature map $\phi_{\widetilde{\boldsymbol{\theta}}}$, there may be an exceptional set $E_{\widetilde{\boldsymbol{\theta}}}$ of nonatomic input distributions $P_X$ for which $\mathrm{FRK}(p)$ is not satisfied. However, Theorem 10 shows that for each nonatomic input distribution $P_X$, the set $\{\widetilde{\boldsymbol{\theta}} \mid P_X \in E_{\widetilde{\boldsymbol{\theta}}}\}$ is a Lebesgue null set. For (deterministic) feature maps where it is not possible to prove $\mathrm{FRK}(p)$ for all nonatomic $P_X$, there is a trick that works in certain cases: If $\boldsymbol{x}$ lives on a submanifold (e.g. a sphere), we might be able to write $\boldsymbol{x} = \psi(\boldsymbol{v})$, where $\boldsymbol{v}$ has a Lebesgue density and $\psi$ is analytic (e.g. the stereographic projection for the sphere). Then, we can apply our theory to the analytic feature map $\phi_{\boldsymbol{\theta}} \circ \psi$.

Under the assumptions of Theorem 10, if $\|\boldsymbol{x}\|_2$ is bounded, (MOM) holds for all $\boldsymbol{\theta}$. This follows since any analytic function $\phi$ is also continuous, and continuous functions preserve boundedness. However, the activation functions from Table 1 even satisfy $|\sigma(x)| \leq a|x| + b$ for some $a, b \geq 0$ and all $x \in \mathbb{R}$. In this case, it is not hard to see that (MOM) already holds for all $\boldsymbol{\theta}$ if $\mathbb{E}\|\boldsymbol{x}\|_2^2 < \infty$.

Theorem 10 does not hold for ReLU, ELU (Clevert et al., 2015), SELU (Klambauer et al., 2017) or other activation functions with a perfectly linear part. To see this, observe that with nonzero probability, all weights and inputs are positive. In this case, the feature map acts as a linear map from $\mathbb{R}^d$ to $\mathbb{R}^p$, and if $d < p = n$, the output matrix $\phi(\boldsymbol{X})$ cannot be invertible.

In Appendix I, we show that (FRK) is satisfied for random Fourier features if $\boldsymbol{x}$ is nonatomic and the frequency distribution (i.e. the Fourier transform of the shift-invariant kernel) has a Lebesgue density.

## 6 Quality of the Lower Bound

In this section, we discuss how sharp the lower bound from Theorem 3 is. To this end, we assume that $\text{Var}(y|\boldsymbol{z}) = \sigma^2$ almost surely over $\boldsymbol{z}$.

In their Lemma 3, Hastie et al. (2019) consider the case where $\boldsymbol{z}$ has i.i.d. entries with zero mean, unit variance and finite fourth moment. They then use the Marchenko-Pastur law to show in the limit $p, n \to \infty, p/n \to \gamma > 1$ that $\mathcal{E}_{\text{Noise}} \to \sigma^2 \frac{1}{\gamma-1}$. In this limit, our lower bound shows

$$\mathcal{E}_{\text{Noise}} \geq \sigma^2 \frac{n}{p+1-n} = \sigma^2 \frac{1}{p/n + 1/n - 1} \to \sigma^2 \frac{1}{\gamma - 1} \,,$$

hence, in this sense, our overparameterized bound is asymptotically sharp. An analogous argument shows that the underparameterized bound is also asymptotically sharp. In order to better understand to which extent our lower bound is non-asymptotically sharp in the over- and underparameterized regimes, we explicitly compute $\mathcal{E}_{\text{Noise}} = \sigma^2 \mathbb{E}_{\boldsymbol{Z}} \text{tr}((\boldsymbol{Z}^+)^\top \boldsymbol{\Sigma} \boldsymbol{Z}^+)$ (cf. Theorem 3) for some distributions $P_Z$:

**Theorem 11.** *Let $P_Z = \mathcal{U}(\mathbb{S}^{p-1})$. Then, $P_Z$ satisfies the assumptions (MOM), (COV) and (FRK) for all $n$ with $\boldsymbol{\Sigma} = \frac{1}{p}\boldsymbol{I}_p$. Moreover, for $n \geq p = 1$ or $p \geq n \geq 1$, we can compute*

$$\mathbb{E}_{\boldsymbol{Z}} \text{tr}((\boldsymbol{Z}^+)^\top \boldsymbol{\Sigma} \boldsymbol{Z}^+) = \begin{cases} \frac{1}{n} & \text{if } n \geq p = 1, \\ \frac{1}{p} & \text{if } p \geq n = 1, \\ \infty & \text{if } 2 \leq n \leq p \leq n+1, \\ \frac{n}{p-1-n} \cdot \frac{p-2}{p} & \text{if } 2 \leq n \leq n+2 \leq p. \end{cases}$$

The formulas in the next theorem have already been computed by Breiman & Freedman (1983) for $p \leq n - 2$ and by Belkin et al. (2019b) for general $p$. Our alternative proof circumvents a technical problem in their proof for $p \in \{n-1, n, n+1\}$, cf. Appendix J.

**Theorem 12.** *Let $P_Z = \mathcal{N}(0, \boldsymbol{I}_p)$. Then, $P_Z$ satisfies the assumptions (MOM), (COV) and (FRK) for all $n$ with $\boldsymbol{\Sigma} = \boldsymbol{I}_p$. Moreover, for $n, p \geq 1$,*

$$\mathbb{E}_{\boldsymbol{Z}} \text{tr}((\boldsymbol{Z}^+)^\top \boldsymbol{\Sigma} \boldsymbol{Z}^+) = \begin{cases} \frac{n}{p-1-n} & \text{if } p \geq n+2, \\ \infty & \text{if } p \in \{n-1, n, n+1\}, \\ \frac{p}{n-1-p} & \text{if } p \leq n-2. \end{cases}$$

For $P_Z = \mathcal{N}(0, \boldsymbol{I}_p)$ and the lower bound from Theorem 3, the formulas for the under- and overparameterized cases can be obtained from each other by switching the roles of $n$ and $p$. Numerical data strongly suggests that this symmetry does *not* hold exactly for $P_Z = \mathcal{U}(\mathbb{S}^{p-1})$.

For $p \geq n+2$, we can relate our lower bound (Theorem 3), the result for the sphere (Theorem 11) and the result for the Gaussian distribution (Theorem 12) as follows:

$$\frac{n}{p+1-n} = \frac{n}{p-1-n} \cdot \frac{(p+1-n)-2}{p+1-n} \leq \frac{n}{p-1-n} \cdot \frac{p-2}{p} < \frac{n}{p-1-n} \,.$$

These values are also shown in Figure 1 together with empirical values of NN feature maps specifically optimized to minimize $\mathcal{E}_{\text{Noise}}$. Since $\boldsymbol{\Sigma}$ affects $\mathcal{E}_{\text{Noise}}$ only for $p > n$ (cf. Remark G.1), it is

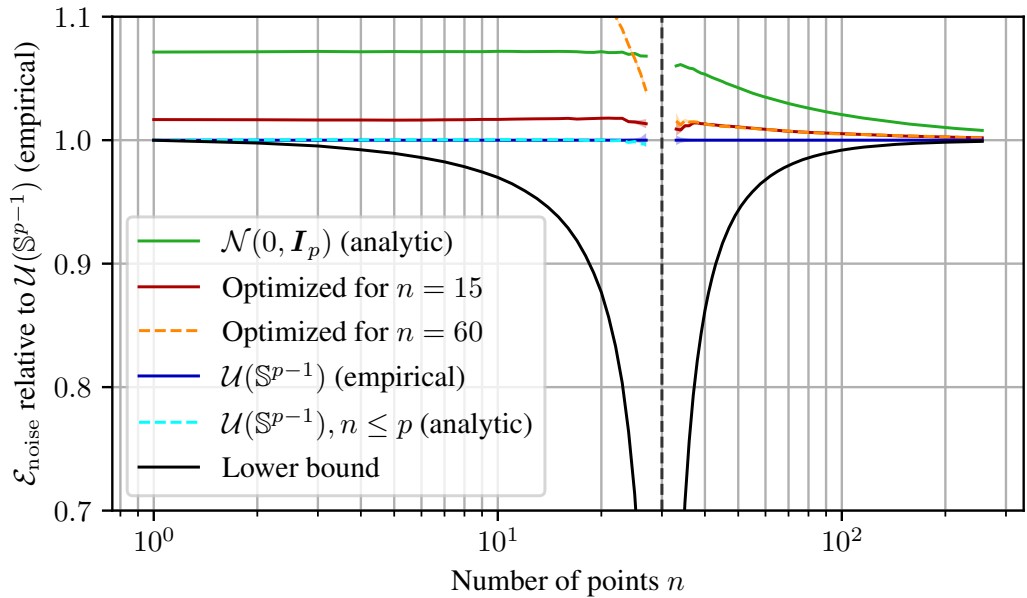

Figure 1: Various estimates and bounds for $\mathcal{E}_{\text{Noise}}$ relative to $\mathcal{E}_{\text{Noise}}$ for $P_Z = \mathcal{U}(\mathbb{S}^{p-1})$, using $\text{Var}(y|\boldsymbol{z}) = 1$ and $p = 30$. The optimized curves correspond to multi-layer NN feature maps whose parameters were trained to minimize $\mathcal{E}_{\text{Noise}}$ for $n = 15$ or $n = 60$. More experiments and details on the setup can be found in Appendix C. We do not plot estimates for $n \in \{28, \ldots, 32\}$ since they have high estimation variances.

not surprising that the feature map optimized for $n = 60 > 30 = p$ performs badly in the over-parameterized regime. The results in Figure 1 support the hypothesis that among all $P_Z$ satisfying (MOM), (COV) and (FRK), $P_Z = \mathcal{U}(\mathbb{S}^{p-1})$ minimizes $\mathbb{E}_{\boldsymbol{Z}} \text{tr}((\boldsymbol{Z}^+)^\top \boldsymbol{\Sigma} \boldsymbol{Z}^+)$. The plausibility of this hypothesis is further discussed in Remark G.3. We can prove the hypothesis for $n = 1$ or $p = 1$ since in this case, the results from Theorem 11 are equal to the lower bound from Theorem 3.

When using a continuous feature map $\phi : \mathbb{R}^d \to \mathbb{R}^p$ with $d \leq p - 2$, it seems to be unclear at first whether the low $\mathcal{E}_{\text{Noise}}$ of $P_Z = \mathcal{U}(\mathbb{S}^{p-1})$ can even be achieved. We show in Proposition J.2 that this is possible using space-filling curve feature maps.

The results in this section and in Appendix C show that while our lower bound presumably does not fully capture the height of the interpolation peak at $p \approx n$, it is quite sharp in the practically relevant regimes $p \gg n$ and $p \ll n$ irrespective of $d$.

### ACKNOWLEDGMENTS

The author would like to thank Ingo Steinwart for proof-reading most of this paper and for providing helpful comments. Funded by Deutsche Forschungsgemeinschaft (DFG, German Research Foundation) under Germany's Excellence Strategy - EXC 2075 - 390740016. The author thanks the International Max Planck Research School for Intelligent Systems (IMPRS-IS) for support.

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

## A OVERVIEW

The appendix is structured as follows: In Appendix B, we provide an overview over some matrix identities used throughout the paper. We provide additional numerical experiments for various (random) feature maps in Appendix C. The counterexample given in Appendix D shows that the assumption (FRK) cannot be omitted from Theorem 3. In Appendix E, we provide an overview over full-rank theorems for random neural networks in the literature and explain why their proofs are incorrect. We then prove our main results in Appendix F before discussing consequences in Appendix G. In Appendix H, we give proofs for the theorems and propositions from Section 5. As an addition, we prove (FRK) for random Fourier features in Appendix I. Finally, proofs for the statements from Section 6 are given in Appendix J.

Whenever we prove theorems or propositions from the main paper (like Theorem 3) in the Appendix, we repeat their statement before the proof for convenience. In contrast, new theorems or propositions are numbered according to the section they are stated in, e.g. Proposition I.1.

## B MATRIX ALGEBRA

In the following, we will present some facts about matrices that are relevant to this paper. For a general reference, we refer to textbooks on the subject (e.g. Golub & Van Loan, 1989; Bhatia, 2013).

Let $n, p \geq 1$ and let $k := \min\{n, p\}$. The singular value decomposition (SVD) of a matrix $\boldsymbol{A} \in \mathbb{R}^{n \times p}$ is a decomposition $\boldsymbol{A} = \boldsymbol{U}\boldsymbol{D}\boldsymbol{V}^\top$ into orthogonal matrices $\boldsymbol{U} \in \mathbb{R}^{n \times k}, \boldsymbol{V} \in \mathbb{R}^{p \times k}$ with $\boldsymbol{U}^\top \boldsymbol{U} = \boldsymbol{V}^\top \boldsymbol{V} = \boldsymbol{I}_k$ and a diagonal matrix $\boldsymbol{D} \in \mathbb{R}^{k \times k}$ with non-negative diagonal elements $s_1(\boldsymbol{A}) \geq \ldots \geq s_k(\boldsymbol{A})$ called *singular values*.

For a given symmetric square matrix $\boldsymbol{A} \in \mathbb{R}^{n \times n}$ with eigenvalues $\lambda_1(\boldsymbol{A}) \geq \ldots \geq \lambda_n(\boldsymbol{A})$, the trace satisfies

$$\operatorname{tr}(\boldsymbol{A}) = \sum_{i=1}^n A_{ii} = \sum_{i=1}^n \lambda_i \ .$$

The trace is linear and invariant under cyclical permutations. We use this multiple times in arguments of the following type: If $\boldsymbol{v} \in \mathbb{R}^p$ and $\boldsymbol{A} \in \mathbb{R}^{p \times p}$ are stochastically independent (e.g. because $\boldsymbol{A}$ is constant), we can write

$$\mathbb{E}_{\boldsymbol{v}}\boldsymbol{v}^\top \boldsymbol{A}\boldsymbol{v} = \mathbb{E}_{\boldsymbol{v}} \operatorname{tr}(\boldsymbol{v}^\top \boldsymbol{A}\boldsymbol{v}) = \mathbb{E}_{\boldsymbol{v}} \operatorname{tr}(\boldsymbol{A}\boldsymbol{v}\boldsymbol{v}^\top) = \operatorname{tr}(\boldsymbol{A}\mathbb{E}_{\boldsymbol{v}}\boldsymbol{v}\boldsymbol{v}^\top) \ .$$

Moreover, if $\boldsymbol{A} \succ 0$, then for all $i \in [n]$, $\lambda_i(\boldsymbol{A}) > 0$ and we have

$$\lambda_{n+1-i}(\boldsymbol{A}^{-1}) = \frac{1}{\lambda_i(\boldsymbol{A})} \ .$$

For a positive definite matrix $\boldsymbol{\Sigma} \in \mathbb{R}^{p \times p}$, the SVD and the eigendecomposition coincide as $\boldsymbol{\Sigma} = \boldsymbol{U}\operatorname{diag}(\lambda_1(\boldsymbol{\Sigma}), \ldots, \lambda_p(\boldsymbol{\Sigma}))\boldsymbol{U}^\top$ and we can define the inverse square root as

$$\boldsymbol{\Sigma}^{-1/2} := \boldsymbol{U}\operatorname{diag}(\lambda_1(\boldsymbol{\Sigma})^{-1/2}, \ldots, \lambda_p(\boldsymbol{\Sigma})^{-1/2})\boldsymbol{U}^\top \succ 0 \ ,$$

which is the unique s.p.d. matrix satisfying $(\boldsymbol{\Sigma}^{-1/2})^2 = \boldsymbol{\Sigma}^{-1}$.

By the Courant-Fischer-Weyl theorem, two symmetric matrices $\boldsymbol{A}, \boldsymbol{B} \in \mathbb{R}^{n \times n}$ with $\boldsymbol{A} \preceq \boldsymbol{B}$ satisfy

$$\lambda_i(\boldsymbol{A}) = \max_{\substack{\mathcal{V} \subseteq \mathbb{R}^d \text{ subspace} \\ \dim \mathcal{V} = i}} \min_{\substack{\boldsymbol{v} \in \mathcal{V} \\ \|\boldsymbol{v}\|_2 = 1}} \boldsymbol{v}^\top \boldsymbol{A}\boldsymbol{v} \leq \max_{\substack{\mathcal{V} \subseteq \mathbb{R}^d \text{ subspace} \\ \dim \mathcal{V} = i}} \min_{\substack{\boldsymbol{v} \in \mathcal{V} \\ \|\boldsymbol{v}\|_2 = 1}} \boldsymbol{v}^\top \boldsymbol{B}\boldsymbol{v} = \lambda_i(\boldsymbol{B}) \ .$$

Let $\boldsymbol{A} \in \mathbb{R}^{n \times p}$. The Moore-Penrose pseudoinverse $\boldsymbol{A}^+$ of $\boldsymbol{A}$ satisfies the following relations (see e.g. Section 1.1.1 in Wang et al., 2018):

- Suppose $\boldsymbol{A}$ has the SVD $\boldsymbol{A} = \boldsymbol{U}\boldsymbol{D}\boldsymbol{V}^\top$, where $\boldsymbol{D} = \operatorname{diag}(s_1, \ldots, s_k)$, $k := \min\{n, p\}$. Using the convention $1/0 := 0$, we can write $\boldsymbol{A}^+ = \boldsymbol{V}\boldsymbol{D}^+\boldsymbol{U}^\top$, where $\boldsymbol{D}^+ := \operatorname{diag}(1/s_1, \ldots, 1/s_k)$.

- $\boldsymbol{A}^+ = (\boldsymbol{A}^\top \boldsymbol{A})^+ \boldsymbol{A}^\top = \boldsymbol{A}^\top (\boldsymbol{A}\boldsymbol{A}^\top)^+$.
- If $\boldsymbol{A}$ is invertible, then $\boldsymbol{A}^+ = \boldsymbol{A}^{-1}$.
- $\boldsymbol{A}^+ (\boldsymbol{A}^+)^\top = (\boldsymbol{A}^\top \boldsymbol{A})^+$.

We will also use a basic fact on the Schur complement that, for example, is outlined in Appendix A.5.5 in Boyd & Vandenberghe (2004): If

$$0 \prec \boldsymbol{A} = \begin{pmatrix} \boldsymbol{A}_{11} & \boldsymbol{A}_{12} \\ \boldsymbol{A}_{21} & \boldsymbol{A}_{22} \end{pmatrix} \in \mathbb{R}^{(m_1+m_2)\times(m_1+m_2)} \ ,$$

then $\boldsymbol{A}_{22} \succ 0$ and $\boldsymbol{A}_{11} - \boldsymbol{A}_{12}\boldsymbol{A}_{22}^{-1}\boldsymbol{A}_{21} \succ 0$ and the top-left $m_1 \times m_1$ block of $\boldsymbol{A}^{-1}$ is given by $(\boldsymbol{A}_{11} - \boldsymbol{A}_{12}\boldsymbol{A}_{22}^{-1}\boldsymbol{A}_{21})^{-1}$.

## C EXPERIMENTS

In the following, we experimentally investigate $\mathcal{E}_{\mathrm{Noise}}$ for different (random) feature maps. We will first give an overview over the plots and then explain the details of how they were generated and some implications. More details can be found in the provided code. All empirical curves show one estimated standard deviation of the mean estimator as a shaded area around the estimated mean, but this standard deviation is sometimes too small to be visible. We assume $\mathrm{Var}(y|\boldsymbol{z}) = 1$ almost surely over $\boldsymbol{z}$ such that (I) in Theorem 3 holds with equality with $\sigma^2 = 1$.

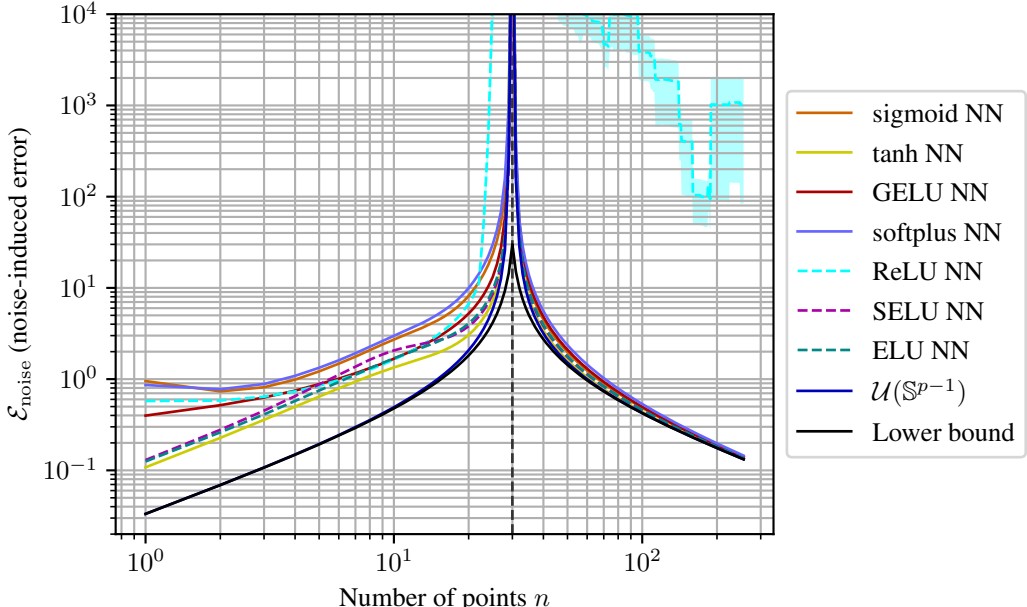

Figure C.1: Estimated $\mathcal{E}_{\mathrm{Noise}}$ for random neural network feature maps (cf. Theorem 10) with different activation functions and $d_0 = d = 10, d_1 = d_2 = 256, d_3 = p = 30$. We include the results from $P_Z = \mathcal{U}(\mathbb{S}^{d-1})$ as comparison, cf. Section 6.

Figure C.1 shows $\mathcal{E}_{\mathrm{Noise}}$ for random three-layer neural network feature maps with $p = 30$, different activation functions and varying $n$. Note that all neural networks produce higher $\mathcal{E}_{\mathrm{Noise}}$ than $P_Z = \mathcal{U}(\mathbb{S}^{p-1})$. The effect of non-isotropic covariance matrices in the overparameterized regime can be clearly seen when comparing Figure C.1 to Figure C.2, where features have been whitened separately for each set of random parameters $\boldsymbol{\theta}$, cf. Remark G.1. Figure C.3 then shows $\mathcal{E}_{\mathrm{Noise}}$ for $n = 30$ and varying $p$.

Note that double descent is usually plotted as a function of the "model complexity" $p$ as in Belkin et al. (2019a), but varying $p$ is only possible when we have a (random) feature map $\phi_{\boldsymbol{\theta}}^{(p)} : \mathbb{R}^d \to \mathbb{R}^p$ for each value of $p$. For the following NTK and polynomial kernels, there is no canonical way to

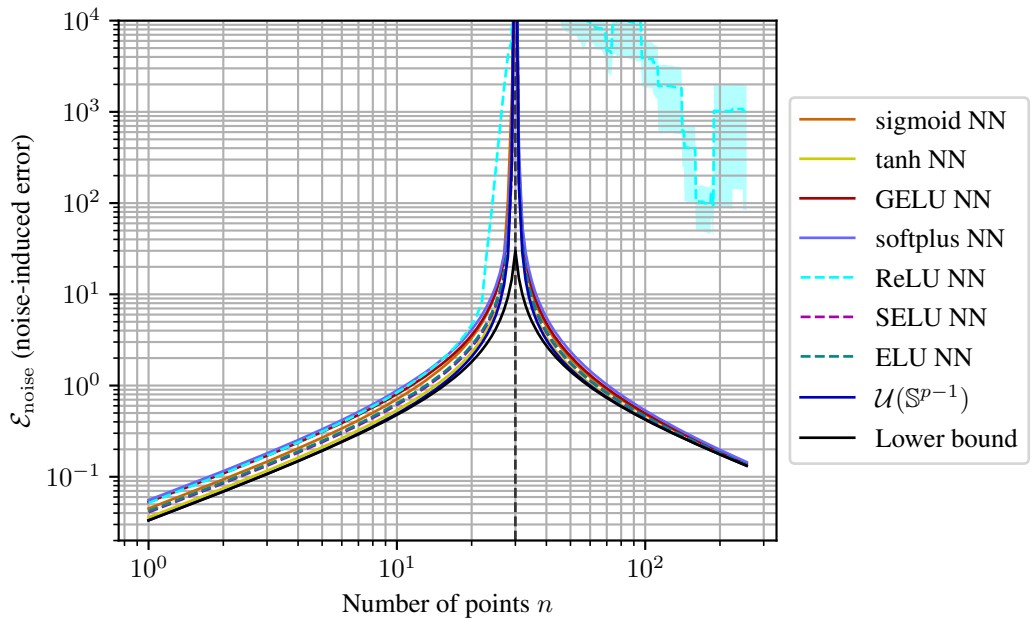

Figure C.2: As in Figure C.1 but with whitened features, i.e. using $\mathbb{E}\,\mathrm{tr}((\boldsymbol{W}\boldsymbol{W}^\top)^{-1})$ in the overparameterized case $n \leq p$, cf. Theorem 3 and Remark G.1.

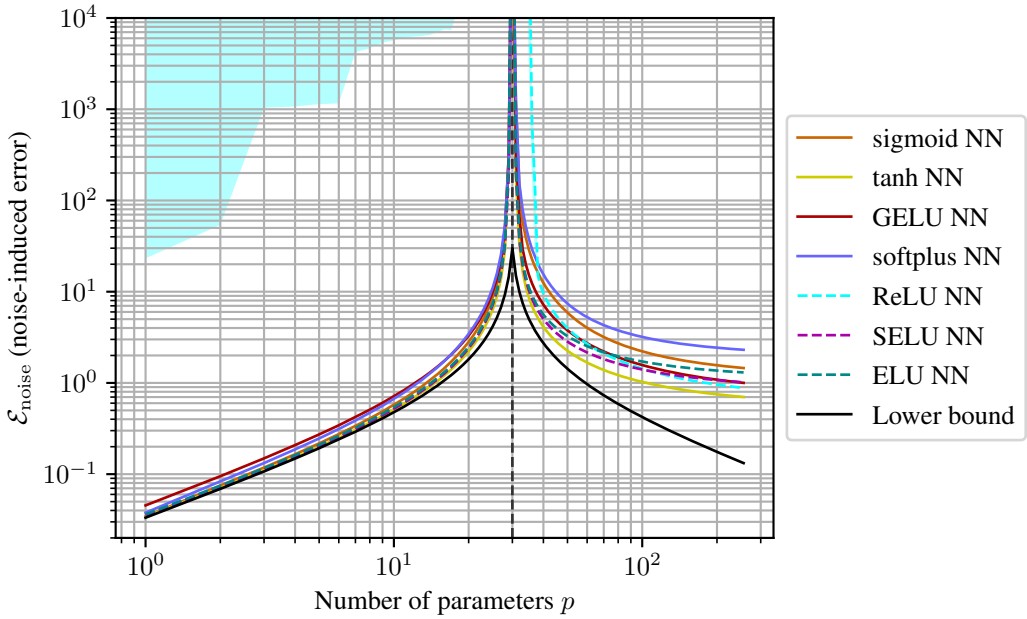

Figure C.3: As in Figure C.1 but with $n = 30$ fixed and varying $d_3 = p$.

define such a sequence of feature maps. For this reason, we will plot their results only with varying $n$. Double descent as a function of the number of samples $n$ has for example been pointed out by Nakkiran et al. (2019) and Nakkiran (2019).

Figure C.4 shows $\mathcal{E}_{\mathrm{Noise}}$ for various random finite-width Neural Tangent Kernels (NTKs), cf. Jacot et al. (2018). These results mostly exhibit larger $\mathcal{E}_{\mathrm{Noise}}$ than the random NN feature maps from Figure C.1, perhaps because of correlations parameter gradients in different layers. However, this

comparison is not really fair since the NTK feature map uses a much smaller underlying neural network and the input dimension $d$ is smaller.

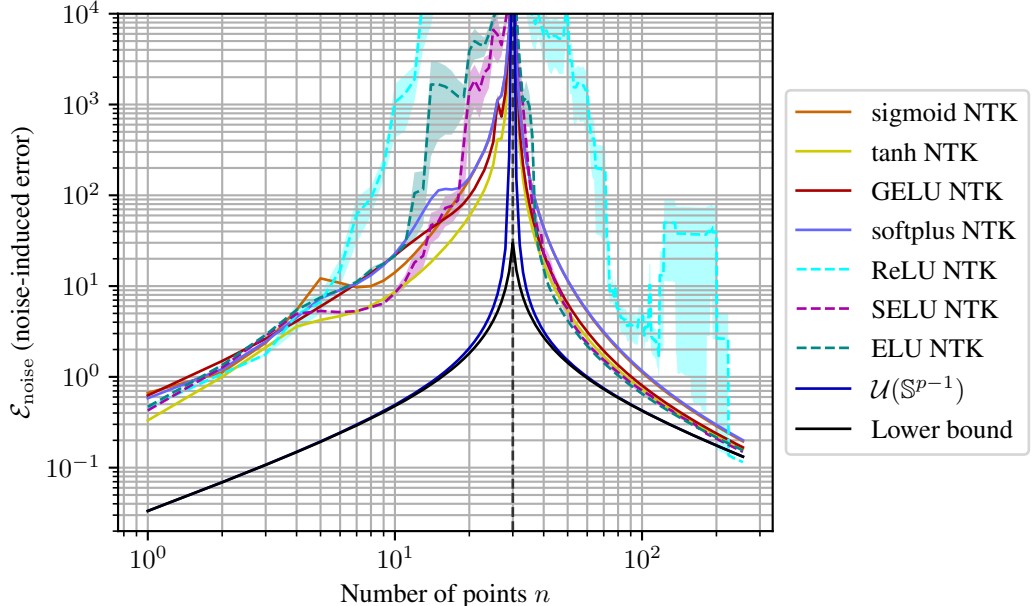

Figure C.4: Estimated $\mathcal{E}_{\text{Noise}}$ for Neural Tangent Kernel (NTK) feature maps given by various random neural networks (cf. Theorem 10) with $d_0 = d = 4, d_1 = 6, d_3 = 1$, resulting in $p = 4 \cdot 6 + 6 \cdot 1 = 30$. We include the empirical results from $P_Z = \mathcal{U}(\mathbb{S}^{d-1})$ as comparison, cf. Section 6.

Figure C.5 and Figure C.6 show $\mathcal{E}_{\text{Noise}}$ for two variants of random Fourier features for two different scalings of the random parameters. Figure C.6 shows that for random Gaussian parameters with large variance (corresponding to an approximated narrow Gaussian kernel), the values of $\mathcal{E}_{\text{Noise}}$ for random Fourier features are very close to the values for $P_Z = \mathcal{U}(\mathbb{S}^{p-1})$. We decided to plot these values relative to each other as in Figure 1, since the curves would overlap in a normal plot like Figure C.5. Note that the the version of random Fourier features with $\sin$ and $\cos$ features automatically yields constant $\|z\|_2$ like for $P_Z = \mathcal{U}(\mathbb{S}^{p-1})$.

Finally, Figure C.7 shows that linear regression with the polynomial kernel is quite sensitive to label noise. We use $p = 35$ for the polynomial kernel since there are no particularly interesting polynomial kernels with $p = 30$.

**Neural Network feature maps** For Figures C.1, C.2 and C.3, we use random neural network feature maps without biases as in Theorem 10 with $d_0 = d = 10, d_1 = d_2 = 256$ and $d_3 = p$. As the input distribution $P_X$, we use $\mathcal{N}(0, \boldsymbol{I}_d)$. We initialize the NN weights independently as $W_{ij}^{(l)} \sim \mathcal{N}(0, 1/V_l)$, where

$$V_l := \begin{cases} d_0 & \text{if } l = 0, \\ d_l \operatorname{Var}_{u \sim \mathcal{N}(0,1)}(\sigma(u)) & \text{if } l \geq 1. \end{cases}$$

Here, $\operatorname{Var}_{u \sim \mathcal{N}(0,1)}(\sigma(u))$ is approximated once by using with $10^4$ samples for $u$. This initialization is similar (and for the ReLU activation essentially equivalent) to the initialization method by He et al. (2015). The initialization variances are chosen such that for fixed input $\boldsymbol{x}$ with $\|\boldsymbol{x}\|_2 \approx 1$, the pre-activations in each layer are approximately $\mathcal{N}(0, 1)$-distributed.

**NTK feature maps** For Figure C.4, we use a NTK parameterization similar to the original one proposed by Jacot et al. (2018), but again with activation-dependent scaling. Our neural network is

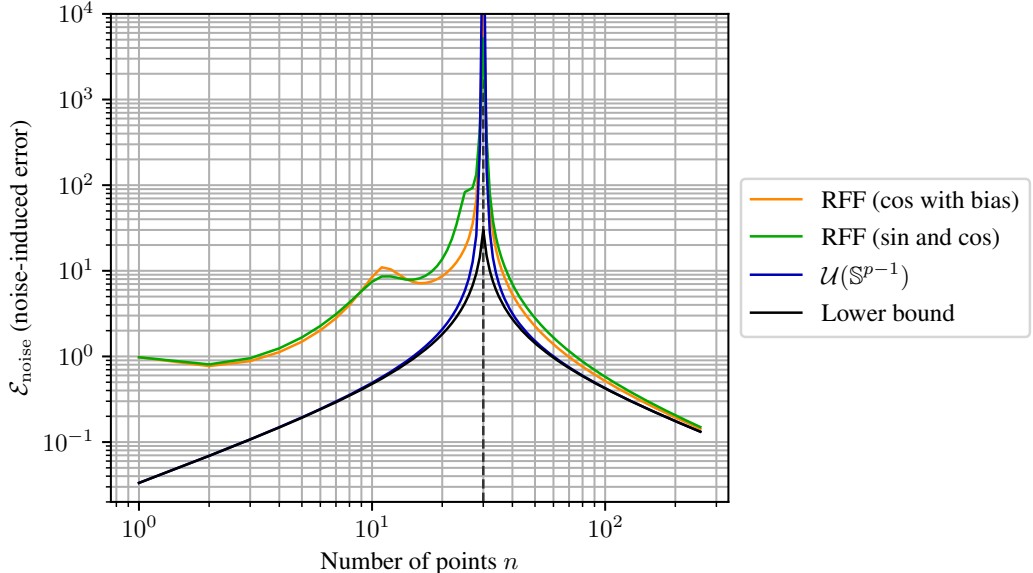

Figure C.5: Estimated $\mathcal{E}_{\text{Noise}}$ for the two versions of random Fourier features described in Appendix I. We use $d = 10$, $P_X = \mathcal{N}(0, \boldsymbol{I}_d)$, $p = 30$ and the weight vector distribution $P_k = \mathcal{N}(0, \frac{1}{p}\boldsymbol{I}_d)$ (cf. Appendix I).

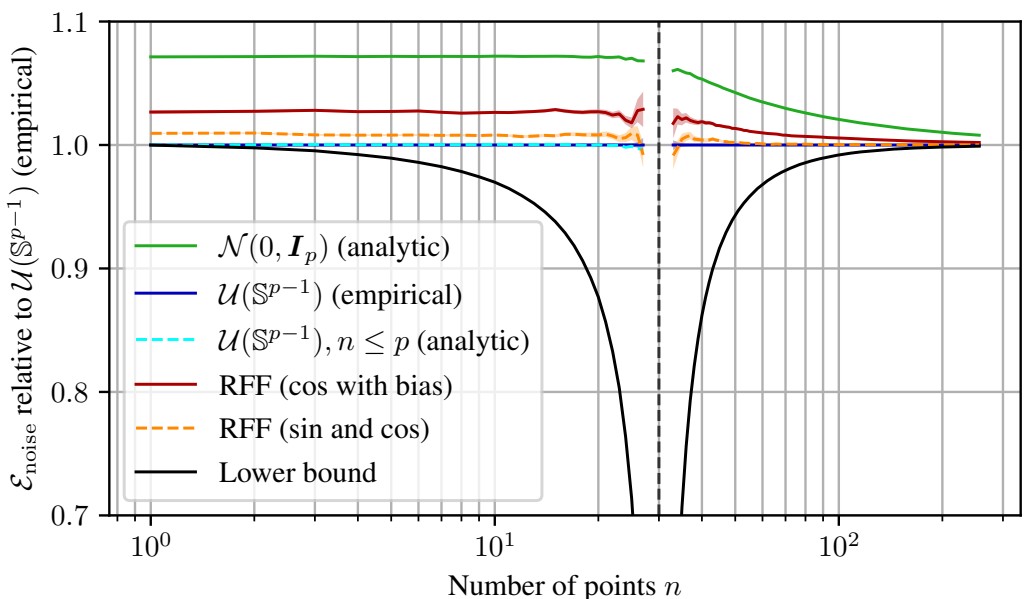

Figure C.6: Estimated $\mathcal{E}_{\text{Noise}}$ for the two versions of random Fourier features described in Appendix I relative to $P_Z = \mathcal{U}(\mathbb{S}^{p-1})$. We use $d = 10$, $P_X = \mathcal{N}(0, \boldsymbol{I}_d)$, $p = 30$ and the weight vector distribution $P_k = \mathcal{N}(0, \boldsymbol{I}_d)$ (cf. Appendix I).

given by[3]

$$\tilde{\phi}_{\boldsymbol{\theta}} : \mathbb{R}^d \to \mathbb{R}^1, \boldsymbol{x} \mapsto \frac{1}{\sqrt{V_1}}\boldsymbol{W}^{(1)}\sigma\left(\frac{1}{\sqrt{V_0}}\boldsymbol{W}^{(0)}\right)$$

---

[3]For random NN feature maps as in Theorem 10, one can interpret the linear regression as being an extra layer on top of the neural network, and therefore the last layer of the feature map should contain an activation function. For NTK feature maps, one can instead interpret the linear regression as performing a "linearized" training of the whole NN, and the whole NN usually does not contain an activation function in the last layer.

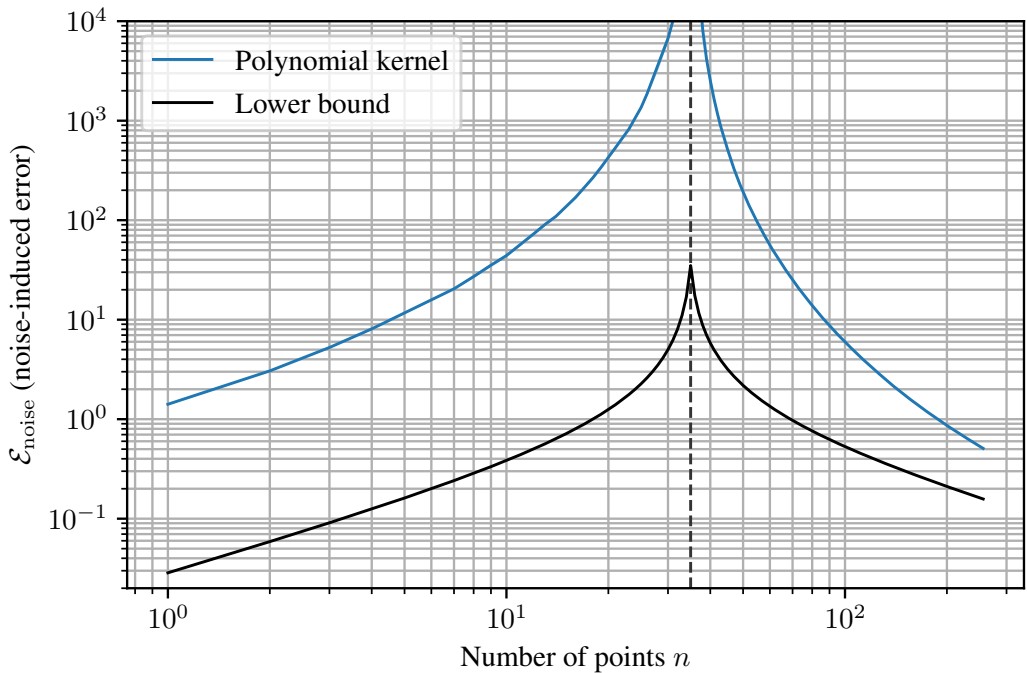

Figure C.7: Estimated $\mathcal{E}_{\mathrm{Noise}}$ for the polynomial kernel (cf. Proposition 8) with $d = 3$, $m = 4$, $c = 1$, $P_X = \mathcal{N}(0, \boldsymbol{I}_d)$, resulting in $p = \binom{m+d}{m} = \binom{7}{4} = 35$.

with $d_0 = d = 4, d_1 = 6, d_2 = 1$ and $\boldsymbol{W}_{ij}^{(l)} \sim \mathcal{N}(0, 1)$ i.i.d. Our input distribution is again $P_X = \mathcal{N}(0, \boldsymbol{I}_d)$. The NTK feature map is then defined as

$$\phi_{\boldsymbol{\theta}} : \mathbb{R}^d \to \mathbb{R}^p, \boldsymbol{x} \mapsto \frac{\partial}{\partial \boldsymbol{\theta}} \tilde{\phi}_{\boldsymbol{\theta}}(\boldsymbol{x}) \,,$$

where $p = 6 \cdot 4 + 1 \cdot 6 = 30$ is the number of parameters in $\boldsymbol{\theta}$. Note that moving the variances $V_l$ outside of the $\boldsymbol{W}^{(l)}$ does not affect the forward pass but only the backward pass (i.e. the derivatives).

While we have not theoretically established the properties (FRK) and (COV) for such feature maps, we can do this experimentally for analytic activation functions like sigmoid, tanh, softplus and GELU: Since the random NTK feature map is a derivative of the analytic random NN feature map, it is also analytic. By Proposition 9, if (FRK) does not hold, then for every fixed $\tilde{\boldsymbol{\theta}}$, the range of $\phi_{\tilde{\boldsymbol{\theta}}}$ must be contained in a proper linear subspace of $\mathbb{R}^p$, and therefore $\boldsymbol{Z} = \phi_{\boldsymbol{\theta}}(\boldsymbol{X})$ never has full rank for $n \geq p$. In this case, the singular value $s_p(\boldsymbol{Z})$ would be zero, and even accounting for numerical errors, the "inverse condition number" $\frac{s_p(\boldsymbol{Z})}{s_1(\boldsymbol{Z})}$ should at most be of the order of 64-bit float machine precision, i.e. around $10^{-16}$. However, among $10^4$ samples of $\boldsymbol{Z}$ for $n := 90 \geq 30 = p$, the maximum observed "inverse condition number" was greater than $10^{-3}$ for all of the activation functions sigmoid, tanh, softplus and GELU.[4] Hence, by Proposition 6 and Proposition 9, we can confidently conclude that (COV) and (FRK) hold for all $n$ almost surely over $\boldsymbol{\theta}$ for this network size and these activation functions.

**Estimation of $\mathcal{E}_{\mathrm{Noise}}$** In order to estimate $\mathcal{E}_{\mathrm{Noise}}$, we proceed as follows: Recall from Section 3 that ridgeless regression is the limit of ridge regression for $\lambda \searrow 0$. We use a small regularization of $\lambda = 10^{-12}$ in order to improve numerical stability. Also for numerical stability, we use a singular value decomposition (SVD) $\boldsymbol{Z} = \boldsymbol{U} \operatorname{diag}(s_1, \ldots, s_k) \boldsymbol{V}^\top$ with $k := \min\{n, p\}$ as in Appendix B. The regularized approximation of $\boldsymbol{Z}^+$ is then

$$\boldsymbol{Z}^+ \approx (\boldsymbol{Z}^\top \boldsymbol{Z} + \lambda \boldsymbol{I}_p)^{-1} \boldsymbol{Z}^\top = \boldsymbol{V} \operatorname{diag}\left(\frac{s_1}{s_1^2 + \lambda}, \ldots, \frac{s_k}{s_k^2 + \lambda}\right) \boldsymbol{U}^\top \,. \tag{3}$$

---

[4] We use $n > p$ since this usually improves the "inverse condition number" of $\boldsymbol{Z}$.

We can then estimate

$$\operatorname{tr}((\boldsymbol{Z}^+)^\top \boldsymbol{\Sigma} \boldsymbol{Z}^+) = \operatorname{tr}(\boldsymbol{Z}^+ (\boldsymbol{Z}^+)^\top \boldsymbol{\Sigma}) \approx \operatorname{tr}\left( \boldsymbol{V} \operatorname{diag}\left( \frac{s_1^2}{(s_1^2 + \lambda)^2}, \ldots, \frac{s_k^2}{(s_k^2 + \lambda)^2} \right) \boldsymbol{V}^\top \boldsymbol{\Sigma} \right). \quad (4)$$

We can then obtain $m = 10^4$ (non-ReLU NNs, polynomial kernel, high-variance random Fourier features) or $m = 10^5$ (all other empirical results) sampled estimates for $\operatorname{tr}((\boldsymbol{Z}^+)^\top \boldsymbol{\Sigma} \boldsymbol{Z}^+)$ in order to obtain a Monte-Carlo estimate of $\mathbb{E}\operatorname{tr}((\boldsymbol{Z}^+)^\top \boldsymbol{\Sigma} \boldsymbol{Z}^+)$ by repeating the following procedure $m$ times:

(1) Sample a random parameter $\boldsymbol{\theta}$.
(2) Sample random matrices $\boldsymbol{X} \in \mathbb{R}^{n \times d}$ and $\tilde{\boldsymbol{X}} \in \mathbb{R}^{l \times d}$, $l = 10^4$, with i.i.d. $P_X$-distributed rows.
(3) Compute $\boldsymbol{Z} := \phi_{\boldsymbol{\theta}}(\boldsymbol{X})$ and $\tilde{\boldsymbol{Z}} := \phi_{\boldsymbol{\theta}}(\tilde{\boldsymbol{X}})$.
(4) Compute the estimate $\boldsymbol{\Sigma} := \frac{1}{l} \tilde{\boldsymbol{Z}}^\top \tilde{\boldsymbol{Z}}$.
(5) Compute a regularized estimate of $\operatorname{tr}((\boldsymbol{Z}^+)^\top \boldsymbol{\Sigma} \boldsymbol{Z}^+)$ using the SVD of $\boldsymbol{Z}$ and Eq. (4).

For performance reasons, we make the following modification of step (2) and (5): Since we perform the computation for all $n \in [256]$, we sample $\boldsymbol{X} \in \mathbb{R}^{256 \times d}$ and then, for all $n \in [256]$, perform the computation for $n$ using the first $n$ rows of $\boldsymbol{Z}$. Hence, the estimates for different $n$ are not independent. In Figure C.3, we perform an analogous optimization for $p$ by taking the first $p \in [256]$ of the $d_3 = 256$ output features in $\boldsymbol{Z}$.

**Curious ReLU results** The curves for the ReLU NNs and ReLU NTKs in the underparameterized regime $p \le n$ may seem odd. The locally almost constant "plateaus" are presumably an artefact of the non-independent estimates for different $n$ as explained in the last paragraph. As discussed in Section 5, networks with ReLU, ELU or SELU activations do not satisfy (FRK). It seems plausible that, since both "halves" of the ReLU function are linear, ReLU networks have a significantly higher chance than SELU or ELU networks to be initialized with "bad" parameters that are likely to generate "degenerate" feature matrices $\boldsymbol{Z}$ that do not have full rank at $n = p$ and only become full rank for some $n > p$. When inspecting the data underlying the plots, the estimate of $\mathcal{E}_{\text{Noise}}$ for ReLU networks in the underparameterized regime seems to be dominated by few outliers. It seems that the distribution of $\operatorname{tr}((\boldsymbol{Z}^+)^\top \boldsymbol{\Sigma} \boldsymbol{Z}^+)$ for ReLU networks is often so heavy-tailed that computing more Monte Carlo samples does not significantly reduce the estimation uncertainty.

**Whitening** For computing $\mathbb{E}((\boldsymbol{W}\boldsymbol{W}^\top)^{-1}) = \mathbb{E}((\boldsymbol{Z}\boldsymbol{\Sigma}^{-1}\boldsymbol{Z}^\top)^{-1})$ in the overparameterized case in Figure C.2, we regularize both matrix inversions on the right-hand side as above: For a symmetric matrix $\boldsymbol{A} \in \mathbb{R}^{m \times m}$, we use a symmetric eigendecomposition $\boldsymbol{A} = \boldsymbol{U} \operatorname{diag}(s_1, \ldots, s_m) \boldsymbol{U}^\top$ and approximate

$$\boldsymbol{A}^{-1} \approx \boldsymbol{U} \operatorname{diag}\left( \frac{s_1}{s_1^2 + \lambda}, \ldots, \frac{s_m}{s_m^2 + \lambda} \right) \boldsymbol{U}^\top .$$

**Optimization** For our optimized feature maps in Figure 1 with $p = 30$, we use a neural network feature map with $d_0 = d = p = 30, d_1 = d_2 = 256, d_3 = p = 30$ and $\tanh$ activation function. We use NTK parameterization and zero-initialized biases, leading to

$$\phi_{\boldsymbol{\theta}}(\boldsymbol{x}) = \sigma(\boldsymbol{b}^{(2)} + V_2^{-1/2} \boldsymbol{W}^{(2)} \sigma(\boldsymbol{b}^{(1)} + V_1^{-1/2} \boldsymbol{W}^{(1)} \sigma(\boldsymbol{b}^{(0)} + V_0^{-1/2} \boldsymbol{W}^{(0)} \boldsymbol{x})))$$

with independent initialization $W_{ij}^{(l)} \sim \mathcal{N}(0, 1)$, $b_i^{(l)} = 0$. As the input distribution, we use $P_X = \mathcal{N}(0, \boldsymbol{I}_d)$. For given $\boldsymbol{\theta}$, let $\boldsymbol{\Sigma}_{\boldsymbol{\theta}} := \mathbb{E}_{\boldsymbol{x}} \phi_{\boldsymbol{\theta}}(\boldsymbol{x}) \phi_{\boldsymbol{\theta}}(\boldsymbol{x})^\top$, i.e. we define the second moment matrix $\boldsymbol{\Sigma}$ as depending on $\boldsymbol{\theta}$. We then optimize the loss function

$$L(\boldsymbol{\theta}) := \mathbb{E}_{\boldsymbol{X}} \operatorname{tr}((\phi_{\boldsymbol{\theta}}(\boldsymbol{X})^+)^\top \boldsymbol{\Sigma}_{\boldsymbol{\theta}} \phi_{\boldsymbol{\theta}}(\boldsymbol{X})^+)$$

using AMSGrad (Reddi et al., 2018) with a learning rate that linearly decays from $10^{-3}$ to 0 over 1000 iterations. In order to approximate $L(\boldsymbol{\theta})$ in each iteration, we approximate $\boldsymbol{\Sigma}_{\boldsymbol{\theta}}$ using 1000 Monte Carlo points and we draw 1024 different realizations of $\boldsymbol{X}$ (this can be considered as using batch size 1024). We also use a regularized version as in Eq. (3), but we omit the SVD for reasons of differentiability.

# D A COUNTEREXAMPLE

**Example D.1.** Let $p \geq 2$. Consider the uniform distribution $P_Z$ on an orthonormal basis $\{e_1, \ldots, e_p\} \subseteq \mathbb{R}^p$. Then, $\mathbf{\Sigma} = \frac{1}{p} \sum_{i=1}^{p} e_i e_i^\top = \frac{1}{p} I_p$ and hence (COV) is satisfied. However, from Proposition 6 it is easy to see that for any $n \geq 2$, (FRK) is not satisfied. Indeed, if the vector $e_i$ occurs $m_i$ times among the samples $z_1, \ldots, z_n$, then $\mathbf{Z}^\top \mathbf{Z} = \mathrm{diag}(m_1, \ldots, m_p)$. Assuming $\mathrm{Var}(y|z) = \sigma^2 := 1$ for all $z$, we then obtain from Theorem 3 with the convention $\frac{1}{0} := 0$:

$$\mathcal{E}_{\text{Noise}} = \mathbb{E}_{\mathbf{Z}} \mathrm{tr}((\mathbf{Z}^+)^\top \mathbf{\Sigma} \mathbf{Z}^+) = \frac{1}{p} \mathbb{E}_{\mathbf{Z}} \mathrm{tr}(\mathbf{Z}^+ (\mathbf{Z}^+)^\top) = \frac{1}{p} \mathbb{E}_{\mathbf{Z}} \mathrm{tr}((\mathbf{Z}^\top \mathbf{Z})^+)$$

$$= \frac{1}{p} \sum_{i=1}^{p} \mathbb{E} \frac{1}{m_i} = \mathbb{E} \frac{1}{m_1} ,$$

where $m_1$ follows a binomial distribution with parameters $n$ and $\frac{1}{p}$. Using $\mathbb{E} \frac{1}{m_1} \leq \mathbb{E} 1 = 1$, it is easy to see that the lower bound is violated for some values of $n$. For example, Figure D.1 shows that for $p = 30$, the lower bound is violated in a large region, especially in the overparameterized regime.[5] The underlying reason is that the function $x \mapsto \frac{1}{x}$ is convex on $(0, \infty)$, but not on $[0, \infty)$. If $\mathbf{Z}$ has singular values $s_i$, the pseudo-inverse $\mathbf{Z}^+$ has singular values $\frac{1}{s_i}$. If $s_i = 0$ is possible, we cannot use a convexity-based argument anymore. ◀

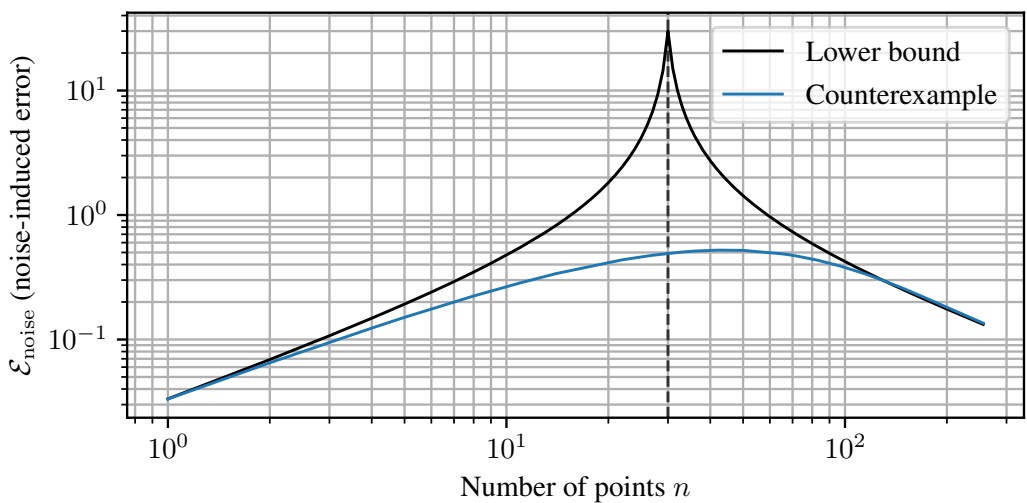

Figure D.1: The counterexample given in Example D.1 for $p = 30$ often has lower $\mathcal{E}_{\text{Noise}}$ than the lower bound from Theorem 3, but violates its assumption (FRK). For the counterexample, $\mathcal{E}_{\text{Noise}}$ was approximated as explained in Example D.1 using $10^6$ Monte Carlo samples for each $n$. We assume $\mathrm{Var}(y|z) = 1$ almost surely over $z$.

**Remark D.2** (Histogram regression). The distribution $P_Z$ in Example D.1 may seem contrived at first, but such a distribution can arise in histogram regression (cf e.g. Chapter 4 in Györfi et al., 2002). For example, suppose that $P_X$ is supported on a domain $\mathcal{D} \subseteq \mathbb{R}^d$ and this domain is partitioned into disjoint sets $\mathcal{A}_1, \ldots, \mathcal{A}_p$. Then, performing histogram regression on this partition is equivalent to performing ridgeless linear regression with the feature map $\phi : \mathbb{R}^d \to \mathbb{R}^p$ with $\phi(x) := e_i$ if $x \in \mathcal{A}_i$. If all partitions are equally likely, i.e. $P_X(\mathcal{A}_i) = 1/p$ for all $i \in \{1, \ldots, p\}$, then $P_Z$ is the uniform distribution on $\{e_1, \ldots, e_p\}$ as in Example D.1. ◀

# E FULL-RANK RESULTS FOR RANDOM WEIGHT NEURAL NETWORKS

In the literature on neural networks with random weights (Schmidt et al., 1992) and the later virtually identical Extreme Learning Machine (ELM) (Huang et al., 2006), properties similar to (FRK) have

---

[5]For $n = 1$, (FRK) holds and it is easy to see that the lower bound holds exactly in this case.

been investigated for random neural network feature maps. In the following, we review the relevant approaches known to us and explain why most of them are flawed.

First, Sartori & Antsaklis (1991) state a result where the assumptions are not clearly specified, but which could look as follows:

**Claim 1** (Sartori & Antsaklis (1991), informally discussed after Lemma 1). *For $n = p$, consider a single-layer random neural network with weights $\boldsymbol{W}^{(0)} \in \mathbb{R}^{(n-1) \times d}$ and signum activation $\sigma$ that appends a 1 to its output:*

$$\phi_{\boldsymbol{\theta}} : \mathbb{R}^n \to \mathbb{R}^p, \boldsymbol{x} \mapsto (\sigma(\boldsymbol{W}^{(0)} \boldsymbol{x}), 1) .$$

*If $\boldsymbol{x}_1, \ldots, \boldsymbol{x}_n \in \mathbb{R}^d$ are distinct, then for almost all $\boldsymbol{W}^{(0)}$, $\phi_{\boldsymbol{\theta}}(\boldsymbol{X})$ is invertible.*

This claim is evidently false for $n \geq 2$. For example, the following argument works for $n \geq 3$: For $\boldsymbol{\tau} \in \{-1, 0, 1\}^n$, let $\mathcal{U}_{\boldsymbol{\tau}} := \{\boldsymbol{w} \in \mathbb{R}^n \mid \sigma(\boldsymbol{w}^\top \boldsymbol{x}_j) = \tau_j \text{ for all } j\}$. Then, $\bigcup_{\boldsymbol{\tau} \in \{-1,0,1\}^n} \mathcal{U}_{\boldsymbol{\tau}} = \mathbb{R}^n$ and since $\{-1, 0, 1\}^n$ is finite, there exists $\boldsymbol{\tau}^*$ such that $\mathcal{U}_{\boldsymbol{\tau}^*}$ is not a Lebesgue null set. Hence the set $\mathcal{W} := \{\boldsymbol{W}^{(0)} \in \mathbb{R}^{(n-1) \times n} \mid \text{at least two rows of } \boldsymbol{W}^{(0)} \text{ are in } \mathcal{U}_{\boldsymbol{\tau}^*}\}$ is not a Lebesgue null set. But for all $\boldsymbol{W}^{(0)} \in \mathcal{W}$, the matrix $\phi_{\boldsymbol{\theta}}(\boldsymbol{X}) = (\sigma(\boldsymbol{X}(\boldsymbol{W}^{(0)})^\top) \mid \boldsymbol{1}_{n \times 1})$ has two columns equal to $\boldsymbol{\tau}^*$ and is therefore not invertible, contradicting Claim 1.

Second, Tamura & Tateishi (1997) state a result where the assumptions are not clearly formulated, but which could look as follows:

**Claim 2** (Tamura & Tateishi (1997)). *For $n = p$, consider a single-layer random neural network with weights $\boldsymbol{W}^{(0)} \in \mathbb{R}^{(n-1) \times d}$, biases $\boldsymbol{b}^{(0)} \in \mathbb{R}^{n-1}$ and sigmoid activation $\sigma$ that appends a 1 to its output:*

$$\phi_{\boldsymbol{\theta}} : \mathbb{R}^n \to \mathbb{R}^p, \boldsymbol{x} \mapsto (\sigma(\boldsymbol{W}^{(0)} \boldsymbol{x} + \boldsymbol{b}^{(0)}), 1) .$$

*If $\boldsymbol{x}_1, \ldots, \boldsymbol{x}_n \in \mathbb{R}^d$ are distinct, then for almost all $\boldsymbol{W}^{(0)}$ and all $a < b$, there exists $\boldsymbol{b}^{(0)} \in [a, b]^{n-1}$ such that $\phi_{\boldsymbol{\theta}}(\boldsymbol{X})$ is invertible.*

Tamura & Tateishi (1997) attempt to prove this claim via showing that the curve $c_i : [a, b] \to \mathbb{R}^n, b_i \mapsto \left(\sigma((\boldsymbol{w}_i^{(0)})^\top \boldsymbol{x}_j + b_i)\right)_{j \in [n]}$ is not contained in any $(n-1)$-dimensional subspace of $\mathbb{R}^n$, which would allow them to pick a suitable bias $b_i$ for each curve $c_i$ such that the $c_i(b_i)$ and $(1, \ldots, 1)^\top$ are linearly independent. Although this is part is formulated confusingly, it should work out. The major problem is that under the counterassumption that $c_i$ lies in an $(n-1)$-dimensional subspace, they construct an infinite (overdetermined) system of equations involving derivatives of the sigmoid function which they claim has no solution. However, in general, overdetermined systems can have solutions and they do not prove why this would not be the case in their situation. Indeed, the only properties of the sigmoid function they use is that it is $C^\infty$ and not a polynomial, and it is not hard to see that the exponential function has these properties as well and leads to a solvable overdetermined system since its derivatives are all identical.

This leads us to the next claim:

**Claim 3** (Huang (2003), Theorem 2.1). *Consider the setting of Claim 2 but with $\boldsymbol{W}^{(0)} \in \mathbb{R}^{n \times d}$ instead of appending a 1 to all feature vectors. If $\boldsymbol{x}_1, \ldots, \boldsymbol{x}_n \in \mathbb{R}^d$ are distinct and $\boldsymbol{\theta}$ has a Lebesgue density, then $\phi_{\boldsymbol{\theta}}(\boldsymbol{X})$ is invertible almost surely over $\boldsymbol{\theta}$.*

Huang (2003) uses the "proof" by Tamura & Tateishi (1997) to show that from any nontrivial intervals one can choose $\boldsymbol{W}^{(0)}, \boldsymbol{b}^{(0)}$ such that $\phi_{\boldsymbol{\theta}}(\boldsymbol{X})$ is invertible, setting all rows of $\boldsymbol{W}^{(0)}$ to be equal. He then concludes without further reasoning that $\phi_{\boldsymbol{\theta}}(\boldsymbol{X})$ must then be invertible almost surely over $\boldsymbol{\theta}$. This major unexplained step cannot be fixed by a continuity-based argument since all $b_i^{(0)}$ were chosen from the same interval and similarly, all rows of $\boldsymbol{W}^{(0)}$ were chosen from the same interval. Since the sigmoid function is analytic, it would however be possible to prove this step using the multivariate identity theorem for analytic functions, Theorem H.3, in a fashion similar to the proof of (d) $\Rightarrow$ (a) in Proposition 6. The approach pursued in Huang (2003) thus introduces an additional problem on top of the problem of Tamura & Tateishi (1997) although this additional problem is in principle fixable.

A similar strategy is reused in the next claim issued in a particularly popular paper:

**Claim 4** (Huang et al. (2006), Theorem 2.1). *Claim 3 holds for all $C^\infty$ activation functions $\sigma$.*

Not only does the "proof" in Huang et al. (2006) inherit the problems from the previous "proofs", but now the result is obviously false as well: For $\sigma \equiv 0$, the matrix $\phi_{\boldsymbol{\theta}}(\boldsymbol{X})$ can never be invertible, and for $\sigma = \mathrm{id}$, it is also easy to find counterexamples. Moreover, it is not sufficient to exclude (low-order) polynomials $\sigma$: For example, the well-known construction (e.g. Remark 3.4 (d) in Chapter V.3 in Amann & Escher, 2005)

$$\sigma(x) := \begin{cases} 0 & , x \leq 0 \\ e^{-1/x} & , x > 0 \end{cases}$$

yields a $C^\infty$ function $\sigma$ that is zero on $(-\infty, 0]$ but not a polynomial. For this function, it is not hard to see that $\phi_{\boldsymbol{\theta}}(\boldsymbol{X})$ would be singular with nonzero probability since there is a chance that $\boldsymbol{W}^{(0)}\boldsymbol{x} + \boldsymbol{b}^{(0)}$ contains only negative components.

Despite these evident problems and the paper's popularity, Claim 4 is restated years later as Theorem 1 in a survey paper by the same author (Huang et al., 2015). In his Theorem 1, Guo (2018) attempts to disprove Claim 4. However, this "disproof" is also invalid: For certain saturating activation functions like $\tanh$, Guo (2018) lets $\boldsymbol{\theta} \to \infty$ in a certain fashion, which causes $\phi_{\boldsymbol{\theta}}(\boldsymbol{X})$ to converge to a singular matrix. The problem with this approach is that just because the limiting matrix is singular, the matrices for finite $\boldsymbol{\theta}$ do not need to be singular. However, this limiting behavior might at least explain the empirical results of Henriquez & Ruz (2017), which find in a certain setting with sigmoid activation that numerically, $\phi_{\boldsymbol{\theta}}(\boldsymbol{X})$ is usually not a full-rank matrix. In contrast, Widrow et al. (2013) numerically reach the opposite conclusion. This is not a contradiction, considering that very small eigenvalues of $\phi_{\boldsymbol{\theta}}(\boldsymbol{X})$ may be numerically rounded to zero, and the probability of having such small eigenvalues depends on the chosen distributions of $\boldsymbol{x}$ and $\boldsymbol{\theta}$.

The only previously published correct result known to us is the following one:

**Lemma E.5** (Lemma 4.4 in Nguyen & Hein (2017)). *Let $\sigma : \mathbb{R} \to \mathbb{R}$ be analytic, strictly increasing and let one of the following two conditions be satsified:*

*(a) $\sigma$ is bounded, or*
*(b) there exist positive constants $\rho_1, \rho_2, \rho_3, \rho_4$ such that $|\sigma(t)| \leq \rho_1 e^{\rho_2 t}$ for $t > 0$ and $|\sigma(t)| \leq \rho_3 t + \rho_4$ for $t \geq 0$.*

*Let $\phi_{\boldsymbol{\theta}}$ be a random NN feature map with biases as in Theorem 10 (b) and let $\boldsymbol{x}_1, \ldots, \boldsymbol{x}_n \in \mathbb{R}^d$ be distinct. If $p \geq n - 1$, then the $n \times (p+1)$ feature matrix*

$$\begin{pmatrix} \phi_{\boldsymbol{\theta}}(\boldsymbol{x}_1)^\top & 1 \\ \vdots & \vdots \\ \phi_{\boldsymbol{\theta}}(\boldsymbol{x}_n)^\top & 1 \end{pmatrix}$$

*has rank $n$ for (Lebesgue-) almost all $\boldsymbol{\theta}$.*

In Theorem H.9, we generalize Lemma E.5 (without the appended ones in the feature matrix) to a substantially larger class of analytic activation functions.[6] As argued above, it is not possible to replace the assumption that $\sigma$ is analytic with $\sigma \in C^\infty$ and as shown in Remark H.17, our exclusion of certain polynomials for $\sigma$ is in general necessary.

## F PROOFS FOR SECTION 4

In this section, we prove our main theorem and corollary from Section 4. Recall from Definition 2 that if $\mathbb{E}\|\boldsymbol{z}\|_2^2 < \infty$, we can define the second moment matrix

$$\boldsymbol{\Sigma} := \mathbb{E}_{\boldsymbol{z} \sim P_Z}\left[\boldsymbol{z}\boldsymbol{z}^\top\right] \in \mathbb{R}^{p \times p},$$

and if $\boldsymbol{\Sigma}$ is invertible, we can also define the "whitened" data matrix $\boldsymbol{W} := \boldsymbol{Z}\boldsymbol{\Sigma}^{-1/2}$, whose rows $\boldsymbol{w}_i = \boldsymbol{\Sigma}^{-1/2}$ satisfy $\mathbb{E}\boldsymbol{w}_i\boldsymbol{w}_i^\top = \boldsymbol{I}_p$.

---

[6]It is in principle possible to extend our results to the case with appended ones in the feature matrix by choosing the activation function for one of the output neurons to be $\sigma \equiv 1$. As discussed in Remark H.16, our arguments have no problem handling different activation functions. In this case, we would only need to adapt the corresponding Taylor series coefficients in Lemma H.8.

**Theorem 3** (Main result). *Let $n, p \geq 1$. Assume that $P$ and $\phi$ satisfy:*

*(INT)* $\mathbb{E}y^2 < \infty$ *and hence* $R_P(f_P^*) < \infty$,
*(NOI)* $\mathrm{Var}(y|\boldsymbol{z}) \geq \sigma^2$ *almost surely over* $\boldsymbol{z}$,
*(MOM)* $\mathbb{E}\|\boldsymbol{z}\|_2^2 < \infty$, *i.e.* $\boldsymbol{\Sigma}$ *exists and is finite,*
*(COV)* $\boldsymbol{\Sigma}$ *is invertible,*
*(FRK)* $\boldsymbol{Z} \in \mathbb{R}^{n \times p}$ *almost surely has full rank, i.e.* $\mathrm{rank}\,\boldsymbol{Z} = \min\{n, p\}$.

*Then, for the ridgeless linear regression estimator $f_{\boldsymbol{Z},\boldsymbol{y}}(\boldsymbol{z}) = \boldsymbol{z}^\top \boldsymbol{Z}^+ \boldsymbol{y}$, the following holds:*

$$\text{If } p \geq n, \qquad \mathcal{E}_{\text{Noise}} \overset{(I)}{\geq} \sigma^2 \mathbb{E}_{\boldsymbol{Z}} \mathrm{tr}((\boldsymbol{Z}^+)^\top \boldsymbol{\Sigma} \boldsymbol{Z}^+) \overset{(II)}{\geq} \sigma^2 \mathbb{E}_{\boldsymbol{Z}} \mathrm{tr}((\boldsymbol{W}\boldsymbol{W}^\top)^{-1}) \overset{(IV)}{\geq} \sigma^2 \frac{n}{p+1-n} \ .$$

$$\text{If } p \leq n, \qquad \mathcal{E}_{\text{Noise}} \overset{(I)}{\geq} \sigma^2 \mathbb{E}_{\boldsymbol{Z}} \mathrm{tr}((\boldsymbol{Z}^+)^\top \boldsymbol{\Sigma} \boldsymbol{Z}^+) \overset{(III)}{=} \sigma^2 \mathbb{E}_{\boldsymbol{Z}} \mathrm{tr}((\boldsymbol{W}^\top \boldsymbol{W})^{-1}) \overset{(V)}{\geq} \sigma^2 \frac{p}{n+1-p} \ .$$

*Here, the matrix inverses exist almost surely in the considered cases. Moreover, we have:*

- *If (NOI) holds with equality, then* (I) *holds with equality.*
- *If $n = p$ or $\boldsymbol{\Sigma} = \lambda \boldsymbol{I}_p$ for some $\lambda > 0$, then* (II) *holds with equality.*

*Proof.* By (FRK), we only consider the case where $\boldsymbol{Z}$ has full rank. For further details on some matrix computations, we refer to Appendix B.

**Step 1: Preparation.** Since the pairs $(\boldsymbol{z}_1, y_1), \ldots, (\boldsymbol{z}_n, y_n)$ are independent, we have the conditional covariance matrix

$$\mathrm{Cov}(\boldsymbol{y}|\boldsymbol{Z}) = \begin{pmatrix} \mathrm{Var}(y_1|\boldsymbol{z}_1) & & \\ & \ddots & \\ & & \mathrm{Var}(y_n|\boldsymbol{z}_n) \end{pmatrix} \overset{(I)}{\succeq} \sigma^2 \boldsymbol{I}_n$$

with equality in (I) if (NOI) holds with equality. Therefore,

$$\begin{aligned}
\mathcal{E}_{\text{Noise}} &\overset{\text{def}}{=} \mathbb{E}_{\boldsymbol{Z},\boldsymbol{y},\boldsymbol{\theta},\boldsymbol{z}} \left( f_{\boldsymbol{Z},\boldsymbol{y},\boldsymbol{\theta}}(\boldsymbol{z}) - \mathbb{E}_{\boldsymbol{y}|\boldsymbol{Z}} f_{\boldsymbol{Z},\boldsymbol{y},\boldsymbol{\theta}}(\boldsymbol{z}) \right)^2 \\
&= \mathbb{E}_{\boldsymbol{Z},\boldsymbol{y},\boldsymbol{z}} \left( \boldsymbol{z}^\top \boldsymbol{Z}^+ \boldsymbol{y} - \mathbb{E}_{\boldsymbol{y}|\boldsymbol{Z}} \boldsymbol{z}^\top \boldsymbol{Z}^+ \boldsymbol{y} \right)^2 \\
&= \mathbb{E}_{\boldsymbol{Z},\boldsymbol{z}} \mathbb{E}_{\boldsymbol{y}|\boldsymbol{Z}} \boldsymbol{z}^\top \boldsymbol{Z}^+ (\boldsymbol{y} - \mathbb{E}_{\boldsymbol{y}|\boldsymbol{Z}}\boldsymbol{y})(\boldsymbol{y} - \mathbb{E}_{\boldsymbol{y}|\boldsymbol{Z}}\boldsymbol{y})^\top (\boldsymbol{Z}^+)^\top \boldsymbol{z} \\
&= \mathbb{E}_{\boldsymbol{Z},\boldsymbol{z}} \boldsymbol{z}^\top \boldsymbol{Z}^+ \mathrm{Cov}(\boldsymbol{y}|\boldsymbol{Z})(\boldsymbol{Z}^+)^\top \boldsymbol{z} \\
&\overset{(I)}{\geq} \sigma^2 \mathbb{E}_{\boldsymbol{Z},\boldsymbol{z}} \boldsymbol{z}^\top \boldsymbol{Z}^+ (\boldsymbol{Z}^+)^\top \boldsymbol{z} \\
&= \sigma^2 \mathbb{E}_{\boldsymbol{Z},\boldsymbol{z}} \mathrm{tr}((\boldsymbol{Z}^+)^\top \boldsymbol{z}\boldsymbol{z}^\top \boldsymbol{Z}^+) \\
&= \sigma^2 \mathbb{E}_{\boldsymbol{Z}} \mathrm{tr}((\boldsymbol{Z}^+)^\top \boldsymbol{\Sigma}(\boldsymbol{Z}^+)^\top) \ .
\end{aligned}$$

**Step 2.1: Reformulation, underparameterized case.** In the case $p \leq n$, we have $\boldsymbol{Z}^+ = (\boldsymbol{Z}^\top \boldsymbol{Z})^{-1}\boldsymbol{Z}^\top$ thanks to (FRK) and thus

$$\mathrm{tr}((\boldsymbol{Z}^+)^\top \boldsymbol{\Sigma} \boldsymbol{Z}^+) = \mathrm{tr}(\boldsymbol{\Sigma}^{1/2}\boldsymbol{Z}^+(\boldsymbol{Z}^+)^\top \boldsymbol{\Sigma}^{1/2}) = \mathrm{tr}(\boldsymbol{\Sigma}^{1/2}(\boldsymbol{Z}^\top \boldsymbol{Z})^{-1}\boldsymbol{\Sigma}^{1/2}) = \mathrm{tr}((\boldsymbol{W}^\top \boldsymbol{W})^{-1}) \ .$$

**Step 2.2: Reformulation, overparameterized case.** In the case $p \geq n$, we have $\boldsymbol{Z}^+ = \boldsymbol{Z}^\top(\boldsymbol{Z}\boldsymbol{Z}^\top)^{-1}$ thanks to (FRK). For $\boldsymbol{\Sigma} = \lambda \boldsymbol{I}_p$, we can show $\mathrm{tr}((\boldsymbol{Z}^+)^\top \boldsymbol{\Sigma} \boldsymbol{Z}^+) = \mathrm{tr}((\boldsymbol{W}\boldsymbol{W}^\top)^{-1})$ similar to Step 2.1. For general $\boldsymbol{\Sigma}$, we can obtain a lower bound by "removing a projection": First, let

$$\begin{aligned}
\boldsymbol{S} &\coloneqq (\boldsymbol{Z}^+)^\top \boldsymbol{\Sigma} \boldsymbol{Z}^+ = (\boldsymbol{Z}\boldsymbol{Z}^\top)^{-1}\boldsymbol{Z}\boldsymbol{\Sigma}\boldsymbol{Z}^\top(\boldsymbol{Z}\boldsymbol{Z}^\top)^{-1} \ , \\
\boldsymbol{A} &\coloneqq \boldsymbol{\Sigma}^{1/2}\boldsymbol{Z}^\top \ .
\end{aligned}$$

Now, since $\boldsymbol{Z}$ and $\boldsymbol{\Sigma}$ have full rank, we can compute

$$\boldsymbol{S}^{-1} = \boldsymbol{Z}\boldsymbol{Z}^\top(\boldsymbol{Z}\boldsymbol{\Sigma}\boldsymbol{Z}^\top)^{-1}\boldsymbol{Z}\boldsymbol{Z}^\top = \boldsymbol{W}\boldsymbol{A}(\boldsymbol{A}^\top \boldsymbol{A})^{-1}\boldsymbol{A}^\top \boldsymbol{W}^\top \ .$$

Since $\boldsymbol{A}(\boldsymbol{A}^\top \boldsymbol{A})^{-1}\boldsymbol{A}^\top$ is the orthogonal projection onto the column space of $\boldsymbol{A}$, we have $\boldsymbol{S}^{-1} \preceq \boldsymbol{W}\boldsymbol{W}^\top$ and hence $\lambda_i(\boldsymbol{S}^{-1}) \leq \lambda_i(\boldsymbol{W}\boldsymbol{W}^\top)$ by the Courant-Fischer-Weyl theorem. This yields

$$\mathrm{tr}((\boldsymbol{Z}^+)^\top \boldsymbol{\Sigma} \boldsymbol{Z}^+) = \mathrm{tr}(\boldsymbol{S}) = \sum_{i=1}^n \lambda_{n+1-i}(\boldsymbol{S}) = \sum_{i=1}^n \frac{1}{\lambda_i(\boldsymbol{S}^{-1})}$$

$$\geq \sum_{i=1}^{n} \frac{1}{\lambda_i(\boldsymbol{W}\boldsymbol{W}^\top)} = \sum_{i=1}^{n} \lambda_{n+1-i}((\boldsymbol{W}\boldsymbol{W}^\top)^{-1}) = \mathrm{tr}((\boldsymbol{W}\boldsymbol{W}^\top)^{-1}) \,.$$

For $n = p$, $\boldsymbol{A}(\boldsymbol{A}^\top\boldsymbol{A})^{-1}\boldsymbol{A}^\top$ projects onto a $p$-dimensional space and is therefore the identity matrix, yielding equality.

**Step 3.0: Random matrix bound, $p = 1$ or $n = 1$.** If $n \geq p = 1$, $\boldsymbol{w}_i = w_i$ is a scalar and we have

$$\mathbb{E}\,\mathrm{tr}((\boldsymbol{W}^\top\boldsymbol{W})^{-1}) = \mathbb{E}\frac{1}{\boldsymbol{W}^\top\boldsymbol{W}} = \mathbb{E}\frac{1}{\sum_{i=1}^{n} w_i^2} \geq \frac{1}{\mathbb{E}\sum_{i=1}^{n} w_i^2} = \frac{1}{\sum_{i=1}^{n} \mathrm{tr}(\mathbb{E}w_i w_i^\top)} = \frac{1}{n}$$
$$= \frac{p}{n+1-p}$$

by Jensen's inequality. Similarly, for $p \geq n = 1$, we obtain

$$\mathbb{E}\,\mathrm{tr}((\boldsymbol{W}\boldsymbol{W}^\top)^{-1}) = \mathbb{E}\frac{1}{\boldsymbol{w}_1^\top\boldsymbol{w}_1} \geq \frac{1}{\mathbb{E}\boldsymbol{w}_1^\top\boldsymbol{w}_1} = \frac{1}{\mathrm{tr}(\mathbb{E}\boldsymbol{w}_1\boldsymbol{w}_1^\top)} = \frac{1}{p} = \frac{n}{p+1-n} \,.$$

**Step 3.1: Random matrix bound, overparameterized case.** We first consider the overparameterized case $p \geq n \geq 2$ and block-decompose

$$\boldsymbol{W} =: \begin{pmatrix} \boldsymbol{w}_1^\top \\ \boldsymbol{W}_2 \end{pmatrix} \in \mathbb{R}^{(1+(n-1))\times p} \quad \Rightarrow \quad \boldsymbol{W}\boldsymbol{W}^\top = \begin{pmatrix} \boldsymbol{w}_1^\top\boldsymbol{w}_1 & \boldsymbol{w}_1^\top\boldsymbol{W}_2^\top \\ \boldsymbol{W}_2\boldsymbol{w}_1 & \boldsymbol{W}_2\boldsymbol{W}_2^\top \end{pmatrix} \,.$$

Since $\boldsymbol{Z}$ has full rank, $\boldsymbol{W}$ has full rank. Because of $n \leq p$, it follows that $\boldsymbol{W}\boldsymbol{W}^\top \succ 0$. Therefore,

$$\left((\boldsymbol{W}\boldsymbol{W}^\top)^{-1}\right)_{11} = \left(\boldsymbol{w}_1^\top\boldsymbol{w}_1 - \boldsymbol{w}_1^\top\boldsymbol{W}_2^\top(\boldsymbol{W}_2\boldsymbol{W}_2^\top)^{-1}\boldsymbol{W}_2\boldsymbol{w}_1\right)^{-1} = \left(\boldsymbol{w}_1^\top(\boldsymbol{I}_p - \boldsymbol{P}_2)\boldsymbol{w}_1\right)^{-1} \,,$$

where

$$\boldsymbol{P}_2 := \boldsymbol{W}_2^\top \left(\boldsymbol{W}_2\boldsymbol{W}_2^\top\right)^{-1}\boldsymbol{W}_2 \in \mathbb{R}^{p\times p}$$

is the orthogonal projection onto the column space of $\boldsymbol{W}_2^\top$. Thus, $\boldsymbol{P}_2$ has the eigenvalues 1 with multiplicity $n - 1$ and 0 with multiplicity $p - (n - 1)$, which yields $\mathrm{tr}(\boldsymbol{P}_2) = n - 1$. Since the $\boldsymbol{z}_i$ are stochastically independent, $\boldsymbol{w}_1$ and $\boldsymbol{W}_2$ are also stochastically independent and we obtain

$$\mathbb{E}\boldsymbol{w}_1^\top(\boldsymbol{I}_p - \boldsymbol{P}_2)\boldsymbol{w}_1 = \mathbb{E}\,\mathrm{tr}((\boldsymbol{I}_p - \boldsymbol{P}_2)(\mathbb{E}_{\boldsymbol{w}_1}\boldsymbol{w}_1\boldsymbol{w}_1^\top)) = \mathbb{E}\,\mathrm{tr}(\boldsymbol{I}_p - \boldsymbol{P}_2) = p + 1 - n \,.$$

Using Jensen's inequality with the convex function $(0, \infty) \to (0, \infty), x \mapsto 1/x$, we thus find that

$$\mathbb{E}\left((\boldsymbol{W}\boldsymbol{W}^\top)^{-1}\right)_{11} = \mathbb{E}\left(\boldsymbol{w}_1^\top(\boldsymbol{I}_p - \boldsymbol{P}_2)\boldsymbol{w}_1\right)^{-1} \geq \frac{1}{\mathbb{E}\boldsymbol{w}_1^\top(\boldsymbol{I}_p - \boldsymbol{P}_2)\boldsymbol{w}_1} = \frac{1}{p+1-n} \,.$$

Since $\mathrm{tr}((\boldsymbol{W}\boldsymbol{W}^\top)^{-1}) = \sum_{i=1}^{n}((\boldsymbol{W}\boldsymbol{W}^\top)^{-1})_{ii}$ and all diagonal entries can be treated in the same fashion (e.g. via permutation of the $\boldsymbol{w}_i$), it follows that

$$\mathbb{E}\,\mathrm{tr}((\boldsymbol{W}\boldsymbol{W}^\top)^{-1}) \geq \frac{n}{p+1-n} \,.$$

**Step 3.2: Random matrix bound, underparameterized case.** In the case $n \geq p \geq 1$, we follow the proof of Corollary 1 in Mourtada (2019), which can be applied in our setting as follows: Introduce a new random variable $\boldsymbol{w}_{n+1}$ such that $\boldsymbol{w}_1, \ldots, \boldsymbol{w}_{n+1}$ are i.i.d. Then,

$$\mathbb{E}\,\mathrm{tr}((\boldsymbol{W}^\top\boldsymbol{W})^{-1}) = \mathbb{E}_{\boldsymbol{W}}\,\mathrm{tr}((\boldsymbol{W}^\top\boldsymbol{W})^{-1}\mathbb{E}_{\boldsymbol{w}_{n+1}}\boldsymbol{w}_{n+1}\boldsymbol{w}_{n+1}^\top) = \mathbb{E}\boldsymbol{w}_{n+1}^\top(\boldsymbol{W}^\top\boldsymbol{W})^{-1}\boldsymbol{w}_{n+1} \,.$$

Now, Lemma 1 in Mourtada (2019) uses the Sherman-Morrison formula to show that

$$\boldsymbol{w}_{n+1}^\top(\boldsymbol{W}^\top\boldsymbol{W})^{-1}\boldsymbol{w}_{n+1} = \frac{\hat{\ell}_{n+1}}{1 - \hat{\ell}_{n+1}}$$

with the so-called leverage score

$$\hat{\ell}_{n+1} := \boldsymbol{w}_{n+1}^\top(\boldsymbol{W}^\top\boldsymbol{W} + \boldsymbol{w}_{n+1}\boldsymbol{w}_{n+1}^\top)^{-1}\boldsymbol{w}_{n+1} \in [0, 1) \,.$$

Since $\boldsymbol{W}^\top \boldsymbol{W} + \boldsymbol{w}_{n+1}\boldsymbol{w}_{n+1}^\top = \sum_{i=1}^{n+1} \boldsymbol{w}_i \boldsymbol{w}_i^\top$ and since $\boldsymbol{w}_1, \ldots, \boldsymbol{w}_{n+1}$ are i.i.d., we obtain

$$\mathbb{E}\hat{\ell}_{n+1} = \mathbb{E}\operatorname{tr}\left(\left(\sum_{i=1}^{n+1} \boldsymbol{w}_i \boldsymbol{w}_i^\top\right)^{-1} \boldsymbol{w}_{n+1}\boldsymbol{w}_{n+1}^\top\right)$$

$$= \frac{1}{n+1}\mathbb{E}\operatorname{tr}\left(\left(\sum_{i=1}^{n+1} \boldsymbol{w}_i \boldsymbol{w}_i^\top\right)^{-1}\sum_{i=1}^{n+1} \boldsymbol{w}_i \boldsymbol{w}_i^\top\right)$$

$$= \frac{1}{n+1}\mathbb{E}\operatorname{tr}(\boldsymbol{I}_p) = \frac{p}{n+1} .$$

Finally, the function $[0,1) \to (0,\infty), x \mapsto \frac{x}{1-x} = \frac{1}{1-x} - 1$ is convex, and Jensen's inequality yields

$$\mathbb{E}\operatorname{tr}((\boldsymbol{W}^\top\boldsymbol{W})^{-1}) = \mathbb{E}\frac{\hat{\ell}_{n+1}}{1-\hat{\ell}_{n+1}} \geq \frac{\mathbb{E}\hat{\ell}_{n+1}}{1-\mathbb{E}\hat{\ell}_{n+1}} = \frac{p}{n+1-p} . \qquad \square$$

Note that it is not possible to analyze Step 3.2 like Step 3.1 since the corresponding matrix blocks are not stochastically independent.

**Corollary 4** (Random features). *Let $\boldsymbol{\theta} \sim P_\Theta$ be a random variable such that $\phi_{\boldsymbol{\theta}} : \mathbb{R}^d \to \mathbb{R}^p$ is a random feature map. Consider the random features regression estimator $f_{\boldsymbol{X},\boldsymbol{y},\boldsymbol{\theta}}(\boldsymbol{x}) = \boldsymbol{z}_{\boldsymbol{\theta}}^\top \boldsymbol{Z}_{\boldsymbol{\theta}}^+ \boldsymbol{y}$ with $\boldsymbol{z}_{\boldsymbol{\theta}} := \phi_{\boldsymbol{\theta}}(\boldsymbol{x})$ and $\boldsymbol{Z}_{\boldsymbol{\theta}} := \phi_{\boldsymbol{\theta}}(\boldsymbol{X})$. If for $P_\Theta$-almost all $\widetilde{\boldsymbol{\theta}}$, the assumptions of Theorem 3 are satisfied for $\boldsymbol{z} = \boldsymbol{z}_{\widetilde{\boldsymbol{\theta}}}$ and $\boldsymbol{Z} = \boldsymbol{Z}_{\widetilde{\boldsymbol{\theta}}}$ (with the corresponding matrix $\boldsymbol{\Sigma} = \boldsymbol{\Sigma}_{\widetilde{\boldsymbol{\theta}}}$), then*

$$\mathcal{E}_{\mathrm{Noise}} \geq \begin{cases} \sigma^2 \frac{n}{p+1-n} & \text{if } p \geq n, \\ \sigma^2 \frac{p}{n+1-p} & \text{if } p \leq n. \end{cases}$$

*Proof.* First, let $p \leq n$. Since $\boldsymbol{\theta}$ is independent from $\boldsymbol{X}, \boldsymbol{x}, \boldsymbol{y}$,

$$\mathcal{E}_{\mathrm{Noise}} = \mathbb{E}_{\boldsymbol{X},\boldsymbol{y},\boldsymbol{\theta},\boldsymbol{x}}\left(\boldsymbol{z}_{\boldsymbol{\theta}}^\top \boldsymbol{Z}_{\boldsymbol{\theta}}^+ \boldsymbol{y} - \mathbb{E}_{\boldsymbol{y}|\boldsymbol{X}}\boldsymbol{z}_{\boldsymbol{\theta}}^\top \boldsymbol{Z}_{\boldsymbol{\theta}}^+ \boldsymbol{y}\right)^2$$

$$= \mathbb{E}_{\boldsymbol{\theta}}\left[\mathbb{E}_{\boldsymbol{X},\boldsymbol{y},\boldsymbol{x}}\left(\boldsymbol{z}_{\boldsymbol{\theta}}^\top \boldsymbol{Z}_{\boldsymbol{\theta}}^+ \boldsymbol{y} - \mathbb{E}_{\boldsymbol{y}|\boldsymbol{X}}\boldsymbol{z}_{\boldsymbol{\theta}}^\top \boldsymbol{Z}_{\boldsymbol{\theta}}^+ \boldsymbol{y}\right)^2\right] \overset{\text{Theorem 3}}{\geq} \mathbb{E}_{\boldsymbol{\theta}}\sigma^2 \frac{p}{n+1-p} = \sigma^2 \frac{p}{n+1-p} .$$

The case $p \geq n$ can be treated analogously. $\qquad \square$

## G  DISCUSSION OF THE MAIN THEOREM

**Remark G.1** (Dependence on $\boldsymbol{\Sigma}$). Hastie et al. (2019) discuss that $\boldsymbol{\Sigma}$ only influences the expected excess risk in the overparameterized regime. In the following, we will illustrate that Theorem 3 even implies that in the overparameterized case, $\boldsymbol{\Sigma} = \lambda \boldsymbol{I}_d$ yields the lowest $\mathcal{E}_{\mathrm{Noise}}$. This fact is also discussed in Muthukumar et al. (2020) in a slightly different setting. Note that since $P_X$ is unknown in general, $\boldsymbol{\Sigma}$ is also unknown in general.

Assume that (NOI) holds with equality such that we have $\mathcal{E}_{\mathrm{Noise}} = \sigma^2 \mathbb{E}_{\boldsymbol{Z}}\operatorname{tr}((\boldsymbol{Z}^+)^\top \boldsymbol{\Sigma}\boldsymbol{Z}^+)$ by Theorem 3. Suppose that we perform linear regression on the whitened data $\tilde{z} := \boldsymbol{\Sigma}^{-1/2}\boldsymbol{z}$. Then,

$$\tilde{\boldsymbol{Z}} = \boldsymbol{Z}\boldsymbol{\Sigma}^{-1/2} = \boldsymbol{W} ,$$
$$\tilde{\boldsymbol{\Sigma}} = \mathbb{E}_{\tilde{z}}\tilde{z}\tilde{z}^\top = \boldsymbol{\Sigma}^{-1/2}\left[\mathbb{E}_{\boldsymbol{z}}\boldsymbol{z}\boldsymbol{z}^\top\right]\boldsymbol{\Sigma}^{-1/2} = \boldsymbol{I}_p ,$$
$$\tilde{\boldsymbol{W}} = \tilde{\boldsymbol{Z}}\tilde{\boldsymbol{\Sigma}}^{-1/2} = \tilde{\boldsymbol{Z}} = \boldsymbol{W} .$$

In the underparameterized case $p \leq n$, we then obtain

$$\mathbb{E}_{\tilde{\boldsymbol{Z}}}\operatorname{tr}((\tilde{\boldsymbol{Z}}^+)^\top \tilde{\boldsymbol{\Sigma}}\tilde{\boldsymbol{Z}}^+) = \mathbb{E}_{\tilde{\boldsymbol{Z}}}\operatorname{tr}((\tilde{\boldsymbol{W}}^\top \tilde{\boldsymbol{W}})^{-1}) = \mathbb{E}_{\boldsymbol{Z}}\operatorname{tr}((\boldsymbol{W}^\top \boldsymbol{W})^{-1}) = \mathbb{E}_{\boldsymbol{Z}}\operatorname{tr}((\boldsymbol{Z}^+)^\top \boldsymbol{\Sigma}\boldsymbol{Z}^+) .$$

Therefore, whitening the data does not make a difference if $p \leq n$. In contrast, for $p > n$, we only know

$$\mathbb{E}_{\tilde{\boldsymbol{Z}}}\operatorname{tr}((\tilde{\boldsymbol{Z}}^+)^\top \tilde{\boldsymbol{\Sigma}}\tilde{\boldsymbol{Z}}^+) \overset{\text{(II)}}{=} \mathbb{E}_{\tilde{\boldsymbol{Z}}}\operatorname{tr}((\tilde{\boldsymbol{W}}\tilde{\boldsymbol{W}}^\top)^{-1}) = \mathbb{E}_{\boldsymbol{Z}}\operatorname{tr}((\boldsymbol{W}\boldsymbol{W}^\top)^{-1}) \overset{\text{(II)}}{\leq} \mathbb{E}_{\boldsymbol{Z}}\operatorname{tr}((\boldsymbol{Z}^+)^\top \boldsymbol{\Sigma}\boldsymbol{Z}^+)$$

since (II) holds with equality for whitened features. From Step 2.1 in the proof of Theorem 3, it is obvious that (II) in general does not hold with equality. Hence, in the overparameterized case $p > n$, whitening the features often reduces and never increases $\mathcal{E}_{\text{Noise}}$. Since $\mathcal{E}_{\text{Noise}}$ is just a lower bound for the expected excess risk, whitening does not necessarily reduce the expected excess risk.

This phenomenon also has a different kernel-based interpretation: Under the assumptions (MOM) and (COV), we can choose an ONB $\boldsymbol{u}_1, \ldots, \boldsymbol{u}_p$ of eigenvectors of $\boldsymbol{\Sigma}$ with corresponding eigenvalues $\lambda_1, \ldots, \lambda_p > 0$. If $\boldsymbol{z} = \phi(\boldsymbol{x})$ and $k(\boldsymbol{x}, \boldsymbol{x}') = \phi(\boldsymbol{x})^\top \phi(\boldsymbol{x}')$, we can write

$$k(\boldsymbol{x}, \boldsymbol{x}') = \phi(\boldsymbol{x})^\top \left( \sum_{i=1}^n \boldsymbol{u}_i \boldsymbol{u}_i^\top \right) \phi(\boldsymbol{x}') = \sum_{i=1}^p \lambda_i \psi_i(\boldsymbol{x}) \psi_i(\boldsymbol{x}') , \tag{5}$$

where the functions $\psi_i : \mathbb{R}^d \to \mathbb{R}, \boldsymbol{x} \mapsto \frac{1}{\sqrt{\lambda_i}} \boldsymbol{u}_i^\top \phi(\boldsymbol{x})$ form an orthonormal system in $L_2(P_X)$:

$$\mathbb{E}_{\boldsymbol{x}} \psi_i(\boldsymbol{x}) \psi_j(\boldsymbol{x}) = \frac{1}{\sqrt{\lambda_i \lambda_j}} \boldsymbol{u}_i^\top \left[ \mathbb{E}_{\boldsymbol{x}} \phi(\boldsymbol{x}) \phi(\boldsymbol{x})^\top \right] \boldsymbol{u}_j = \frac{1}{\sqrt{\lambda_i \lambda_j}} \boldsymbol{u}_i^\top \boldsymbol{\Sigma} \boldsymbol{u}_j = \frac{\lambda_j}{\sqrt{\lambda_i \lambda_j}} \boldsymbol{u}_i^\top \boldsymbol{u}_j = \delta_{ij}. \tag{6}$$

Therefore, Eq. (5) is a Mercer representation of $k$ and the eigenvalues $\lambda_i$ of $\boldsymbol{\Sigma}$ are also the eigenvalues of the integral operator $T_k : L_2(P_X) \to L_2(P_X)$ associated with $k$:

$$(T_k f)(\boldsymbol{x}) := \int k(\boldsymbol{x}, \boldsymbol{x}') f(\boldsymbol{x}') \, \mathrm{d}P_X(\boldsymbol{x}') = \sum_{i=1}^n \lambda_i \langle \psi_i, f \rangle_{L_2(P_X)} \psi_i(\boldsymbol{x}) .$$

We can define a kernel $\tilde{k}$ with flattened eigenspectrum via

$$\tilde{k}(\boldsymbol{x}, \boldsymbol{x}') := \sum_{i=1}^p \psi_i(\boldsymbol{x}) \psi_i(\boldsymbol{x}') .$$

Its feature map $\boldsymbol{x} \mapsto (\psi_i(\boldsymbol{x}))_{i \in [n]}$ by Eq. (6) satisfies $\boldsymbol{\Sigma} = \boldsymbol{I}_p$. By the above discussion, $\mathcal{E}_{\text{Noise}}(\tilde{k}) \leq \mathcal{E}_{\text{Noise}}(k)$. However, this needs to be taken with caution: Assume for simplicity that for independent $\boldsymbol{x}, \tilde{\boldsymbol{x}}$, the feature map $\phi$ yields $\phi(\boldsymbol{x}), \phi(\tilde{\boldsymbol{x}}) \sim \mathcal{N}(0, \frac{1}{p} \boldsymbol{I}_p)$. Then, $k(\boldsymbol{x}, \boldsymbol{x}) = 1$ but $\mathbb{E} k(\boldsymbol{x}, \tilde{\boldsymbol{x}}) = 0$ and $\operatorname{Var} k(\boldsymbol{x}, \tilde{\boldsymbol{x}}) = \frac{1}{p^2} \sum_{i=1}^p \mathbb{E}_{u,v \sim \mathcal{N}(0,1)} u^2 v^2 = \frac{1}{p}$. In this sense, for $p \to \infty$, $k$ converges to the Dirac kernel $k(\boldsymbol{x}, \tilde{\boldsymbol{x}}) = \delta_{\boldsymbol{x}, \tilde{\boldsymbol{x}}}$, which satisfies $\mathcal{E}_{\text{Noise}} = 0$ but provides bad interpolation performance if $f_P^* \not\equiv 0$. ◀

The next lemma and its proof are an adaptation of Lemma 4.14 in Bordenave & Chafaï (2012).

**Lemma G.2.** *For $i \in [n]$, let $\mathcal{W}_{-i} := \operatorname{Span}\{\boldsymbol{w}_j \mid j \in [n] \setminus \{i\}\}$. Then, under the assumptions of Theorem 3, in the overparameterized case $p \geq n$,*

$$\operatorname{tr}((\boldsymbol{W}\boldsymbol{W}^\top)^{-1}) = \sum_{i=1}^n \operatorname{dist}(\boldsymbol{w}_i, \mathcal{W}_{-i})^{-2} . \tag{7}$$

*Proof.* The case $n = 1$ is trivial, hence let $n \geq 2$. In Step 3.1 in the proof of Theorem 3, since $\boldsymbol{P}_2$ is the orthogonal projection onto $\mathcal{W}_{-1}$, we have

$$\boldsymbol{w}_1^\top (\boldsymbol{I}_p - \boldsymbol{P}_2) \boldsymbol{w}_1 = \|\boldsymbol{w}_1\|_2^2 - \|\boldsymbol{P}_2 \boldsymbol{w}_1\|_2^2 \stackrel{\text{Pythagoras}}{=} \operatorname{dist}(\boldsymbol{w}_1, \mathcal{W}_{-1})^2 ,$$

where $\operatorname{dist}(\boldsymbol{w}_1, \mathcal{W}_{-1})$ is the Euclidean distance between $\boldsymbol{w}_1$ and $\mathcal{W}_{-1}$. □

**Remark G.3** (Is $\mathcal{U}(\mathbb{S}^{p-1})$ optimal?)**.** From (I) in Theorem 3 it is clear that the best possible lower bound for $\mathcal{E}_{\text{Noise}}$ under the assumptions of Theorem 3 given $n, p \geq 1$ is

$$\mathcal{E}_{\text{Noise}} \geq \sigma^2 \inf_{\substack{\text{Distribution } P_Z \text{ on } \mathbb{R}^p \text{ satisfying (MOM), (COV), (FRK)}}} \mathbb{E}_{\boldsymbol{Z} \sim P_Z^n} \operatorname{tr}((\boldsymbol{Z}^+)^\top \boldsymbol{\Sigma} \boldsymbol{Z}^+) . \tag{8}$$

Here, we want to discuss the hypothesis that the infimum in Eq. (8) is achieved (for example) by $P_Z = \mathcal{U}(\mathbb{S}^{p-1})$. Figure 1 shows that we were not able to obtain a lower $\mathcal{E}_{\text{Noise}}$ by optimizing a neural network feature map to minimize $\mathcal{E}_{\text{Noise}}$. In the following, we want to discuss some theoretical evidence as to why this is plausible in the overparameterized case $p \geq n$. Lemma G.2 shows that Step 3.1 of the proof of Theorem 3 has a distance-based interpretation. In this interpretation, Step 3.1 then applies Jensen's inequality to the convex function $(0, \infty) \to (0, \infty), x \mapsto 1/x$ using

$$\mathbb{E} \operatorname{dist}(\boldsymbol{w}_i, \mathcal{W}_{-i})^2 = p + 1 - n . \tag{9}$$

We can use this perspective to gain insights on how distributions $P_Z$ with small $\mathcal{E}_{\text{Noise}}$ in the overparameterized case $p \geq n$ look like. First of all, Remark G.1 suggests that for minimizing $\mathcal{E}_{\text{Noise}}$ in the overparameterized case, $\boldsymbol{\Sigma}$ should be a multiple of $\boldsymbol{I}_p$, which is clearly satisfied for $\mathcal{U}(\mathbb{S}^{p-1})$ by Theorem 11. Since the lower bound obtained from (9) is independent of the distribution of $\boldsymbol{W}$, minimizing $\mathbb{E}\operatorname{tr}((\boldsymbol{W}\boldsymbol{W}^\top)^{-1})$ amounts to minimizing the error made by Jensen's inequality, which essentially amounts to reducing the variance of the random variables $\operatorname{dist}(\boldsymbol{w}_i, \mathcal{W}_{-i})$. We can decompose $\operatorname{dist}(\boldsymbol{w}_i, \mathcal{W}_{-i}) = \|\boldsymbol{w}_i\|_2 \cdot \operatorname{dist}(\boldsymbol{w}_i/\|\boldsymbol{w}_i\|_2, \mathcal{W}_{-i})$, where $\operatorname{dist}(\boldsymbol{w}_i/\|\boldsymbol{w}_i\|_2, \mathcal{W}_{-i})$ only depends on the angular components $\boldsymbol{w}_j/\|\boldsymbol{w}_j\|_2$ for $j \in [n]$. This suggests that for a "good" distribution $P_W$,

- the radial component $\|\boldsymbol{w}_i\|_2$ should have low variance, and
- the distribution of the angular component $\boldsymbol{w}_i/\|\boldsymbol{w}_i\|_2$ should not contain "clusters", since clusters would increase the probability of $\operatorname{dist}(\boldsymbol{w}_i, \mathcal{W}_{-i})$ being very small.

Clearly, both points are perfectly achieved for $P_Z = \mathcal{U}(\mathbb{S}^{p-1})$. ◀

# H    PROOFS FOR SECTION 5

In this section, we prove all theorems and propositions from Section 5 as well as some additional results.

## H.1    MISCELLANEOUS

First, we prove a statement about conditional variances.

**Lemma H.1.** *In the setting of Theorem 3, we have*

$$\operatorname{Var}(y|\boldsymbol{z}) = \mathbb{E}(\operatorname{Var}(y|\boldsymbol{x})|\boldsymbol{z}) + \operatorname{Var}(\mathbb{E}(y|\boldsymbol{x})|\boldsymbol{z}) .$$

*Hence, if $\operatorname{Var}(y|\boldsymbol{x}) \geq \sigma^2$ almost surely over $\boldsymbol{x}$, then $\operatorname{Var}(y|\boldsymbol{z}) \geq \sigma^2$ almost surely over $\boldsymbol{z}$. The converse holds, for example, if $\phi$ is injective.*

*Proof.* For properties of conditional expectations, we refer to the literature, e.g. Chapter 4.1 in Durrett (2019). Since $\boldsymbol{z} = \phi(\boldsymbol{x})$ is a function of $\boldsymbol{x}$, we have $\mathbb{E}[\cdot|\boldsymbol{z}] = \mathbb{E}[\mathbb{E}(\cdot|\boldsymbol{x})|\boldsymbol{z}]$. Thus,

$$
\begin{aligned}
\operatorname{Var}(y|\boldsymbol{z}) &= \mathbb{E}\left[(y - \mathbb{E}(y|\boldsymbol{z}))^2\big|\boldsymbol{z}\right] \\
&= \mathbb{E}\left[\left((y - \mathbb{E}(y|\boldsymbol{x})) + (\mathbb{E}(y|\boldsymbol{x}) - \mathbb{E}(\mathbb{E}(y|\boldsymbol{x})|\boldsymbol{z}))\right)^2\Big|\boldsymbol{z}\right] \\
&= \mathbb{E}\left[\mathbb{E}\left((y - \mathbb{E}(y|\boldsymbol{x}))^2|\boldsymbol{x}\right)\big|\boldsymbol{z}\right] \\
&\quad + \mathbb{E}\left[\mathbb{E}\left((y - \mathbb{E}(y|\boldsymbol{x}))(\mathbb{E}(y|\boldsymbol{x}) - \mathbb{E}(\mathbb{E}(y|\boldsymbol{x})|\boldsymbol{z}))|\boldsymbol{x}\right)|\boldsymbol{z}\right] \\
&\quad + \mathbb{E}\left[\mathbb{E}\left((\mathbb{E}(y|\boldsymbol{x}) - \mathbb{E}(\mathbb{E}(y|\boldsymbol{x})|\boldsymbol{z}))^2|\boldsymbol{x}\right)\big|\boldsymbol{z}\right] .
\end{aligned}
$$

For the first term, we have $\mathbb{E}\left((y - \mathbb{E}(y|\boldsymbol{x}))^2|\boldsymbol{x}\right) = \operatorname{Var}(y|\boldsymbol{x})$ by definition. The second term is zero: Since $\mathbb{E}(y|\boldsymbol{x}) - \mathbb{E}(\mathbb{E}(y|\boldsymbol{x})|\boldsymbol{z})$ is already a function of $\boldsymbol{x}$, we have

$$
\begin{aligned}
\mathbb{E}\left((y - \mathbb{E}(y|\boldsymbol{x}))(\mathbb{E}(y|\boldsymbol{x}) - \mathbb{E}(\mathbb{E}(y|\boldsymbol{x})|\boldsymbol{z}))|\boldsymbol{x}\right) &= \mathbb{E}\left((y - \mathbb{E}(y|\boldsymbol{x}))|\boldsymbol{x}\right)(\mathbb{E}(y|\boldsymbol{x}) - \mathbb{E}(\mathbb{E}(y|\boldsymbol{x})|\boldsymbol{z})) \\
&= (\mathbb{E}(y|\boldsymbol{x}) - \mathbb{E}(y|\boldsymbol{x}))(\mathbb{E}(y|\boldsymbol{x}) - \mathbb{E}(\mathbb{E}(y|\boldsymbol{x})|\boldsymbol{z})) \\
&= 0 .
\end{aligned}
$$

Finally, the third term is by definition equal to $\operatorname{Var}(\mathbb{E}(y|\boldsymbol{x})|\boldsymbol{z})$. Therefore,

$$\operatorname{Var}(y|\boldsymbol{z}) = \mathbb{E}(\operatorname{Var}(y|\boldsymbol{x})|\boldsymbol{z}) + \operatorname{Var}(\mathbb{E}(y|\boldsymbol{x})|\boldsymbol{z}) .$$

If $\phi$ is injective, then $\boldsymbol{x}$ is also a function of $\boldsymbol{z}$ and we obtain $\operatorname{Var}(y|\boldsymbol{z}) = \operatorname{Var}(y|\boldsymbol{x})$. □

Our main ingredient for analyzing analytic activation functions will be the univariate and multivariate identity theorems.

**Theorem H.2** (Identity theorem for univariate real analytic functions)**.** *Let $f : \mathbb{R} \to \mathbb{R}$ be analytic. If $f$ is not the zero function, the set $\mathcal{N}(f) := \{z \in \mathbb{R} \mid f(z) = 0\}$ has no accumulation point. In particular, $\mathcal{N}(f)$ is countable.*

*Proof.* See e.g. Corollary 1.2.7 in Krantz & Parks (2002) for the first statement. If $\mathcal{N}(f)$ is uncountable, there exists $k \in \mathbb{Z}$ such that $[k, k+1] \cap \mathcal{N}(f)$ is also uncountable, hence it contains a strictly increasing and bounded sequence of points, and the limit of this sequence is an accumulation point of $\mathcal{N}(f)$. $\qquad\square$

**Theorem H.3** (Multivariate version of the identity theorem). *Let $f : \mathbb{R}^d \to \mathbb{R}$ be analytic. If $f$ is not the zero function, then $\mathcal{N}(f) := \{\boldsymbol{x} \in \mathbb{R}^d \mid f(\boldsymbol{x}) = 0\}$ is a Lebesgue null set.*

*Proof.* Although less well-known than the univariate version, this multivariate version has been proven several times in the literature. For example, different proofs are given in Section 3.1.24 in Federer (1969), Lemma 1.2 in Nguyen (2015) and Proposition 0 in Mityagin (2015). More proof strategies have been hinted at in Lemma 5.22 in Kuchment (2015). Here, we provide an elementary proof following the proof strategy briefly mentioned at the beginning of Section 4.1 in Krantz & Parks (2002).

Let $\lambda^d$ be the Lebesgue measure on $\mathbb{R}^d$ and let $\lambda := \lambda^1$. We prove the statement by induction on $d \geq 1$. For $d = 1$, if $\lambda(\mathcal{N}(f)) > 0$, then $\mathcal{N}(f)$ is uncountable and hence $f \equiv 0$ by Theorem H.2.

Now, let the statement hold for $d - 1 \geq 1$ and assume $\lambda^d(\mathcal{N}(f)) > 0$. For $a \in \mathbb{R}$, define the functions $f_a : \mathbb{R}^{d-1} \to \mathbb{R}, f_a(\boldsymbol{x}) = f(a, \boldsymbol{x})$. Then,

$$0 < \lambda^d(\mathcal{N}(f)) = \int_{\mathbb{R}^p} \mathbb{1}_{\mathcal{N}(f)} \,\mathrm{d}\lambda^d = \int_{\mathbb{R}} \int_{\mathbb{R}^{d-1}} \mathbb{1}_{\mathcal{N}(f)}(a, \boldsymbol{x}) \,\mathrm{d}\lambda^{d-1}(\boldsymbol{x}) \,\mathrm{d}\lambda(a)$$
$$= \int_{\mathbb{R}} \lambda^{d-1}(\mathcal{N}(f_a)) \,\mathrm{d}a \ .$$

It follows that the set $U := \{x \in \mathbb{R} \mid \lambda^{d-1}(\mathcal{N}(f_x)) > 0\}$ satisfies $\lambda(U) > 0$. By induction, for all $x \in U$, we have $f_x \equiv 0$. Then, for all $\boldsymbol{x} \in \mathbb{R}^{d-1}$, we can conclude that the function $f_{\boldsymbol{x}} : \mathbb{R} \to \mathbb{R}, a \mapsto f(a, \boldsymbol{x})$ satisfies $\mathcal{N}(f_{\boldsymbol{x}}) \supseteq U$ and therefore $\lambda(\mathcal{N}(f_{\boldsymbol{x}})) \geq \lambda(U) > 0$. Using the case $d = 1$ again, it follows that $f_{\boldsymbol{x}} \equiv 0$ and therefore $f(a, \boldsymbol{x}) = 0$ for all $a \in \mathbb{R}$ and $\boldsymbol{x} \in \mathbb{R}^{d-1}$. $\quad\square$

The following lemma provides some intuition about null sets for readers less familiar with measure theory. Recall that a property $Q$ holds almost surely with respect to a measure $P$ on $\mathbb{R}^d$ if there exists a null set $N$, i.e. a measurable set with $P(N) = 0$, such that $Q(\boldsymbol{x})$ holds for all $\boldsymbol{x} \in \mathbb{R}^d \setminus N$.

**Lemma H.4.** *Let $\lambda^d$ be the Lebesgue measure on $\mathbb{R}^d$ and let $P$ be a measure on $\mathbb{R}^d$ with a Lebesgue density function (i.e. a probability density function) $p$. Then, a null set with respect to $\lambda^d$ is also a null set with respect to $P$. The converse holds if $p(\boldsymbol{x}) \neq 0$ for (almost) all $\boldsymbol{x}$.*

*Proof.* A well-known fact from measure and integration theory states that if a measure $\mu$ has a density with respect to a measure $\nu$, then $\nu$-null sets are also $\mu$-null sets. Setting $\mu = P$ and $\nu = \lambda^d$ yields the first fact. If $p(\boldsymbol{x}) \neq 0$ for (almost) all $\boldsymbol{x}$, then $\mu := \lambda^d$ has density $1/p$ with respect to $\nu := P$, and hence the converse follows. $\qquad\square$

**Proposition 6** (Characterization of (COV) and (FRK)). *Consider the setting of Theorem 3 and let $\mathrm{FRK}(n)$ be the statement that (FRK) holds for $n$. Then,*

*(i) Let $n \geq 1$. Then, $\mathrm{FRK}(n)$ iff $P(\boldsymbol{z} \in U) = 0$ for all linear subspaces $U \subseteq \mathbb{R}^p$ of dimension $\min\{n, p\} - 1$.*

*(ii) Let (MOM) hold. Then, (COV) holds iff $P(\boldsymbol{z} \in U) < 1$ for all linear subspaces $U \subseteq \mathbb{R}^p$ of dimension $p - 1$.*

*Assuming that (MOM) holds such that (COV) is well-defined, consider the following statements:*

*(a) $\mathrm{FRK}(p)$ holds.*
*(b) $\mathrm{FRK}(n)$ holds for all $n \geq 1$.*
*(c) (COV) holds.*
*(d) There exists a fixed deterministic matrix $\widetilde{\boldsymbol{X}} \in \mathbb{R}^{p \times d}$ such that $\det(\phi(\widetilde{\boldsymbol{X}})) \neq 0$.*

*We have (a) $\Leftrightarrow$ (b) $\Rightarrow$ (c) $\Rightarrow$ (d). Furthermore, if $\boldsymbol{x} \in \mathbb{R}^d$ has a Lebesgue density and $\phi$ is analytic, then (a) – (d) are equivalent.*

*Proof.* **Step 1: Prove (i) and (ii).**

(i) Denote the $n$ (stochastically independent) rows of $\boldsymbol{Z}$ by $\boldsymbol{z}_1, \ldots, \boldsymbol{z}_n$. First, assume $P(\boldsymbol{z} \in U) > 0$ for some subspace $U$ of dimension $\min\{n, p\} - 1$. Then,

$$P\Big( \operatorname{rank}(\boldsymbol{Z}) \leq \min\{n, p\} - 1 \Big) \geq P(\boldsymbol{z}_1, \ldots, \boldsymbol{z}_n \in U) = (P(\boldsymbol{z} \in U))^n > 0 \,.$$

For the converse, it suffices to consider the case $n \leq p$ since if $n > p$ and if an arbitrary $p \times p$ submatrix of $\boldsymbol{Z} \in \mathbb{R}^{n \times p}$ almost surely has full rank, then $\boldsymbol{Z}$ also almost surely has full rank. We prove the statement for $n \leq p$ by induction on $n$. For $n = 1$, the claim is trivial. Thus, let $n > 1$ and let $\boldsymbol{z}_1, \ldots, \boldsymbol{z}_n$ be the (stochastically independent) rows of $\boldsymbol{Z}$. Assume $P(\boldsymbol{z} \in U) = 0$ for all linear subspaces $U \subseteq \mathbb{R}^p$ of dimension $n - 1$. Then, we also have $P(\boldsymbol{z} \in U) = 0$ for all linear subspaces $U \subseteq \mathbb{R}^p$ of dimension $n - 2$ and by the induction hypothesis, we obtain that $\boldsymbol{z}_1, \ldots, \boldsymbol{z}_{n-1}$ are almost surely linearly independent. Hence, almost surely, $\boldsymbol{z}_1, \ldots, \boldsymbol{z}_n$ are linearly independent iff $\boldsymbol{z}_n \notin U_{n-1} := \operatorname{Span}\{\boldsymbol{z}_1, \ldots, \boldsymbol{z}_{n-1}\}$. Then, since $\boldsymbol{z}_n$ and $U_{n-1}$ are stochastically independent,

$$P(\boldsymbol{Z} \in \mathbb{R}^{n \times p} \text{ has full rank}) = P(\boldsymbol{z}_1, \ldots, \boldsymbol{z}_n \text{ are linearly independent})$$

$$= P(\boldsymbol{z}_n \notin U_{n-1}) = \iint \mathbb{1}_{U_{n-1}^c}(\boldsymbol{z}_n) \, \mathrm{d}P_Z(\boldsymbol{z}_n) \, \mathrm{d}P(U_{n-1})$$

$$= \int P_Z(\boldsymbol{z}_n \notin U_{n-1}) \, \mathrm{d}P(U_{n-1}) = \int 1 \, \mathrm{d}P(U_{n-1}) = 1 \,.$$

For the case $p \leq n$, (i) is also proven after Definition 1 in Mourtada (2019).

(ii) If (COV) does not hold, there exists a vector $0 \neq \boldsymbol{v} \in \mathbb{R}^p$ with $\boldsymbol{v}^\top \boldsymbol{\Sigma} \boldsymbol{v} = 0$, hence

$$0 = \boldsymbol{v}^\top \big( \mathbb{E} \boldsymbol{z} \boldsymbol{z}^\top \big) \boldsymbol{v} = \mathbb{E}(\boldsymbol{v}^\top \boldsymbol{z})^2 \,,$$

and therefore $\boldsymbol{z}$ is almost surely orthogonal to $\boldsymbol{v}$. If $U$ is the orthogonal complement of $\operatorname{Span}\{\boldsymbol{v}\}$ in $\mathbb{R}^p$, then $P(\boldsymbol{z} \in U) = 1$.

For the converse, if there exists a $(p-1)$-dimensional subspace $U$ with $P(\boldsymbol{z} \in U) = 1$, then we can again take a vector $0 \neq \boldsymbol{v} \in \mathbb{R}^p$ that is orthogonal to $U$, reverse the above computation and obtain $\boldsymbol{v}^\top \boldsymbol{\Sigma} \boldsymbol{v} = 0$, hence (COV) does not hold.

**Step 2: (a) $\Leftrightarrow$ (b).** The implication (b) $\Rightarrow$ (a) is trivial and the implication (a) $\Rightarrow$ (b) follows immediately from (i).

**Step 3: (b) $\Rightarrow$ (c).** This also follows from (i) and (ii).

**Step 4: (c) $\Rightarrow$ (d).** We will prove by induction on $n$ that for all $n \in \{0, \ldots, p\}$, there exist $\boldsymbol{x}_1, \ldots, \boldsymbol{x}_n \in \mathbb{R}^d$ such that $\phi(\boldsymbol{x}_1), \ldots, \phi(\boldsymbol{x}_n)$ are linearly independent. For $n = 0$, the statement is trivial. Now assume that the statement holds for $\boldsymbol{x}_1, \ldots, \boldsymbol{x}_{n-1}$, where $0 \leq n - 1 \leq p - 1$. Since (COV) holds, by (ii) the subspace $U := \operatorname{Span}\{\phi(\boldsymbol{x}_1), \ldots, \phi(\boldsymbol{x}_{n-1})\}$ satisfies $P(\phi(\boldsymbol{x}) \in U) < 1$, hence there is $\boldsymbol{x}_n$ such that $\boldsymbol{x}_n \notin U$ and for this choice, $\phi(\boldsymbol{x}_1), \ldots, \phi(\boldsymbol{x}_n)$ are linearly independent.

Finally, the statement for $n = p$ yields the existence of $\boldsymbol{X} \in \mathbb{R}^{p \times d}$ such that $\phi(\boldsymbol{X})$ has linearly independent rows, which implies $\det(\phi(\boldsymbol{X})) \neq 0$.

**Step 5: Analytic feature map.** Assume that $\phi$ is analytic, $\boldsymbol{x}$ has a Lebesgue density and (d) holds. Let $n = p$. For $\widetilde{\boldsymbol{X}} \in \mathbb{R}^{p \times d}$, consider the analytic function $f(\widetilde{\boldsymbol{X}}) := \det(\phi(\widetilde{\boldsymbol{X}}))$. (The determinant is analytic since it is a polynomial in the matrix entries.) Since (d) holds, Theorem H.3 shows that $f(\widetilde{\boldsymbol{X}}) \neq 0$ for (Lebesgue-) almost all $\widetilde{\boldsymbol{X}}$. Since $\boldsymbol{x}$ has a Lebesgue density and $\boldsymbol{X}$ has independent $\boldsymbol{x}$-distributed rows, $\boldsymbol{X}$ has a Lebesgue density. Therefore, $f(\boldsymbol{X}) \neq 0$ almost surely over $\boldsymbol{X}$, hence (a) holds. $\qquad \square$

**Proposition 8** (Polynomial kernel). *Let $m, d \geq 1$ and $c > 0$. For $\boldsymbol{x}, \tilde{\boldsymbol{x}} \in \mathbb{R}^d$, define the polynomial kernel $k(\boldsymbol{x}, \tilde{\boldsymbol{x}}) := (\boldsymbol{x}^\top \tilde{\boldsymbol{x}} + c)^m$. Then, there exists a feature map $\phi : \mathbb{R}^d \to \mathbb{R}^p$, $p := \binom{m+d}{m}$, such that:*

*(a) $k(\boldsymbol{x}, \tilde{\boldsymbol{x}}) = \phi(\boldsymbol{x})^\top \phi(\tilde{\boldsymbol{x}})$ for all $\boldsymbol{x}, \tilde{\boldsymbol{x}} \in \mathbb{R}^d$, and*
*(b) if $\boldsymbol{x} \in \mathbb{R}^d$ has a Lebesgue density and we use $\boldsymbol{z} = \phi(\boldsymbol{x})$, then (FRK) is satisfied for all $n$.*

*Proof.* Let $\mathcal{M} := \{(m_1, \ldots, m_{d+1}) \in \mathbb{N}_0^{d+1} \mid m_1 + \ldots + m_{d+1} = m\}$ and for $\boldsymbol{m} = (m_1, \ldots, m_{d+1}) \in \mathcal{M}$, let

$$C(\boldsymbol{m}) := \begin{pmatrix} & m & \\ m_1 & \ldots & m_{d+1} \end{pmatrix}$$

be the corresponding multinomial coefficient. Then, $|\mathcal{M}| = \binom{m+d}{d} = p$. Define the feature map $\phi : \mathbb{R}^d \to \mathbb{R}^p$ by

$$\phi(\boldsymbol{x}) := \left( \sqrt{C(\boldsymbol{m})} z_1^{m_1} \cdots z_d^{m_d} \cdot (\sqrt{c})^{m_{d+1}} \right)_{\boldsymbol{m} \in \mathcal{M}} .$$

(a) We have

$$\phi(\boldsymbol{x})^\top \phi(\tilde{\boldsymbol{x}}) = \sum_{\boldsymbol{m} \in \mathcal{M}} C(\boldsymbol{m}) (x_1 \tilde{x}_1)^{m_1} \cdots (x_d \tilde{x}_d)^{m_d} \cdot c^{m_{d+1}}$$
$$= (x_1 \tilde{x}_1 + \ldots + x_d \tilde{x}_d + c)^m = k(\boldsymbol{x}, \tilde{\boldsymbol{x}}) .$$

(b) Assume that $\boldsymbol{x}$ has a Lebesgue density. Let $U$ be an arbitrary $(p-1)$-dimensional linear subspace of $\mathbb{R}^p$. Then, there exists $0 \neq \boldsymbol{v} \in \mathbb{R}^p$ such that $U = (\mathrm{Span}\{\boldsymbol{v}\})^\perp$. Since the monomials $x_1^{m_1} \cdots x_d^{m_d}$ for $\boldsymbol{m} \in \mathcal{M}$ are all distinct, the polynomial

$$\boldsymbol{v}^\top \phi(\boldsymbol{x}) = \sum_{\boldsymbol{m} \in \mathcal{M}} \left( v_{\boldsymbol{m}} \sqrt{C(\boldsymbol{m}) c^{m_{d+1}}} x_1^{m_1} \cdots x_d^{m_d} \right)$$

is not the zero polynomial. By the identity theorem (Theorem H.3), since $\boldsymbol{x}$ has a Lebesgue density and since polynomials are analytic,

$$P(\phi(\boldsymbol{x}) \in U) = P(\boldsymbol{v}^\top \phi(\boldsymbol{x}) = 0) = 0 .$$

Hence, Proposition 6 shows that (FRK) is satisfied for $n = p$ and hence for all $n$. $\qquad \square$

We want to remark at this point that the proof strategy of Proposition 8, where the identity theorem is applied to the functions $\boldsymbol{v}^\top \phi(\boldsymbol{x})$ for all $0 \neq \boldsymbol{v} \in \mathbb{R}^p$, does not work for random feature maps: The statements

- For all $0 \neq \boldsymbol{v} \in \mathbb{R}^p$ for almost all $\boldsymbol{x}$ for almost all $\boldsymbol{\theta}$, $\boldsymbol{v}^\top \phi_{\boldsymbol{\theta}}(\boldsymbol{x}) \neq 0$
- For almost all $\boldsymbol{\theta}$ for all $0 \neq \boldsymbol{v} \in \mathbb{R}^p$ for almost all $\boldsymbol{x}$, $\boldsymbol{v}^\top \phi_{\boldsymbol{\theta}}(\boldsymbol{x}) \neq 0$

are not equivalent since in the first statement, the null set for $\boldsymbol{\theta}$ may depend on $\boldsymbol{v}$, and the union of the null sets for all $\boldsymbol{v}$ is an uncountable union. Perhaps the simplest counterexample is $n = p = 2$, $\boldsymbol{\theta} \sim \mathcal{N}(0, \boldsymbol{I}_2)$ and $\phi_{\boldsymbol{\theta}}(\boldsymbol{x}) := \boldsymbol{\theta}$, which satisfies the first but not the second statement. For our rescue, we can replace the uncountable analytic function family $(\boldsymbol{x} \mapsto \boldsymbol{v}^\top \phi_{\boldsymbol{\theta}}(\boldsymbol{x}))_{\boldsymbol{v} \in \mathbb{R}^p \setminus \{0\}}$ by the single analytic function $(\boldsymbol{\theta}, \boldsymbol{X}) \mapsto \det(\phi_{\boldsymbol{\theta}}(\boldsymbol{X}))$:

**Proposition 9** (Random feature maps). *Consider feature maps $\phi_{\boldsymbol{\theta}} : \mathbb{R}^d \to \mathbb{R}^p$ with (random) parameter $\boldsymbol{\theta} \in \mathbb{R}^q$. Suppose the map $(\boldsymbol{\theta}, \boldsymbol{x}) \mapsto \phi_{\boldsymbol{\theta}}(\boldsymbol{x})$ is analytic and that $\boldsymbol{\theta}$ and $\boldsymbol{x}$ are independent and have Lebesgue densities. If there exist fixed $\widetilde{\boldsymbol{\theta}} \in \mathbb{R}^q, \widetilde{\boldsymbol{X}} \in \mathbb{R}^{p \times d}$ with $\det(\phi_{\widetilde{\boldsymbol{\theta}}}(\widetilde{\boldsymbol{X}})) \neq 0$, then almost surely over $\boldsymbol{\theta}$, (FRK) holds for all $n$ for $\boldsymbol{z} = \phi_{\boldsymbol{\theta}}(\boldsymbol{x})$.*

*Proof.* Consider the analytic map $(\boldsymbol{\theta}, \boldsymbol{X}) \mapsto \det(\phi_{\boldsymbol{\theta}}(\boldsymbol{X}))$. Suppose there exist $\widetilde{\boldsymbol{\theta}} \in \mathbb{R}^q$ and $\widetilde{\boldsymbol{X}} \in \mathbb{R}^{p \times d}$ with $\det(\phi_{\widetilde{\boldsymbol{\theta}}}(\widetilde{\boldsymbol{X}})) \neq 0$. Then, Theorem H.3 tells us that $\det(\tilde{\phi}(\boldsymbol{\theta}, \boldsymbol{X})) \neq 0$ for almost all $(\boldsymbol{\theta}, \boldsymbol{X})$. This implies that for almost all $\boldsymbol{\theta}$, we have for almost all $\boldsymbol{X}$ that $\det(\phi_{\boldsymbol{\theta}}(\boldsymbol{X})) \neq 0$. Since by assumption, $\boldsymbol{\theta}$ has a density, this implies that almost surely over $\boldsymbol{\theta}$, there exists $\boldsymbol{X}$ such that $\det(\phi_{\boldsymbol{\theta}}(\boldsymbol{X})) \neq 0$. Since all $\phi_{\boldsymbol{\theta}}$ are analytic, the claim now follows using (d) $\Rightarrow$ (b) from Proposition 6. (The proof of (d) $\Rightarrow$ (a) $\Leftrightarrow$ (b) in Proposition 6 does not require (MOM).) $\qquad \square$

### H.2 RANDOM NETWORKS WITH BIASES

In order to prove (FRK) for random deep neural networks, we pursue slightly different approaches for networks with bias (this section) and without bias (Appendix H.3). In both approaches, we consider a property related to the diversity of the $\boldsymbol{x}_1, \ldots, \boldsymbol{x}_n \in \mathbb{R}^d$ and proceed as follows:

(1) **Projections**: Ensure that random projections of the $\boldsymbol{x}_1, \ldots, \boldsymbol{x}_n$ almost surely preserve the diversity property, such that it is sufficient to consider the case $d = 1$.

(2) **Propagation**: Using the projection result from (1), prove that if the inputs to a layer have the diversity property, then the outputs also have the diversity property almost surely over the random parameters of the layer.

(3) **Independence**: Prove that if the inputs to the last layer have the diversity property and $n = p$, then the outputs of the last layer are almost surely linearly independent.

Our main tools will be the identity theorems for analytic functions (Theorem H.2 and Theorem H.3), expanding $\sigma$ into its power series around a point and the Leibniz formula for the determinant of a $n \times n$ matrix, which is based on the permutation group $S_n$ on $[n]$.

We consider two diversity properties:

(a) The first property is that $\boldsymbol{x}_1, \ldots, \boldsymbol{x}_n$ are distinct. This is the weakest possible diversity property. However, it cannot always be used for networks without (random) biases: For example, if $\sigma$ is an even function and $\boldsymbol{x}_i = -\boldsymbol{x}_j$ for some $i \neq j$, the propagation property is violated. As another example, if $\sigma(0) = 0$ and $\boldsymbol{x}_i = \boldsymbol{0}$ for some $i$, the independence property is violated.

(b) The second property, which works for networks with and without bias, is the property that $\boldsymbol{x}_1, \ldots, \boldsymbol{x}_n$ are independent nonatomic random variables.

Using the first property yields shorter proofs and a slightly stronger theorem for networks with bias, Theorem H.9. If we only care about probability distributions $P_X$ on $\boldsymbol{x}$ that almost surely generate distinct $\boldsymbol{x}_1, \ldots, \boldsymbol{x}_n$, these probability distributions are exactly the nonatomic distributions. Hence from the viewpoint of probability distributions $P_X$ on $\boldsymbol{x}$, which we take in the main part of the paper, the first property (a) does not provide a benefit over the second property (b) except for the shorter proofs.

The advantage of having biases is in being able to choose the point in which $\sigma$ is Taylor-expanded. The following lemma shows that this choice enables us to make certain coefficients of the Taylor expansion nonzero:

**Lemma H.5.** *Let $m \geq 1$. Let $\sigma : \mathbb{R} \to \mathbb{R}$ be analytic and not a polynomial of degree less than $m$. Then, there exists $b \in \mathbb{R}$ such that the Taylor expansion*

$$\sigma(x) = \sum_{k=0}^{\infty} a_k (x - b)^k$$

*of $\sigma$ around $b$ satisfies $a_0, \ldots, a_m \neq 0$.*

*Proof.* Since $\sigma$ is not a polynomial of degree less than $m$, neither of the derivatives $\sigma^{(0)}, \ldots, \sigma^{(m)}$ is the zero function. Since all of these derivatives are analytic, the set

$$\bigcup_{k=0}^{m} \{ b \in \mathbb{R} \mid \sigma^{(k)}(b) = 0 \}$$

is (by Theorem H.3) a finite union of Lebesgue null sets and hence a Lebesgue null set. Hence, there exists $b \in \mathbb{R}$ such that $\sigma^{(0)}(b) \neq 0, \ldots, \sigma^{(m)}(b) \neq 0$. This implies that the corresponding coefficients $a_0, \ldots, a_m$ in the Taylor expansion around $b$ are nonzero. $\qquad \square$

We now prove our three-step program (1) – (3) from above for the distinctness property.

**Lemma H.6** (Projections of distinct variables)**.** *If $\boldsymbol{x}_1, \ldots, \boldsymbol{x}_n \in \mathbb{R}^d$ are distinct, there exists $\boldsymbol{u} \in \mathbb{R}^d$ such that $\boldsymbol{u}^\top \boldsymbol{x}_1, \ldots, \boldsymbol{u}^\top \boldsymbol{x}_d$ are also distinct.*

*Proof.* We essentially follow the corresponding proof in Lemma 4.3 in Nguyen & Hein (2017). By the identity theorem (Theorem H.3), the functions

$$f_{ij} : \mathbb{R}^d \to \mathbb{R}, \boldsymbol{u} \mapsto \boldsymbol{u}^\top \boldsymbol{x}_i - \boldsymbol{u}^\top \boldsymbol{x}_j$$

for $i, j \in [n], i \neq j$ are nonzero almost everywhere, hence there exists $\boldsymbol{u} \in \mathbb{R}^p$ such that $\boldsymbol{u}^\top \boldsymbol{x}_1, \ldots, \boldsymbol{u}^\top \boldsymbol{x}_n$ are all distinct. $\qquad \square$

**Lemma H.7** (Propagation of distinct variables)**.** *Let $\sigma : \mathbb{R} \to \mathbb{R}$ be analytic and non-constant. If $\boldsymbol{x}_1, \ldots, \boldsymbol{x}_n \in \mathbb{R}^d$ are distinct, then for (Lebesgue-) almost all $(\boldsymbol{W}, \boldsymbol{b}) \in \mathbb{R}^{p \times d} \times \mathbb{R}^p$, the vectors $\sigma(\boldsymbol{W} \boldsymbol{x}_i + \boldsymbol{b})$ (with $\sigma$ applied element-wise) are also distinct.*

*Proof.* **Step 1: Bias.** By assumption, $\sigma$ is not a polynomial of degree less than $m = 2$. By Lemma H.5, there exist $(a_k)_{k\geq 0}$, $b \in \mathbb{R}$ and $\varepsilon > 0$ such that $a_0, a_1 \neq 0$ and for all $x \in \mathbb{R}$ with $|x - b| < \varepsilon$,

$$\sigma(x) = \sum_{k=0}^{\infty} a_k (x - b)^k .$$

**Step 2: Weight.** By Lemma H.6, we can choose $\boldsymbol{u} \in \mathbb{R}^d$ such that $\boldsymbol{u}^\top \boldsymbol{x}_1, \ldots, \boldsymbol{u}^\top \boldsymbol{x}_n$ are distinct. Now, consider $i, j \in [n]$ with $i \neq j$. The function

$$f_{ij} : \mathbb{R} \to \mathbb{R}, \lambda \mapsto \sigma(\lambda \boldsymbol{u}^\top \boldsymbol{x}_i + b) - \sigma(\lambda \boldsymbol{u}^\top \boldsymbol{x}_j + b)$$

satisfies

$$f_{ij}(\lambda) = \sum_{k=0}^{\infty} a_k ((\boldsymbol{u}^\top \boldsymbol{x}_i)^k - (\boldsymbol{u}^\top \boldsymbol{x}_j)^k)\lambda^k$$

for sufficiently small $|\lambda|$. Here, the coefficient $a_k((\boldsymbol{u}^\top \boldsymbol{x}_i)^k - (\boldsymbol{u}^\top \boldsymbol{x}_j)^k)$ is nonzero for $k = 1$, and hence $f_{ij}$ is not the zero function. Using the identity theorem again (as in the proof of Lemma H.6, we find that there exists $\lambda \in \mathbb{R}$ with $f_{ij}(\lambda) \neq 0$ for all $i, j \in [n]$ with $i \neq j$.

**Step 3: Generalization.** Now, choose

$$\boldsymbol{W} := \begin{pmatrix} \lambda \boldsymbol{u}^\top \\ \vdots \\ \lambda \boldsymbol{u}^\top \end{pmatrix} \in \mathbb{R}^{p \times d}, \qquad \boldsymbol{b} := \begin{pmatrix} b \\ \vdots \\ b \end{pmatrix} \in \mathbb{R}^p .$$

Then, by construction, the first components $(\sigma(\boldsymbol{W}\boldsymbol{x}_i + \boldsymbol{b}))_1$ for $i \in [n]$ are distinct. Hence, the analytic function

$$(\boldsymbol{W}, \boldsymbol{b}) \mapsto \prod_{\substack{i,j \in [n] \\ i \neq j}} ((\sigma(\boldsymbol{W}\boldsymbol{x}_i + \boldsymbol{b}))_1 - (\sigma(\boldsymbol{W}\boldsymbol{x}_j + \boldsymbol{b}))_1)$$

is not the zero function. By the identity theorem (Theorem H.3), for (Lebesgue-) almost all $(\boldsymbol{W}, \boldsymbol{b})$, the first components of the vectors $\sigma(\boldsymbol{W}\boldsymbol{x}_i + \boldsymbol{b})$ are distinct, and therefore also the vectors themselves are distinct. $\square$

**Lemma H.8** (Independence from distinct variables)**.** *Let $p, d \geq 1$. Let $\sigma : \mathbb{R} \to \mathbb{R}$ be analytic but not a polynomial of degree less than $p - 1$. If $\boldsymbol{x}_1, \ldots, \boldsymbol{x}_p \in \mathbb{R}^d$ are distinct, then for (Lebesgue-) almost all $\boldsymbol{W} \in \mathbb{R}^{p \times d}$ and $\boldsymbol{b} \in \mathbb{R}^p$, the vectors $\sigma(\boldsymbol{W}\boldsymbol{x}_1 + \boldsymbol{b}), \ldots, \sigma(\boldsymbol{W}\boldsymbol{x}_p + \boldsymbol{b})$ are linearly independent.*

*Proof.* **Step 1: Preparation.** By Lemma H.5, there exist $(a_k)_{k\geq 0}$, $b \in \mathbb{R}$ and $\varepsilon > 0$ such that $a_0, \ldots, a_{p-1} \neq 0$ and for all $x \in \mathbb{R}$ with $|x - b| < \varepsilon$,

$$\sigma(x) = \sum_{k=0}^{\infty} a_k (x - b)^k .$$

By Lemma H.6, there exists $\boldsymbol{u} \in \mathbb{R}^d$ such that $x_i := \boldsymbol{u}^\top \boldsymbol{x}_i$ for $i \in [p]$ are all distinct. Using the fact that Vandermonde matrices of distinct $x_i$ are invertible and using the Leibniz formula for the determinant (with the permutation group $S_p$ on $[p]$), we obtain

$$D_x := \sum_{\pi \in S_p} \mathrm{sgn}(\pi) \prod_{i=1}^{p} x_{\pi(i)}^{i-1} = \det \begin{pmatrix} x_1^0 & \cdots & x_p^0 \\ \vdots & \ddots & \vdots \\ x_1^{p-1} & \cdots & x_p^{p-1} \end{pmatrix} \neq 0 . \tag{10}$$

**Step 2: Determinant expansion.** Define the analytic function

$$f : \mathbb{R}^p \to \mathbb{R}, \boldsymbol{w} \mapsto \det \begin{pmatrix} \sigma(w_1 x_1 + b) & \cdots & \sigma(w_p x_1 + b) \\ \vdots & \ddots & \vdots \\ \sigma(w_1 x_p + b) & \cdots & \sigma(w_p x_p + b) \end{pmatrix} .$$

For small enough $\|\boldsymbol{w}\|_2$, we can use the Leibniz formula for the determinant to write

$$
\begin{aligned}
f(\boldsymbol{w}) &= \sum_{\pi \in S_p} \operatorname{sgn}(\pi) \prod_{i=1}^{p} \sum_{k=0}^{\infty} a_k (w_i x_{\pi(i)})^k \\
&= \sum_{k_1,\ldots,k_p \geq 0} \sum_{\pi \in S_p} \operatorname{sgn}(\pi) \prod_{i=1}^{p} a_{k_i} w_i^{k_i} x_{\pi(i)}^{k_i} \\
&= \sum_{k_1,\ldots,k_p \geq 0} \left( \prod_{i=1}^{p} a_{k_i} \right) \left( \sum_{\pi \in S_p} \operatorname{sgn}(\pi) \prod_{i=1}^{p} x_{\pi(i)}^{k_i} \right) w_1^{k_1} \cdots w_p^{k_p} \ .
\end{aligned}
$$

For $k_i := i - 1$, we find the coefficient of $w_1^{k_1} \cdots w_p^{k_p}$ to be

$$
\left( \prod_{i=1}^{p} a_{k_i} \right) \left( \sum_{\pi \in S_p} \operatorname{sgn}(\pi) \prod_{i=1}^{p} x_{\pi(i)}^{k_i} \right) \overset{(10)}{=} \left( \prod_{i=1}^{p} a_{k_i} \right) \cdot D_x \neq 0 \ ,
$$

hence $f$ is not the zero function and there exists $\boldsymbol{w} \in \mathbb{R}^p$ such that $f(\boldsymbol{w}) \neq 0$.

**Step 3: Generalization.** Consider the analytic function

$$
g(\boldsymbol{W}, \boldsymbol{b}) := \det \begin{pmatrix} \sigma(\boldsymbol{W}\boldsymbol{x}_1 + \boldsymbol{b})^\top \\ \vdots \\ \sigma(\boldsymbol{W}\boldsymbol{x}_p + \boldsymbol{b})^\top \end{pmatrix} \ .
$$

When setting $\boldsymbol{W} := \boldsymbol{w}\boldsymbol{u}^\top \in \mathbb{R}^{p \times d}$ and $\boldsymbol{b} := (b, \ldots, b)^\top \in \mathbb{R}^p$, we obtain from Step 2 that $g(\boldsymbol{W}, \boldsymbol{b}) = f(\boldsymbol{w}) \neq 0$, hence $g$ is not the zero function. But then, by the identity theorem (Theorem H.3), $g$ is nonzero for (Lebesgue-) almost all $(\boldsymbol{W}, \boldsymbol{b})$. $\qquad\square$

**Theorem H.9.** *Let $p, d \geq 1$, let $\sigma : \mathbb{R} \to \mathbb{R}$ be analytic but not a polynomial of degree less than $\max\{1, p-1\}$ and let $\boldsymbol{x}_1, \ldots, \boldsymbol{x}_p \in \mathbb{R}^d$ be distinct. Let $L \geq 1$ and $d_0 := d, d_1, \ldots, d_{L-1} \geq 1, d_L := p$. For $l \in \{0, \ldots, l-1\}$, let $\boldsymbol{W}^{(l)} \in \mathbb{R}^{d_{l+1} \times d_l}$ and $\boldsymbol{b}^{(l)} \in \mathbb{R}^{d_{l+1}}$ be random variables such that $\boldsymbol{\theta} := (\boldsymbol{W}^{(0)}, \ldots, \boldsymbol{W}^{(L-1)}, \boldsymbol{b}^{(0)}, \ldots, \boldsymbol{b}^{(L-1)})$ has a Lebesgue density. Consider the random feature map given by*

$$
\phi_{\boldsymbol{\theta}}(\boldsymbol{x}^{(0)}) := \boldsymbol{x}^{(L)}, \quad \text{where} \quad \boldsymbol{x}^{(l+1)} := \boldsymbol{W}^{(l)} \boldsymbol{x}^{(l)} + \boldsymbol{b}^{(l)} \ .
$$

*Then, almost surely over $\boldsymbol{\theta}$, $\phi_{\boldsymbol{\theta}}(\boldsymbol{x}_1), \ldots, \phi_{\boldsymbol{\theta}}(\boldsymbol{x}_p)$ are linearly independent.*

*Proof.* By Lemma H.4, it suffices to consider the case where $\boldsymbol{\theta}$ has a standard normal distribution, since a standard normal distribution has a nonzero probability density everywhere. Especially, in this case, all weights and biases are independent. Using Lemma H.7 and that $\sigma$ is non-constant, it follows by induction on $l \in \{0, \ldots, L-1\}$ that $\boldsymbol{x}_1^{(l)}, \ldots, \boldsymbol{x}_p^{(l)}$ are distinct almost surely over $\boldsymbol{\theta}$. But if $\boldsymbol{x}_1^{(L-1)}, \ldots, \boldsymbol{x}_p^{(L-1)}$ are distinct, then $\boldsymbol{x}_1^{(L)}, \ldots, \boldsymbol{x}_p^{(L)}$ are linearly independent almost surely over $\boldsymbol{\theta}$ by Lemma H.8, which is what we wanted to show. $\qquad\square$

### H.3 RANDOM NETWORKS WITHOUT BIASES

As discussed at the beggining of Appendix H.2, we will now consider the property of having independent nonatomic random variables $\boldsymbol{x}_1, \ldots, \boldsymbol{x}_n$. We will again consider projection and propagation lemmas first.

**Lemma H.10** (Projections of nonatomic random variables)**.** *A random variable $\boldsymbol{x} \in \mathbb{R}^d$ is nonatomic iff for (Lebesgue-) almost all $\boldsymbol{u} \in \mathbb{R}^d$, $\boldsymbol{u}^\top \boldsymbol{x}$ is nonatomic.*

*Proof.* **Step 1: Decompose into subspace contributions.** For $k \in \{0, \ldots, d\}$, let

$$
\mathcal{S}_k := \{\boldsymbol{w} + V \mid \boldsymbol{w} \in \mathbb{R}^d, V \text{ linear subspace of } \mathbb{R}^d, \dim V = k\}
$$

be the set of $k$-dimensional affine subspaces of $\mathbb{R}^d$. Let $\mathcal{A}_0 := \{A \in \mathcal{S}_0 \mid P_X(A) > 0\}$, which corresponds to the set of atoms of $P_X$. We then recursively define "minimally contributing subspaces" for $k \in [d]$:

$$\mathcal{A}_k := \{A \in \mathcal{S}_k \mid P_X(A) > 0 \text{ and for all } \tilde{A} \in \mathcal{S}_{k-1} \text{ with } \tilde{A} \subseteq A, P_X(\tilde{A}) = 0\} .$$

We can define corresponding sets of "annihilating" vectors as follows: For $A = \boldsymbol{w} + V \in \mathcal{S}_k$, let $A^\perp := V^\perp = \{\boldsymbol{u} \in \mathbb{R}^d \mid \text{for all } \boldsymbol{v} \in V, \boldsymbol{u}^\top \boldsymbol{v} = 0\}$. This is well-defined since it is independent of the choice of $\boldsymbol{w}$. Define

$$\mathcal{U} := \bigcup_{k=0}^{d} \bigcup_{A \in \mathcal{A}_k} A^\perp$$

$$\mathcal{N} := \{\boldsymbol{u} \in \mathbb{R}^d \mid \boldsymbol{u}^\top \boldsymbol{x} \text{ is not nonatomic}\} .$$

We want to show $\mathcal{U} = \mathcal{N}$.

**Step 2: Show $\mathcal{U} \subseteq \mathcal{N}$.** Let $A = \boldsymbol{w} + V \in \mathcal{A}_k$ for some $k \in \{0, \ldots, d\}$ and let $\boldsymbol{u} \in A^\perp$. Then,

$$0 < P_X(A) = P(\boldsymbol{x} \in A) \leq P(\boldsymbol{u}^\top \boldsymbol{x} \in \{\boldsymbol{u}^\top (\boldsymbol{w} + \boldsymbol{v}) \mid \boldsymbol{v} \in V\}) = P(\boldsymbol{u}^\top \boldsymbol{x} = \boldsymbol{u}^\top \boldsymbol{w}) ,$$

which shows $\boldsymbol{u} \in \mathcal{N}$.

**Step 3: Show $\mathcal{N} \subseteq \mathcal{U}$.** Let $\boldsymbol{u} \in \mathcal{N}$, i.e. there exists $a \in \mathbb{R}$ such that $P(\boldsymbol{u}^\top \boldsymbol{x} = a) > 0$. Define $A := \{\boldsymbol{v} \in \mathbb{R}^d \mid \boldsymbol{u}^\top \boldsymbol{v} = a\}$. Then, $A$ is an affine subspace of $\mathbb{R}^d$ and we have $P_X(A) > 0$ by construction of $A$. Among all affine subspaces $\tilde{A}$ of $A$ with $P_X(\tilde{A} > 0)$, there exists one with minimal dimension. This subspace then satisfies $\tilde{A} \in \mathcal{A}_{\dim \tilde{A}}$ and it is not hard to show that $\boldsymbol{u} \in A^\perp \subseteq \tilde{A}^\perp$, hence $\boldsymbol{u} \in \mathcal{U}$.

**Step 4: For all $k \in \{0, \ldots, d\}$, $\mathcal{A}_k$ is countable.** To derive a contradiction, assume that $\mathcal{A}_k$ is uncountable. Then, there exists $\varepsilon > 0$ such that

$$\mathcal{A}_{k,\varepsilon} := \{A \in \mathcal{A}_k \mid P_X(A) \geq \varepsilon\}$$

is also uncountable. Pick an integer $n > 1/\varepsilon$ and pick $n$ distinct sets $A_1, \ldots, A_n \in \mathcal{A}_{k,\varepsilon}$. We will show by induction on $l \in [n]$ that $\tilde{A}_l := A_1 \cup \ldots \cup A_l$ satisfies

$$P_X(\tilde{A}_l) = P_X(A_1) + \ldots + P_X(A_l) ,$$

which will then yield the contradiction

$$1 \geq P_X(\tilde{A}_l) = P_X(A_1) + \ldots + P_X(A_n) \geq n\varepsilon > 1 .$$

Obviously, $P_X(\tilde{A}_1) = P_X(A_1)$. Assuming that the statement holds for $l \in [n-1]$, we first derive

$$P_X(\tilde{A}_{l+1}) = P_X(\tilde{A}_l \cup A_{l+1}) = P_X(\tilde{A}_l) + P_X(A_{l+1}) - P_X(\tilde{A}_l \cap A_{l+1})$$
$$= P_X(A_1) + \ldots + P_X(A_{l+1}) - P_X(\tilde{A}_l \cap A_{l+1}) .$$

For $i \neq j$, the intersection $A_i \cap A_j$ of two distinct $k$-dimensional affine subspaces is either empty or an affine subspace of dimension less than $k$, hence the definition of $\mathcal{A}_k$ yields $P_X(A_i \cap A_j) = 0$. Therefore,

$$P_X(\tilde{A}_l \cap A_{l+1}) = P_X((A_1 \cap A_{l+1}) \cup \ldots \cup (A_l \cap A_{l+1}))$$
$$\leq P_X(A_1 \cap A_{l+1}) + \ldots + P_X(A_l \cap A_{l+1})$$
$$= 0 + \ldots + 0 = 0 ,$$

which concludes the induction.

**Step 5: Conclusion.** Let $A \in \mathcal{S}_k$, then $A^\perp$ is a $(d-k)$-dimensional linear subspace of $\mathbb{R}^d$. If $\boldsymbol{x}$ is not nonatomic, the set $\mathcal{A}_0$ is not empty, and hence

$$\mathcal{N} = \mathcal{U} \supseteq \bigcup_{A \in \mathcal{A}_0} A^\perp = \mathbb{R}^d ,$$

which means that $\mathcal{N}$ is not a Lebesgue null set. Conversely, if $\boldsymbol{x}$ is nonatomic, the set $\mathcal{A}_0$ is empty, and therefore

$$\mathcal{N} = \mathcal{U} = \bigcup_{k=1}^{d} \bigcup_{A \in \mathcal{A}_k} A^\perp$$

is (by Step 4) a countable union of proper affine subspaces of $\mathbb{R}^d$, all of which are Lebesgue null sets. Therefore, $\mathcal{N}$ is a Lebesgue null set. □

**Lemma H.11** (Propagation of nonatomic random variables). *Let the random variable $\boldsymbol{x} \in \mathbb{R}^d$ be nonatomic.*

    *(a) For all $p \geq 1$ and (Lebesgue-) almost all $\boldsymbol{W} \in \mathbb{R}^{p \times d}$, $\boldsymbol{W}\boldsymbol{x}$ is nonatomic.*
    *(b) For all $\boldsymbol{b} \in \mathbb{R}^d$, $\boldsymbol{x} + \boldsymbol{b}$ is nonatomic.*
    *(c) If $\sigma : \mathbb{R} \to \mathbb{R}$ is analytic and not constant, then $\sigma(\boldsymbol{x}) \in \mathbb{R}^d$ is nonatomic, where $\sigma$ is applied element-wise.*

*Proof.*

    (a) By Lemma H.10, the set $\mathcal{N} := \{\boldsymbol{u} \in \mathbb{R}^d \mid \boldsymbol{u}^\top \boldsymbol{x}$ is not nonatomic$\}$ is a Lebesgue null set. If $\boldsymbol{w}_1$ is the first row of $\boldsymbol{W}$, then clearly,

$$\{\boldsymbol{W} \in \mathbb{R}^{p \times d} \mid \boldsymbol{W}\boldsymbol{x} \text{ is not nonatomic}\} \subseteq \{\boldsymbol{W} \in \mathbb{R}^{p \times d} \mid \boldsymbol{w}_1 \in \mathcal{N}\} \,,$$

    where the right-hand side is a Lebesgue null set. This proves the claim.
    (b) This is trivial.
    (c) Let $\boldsymbol{z} \in \mathbb{R}^d$. Since the function $\mathbb{R} \to \mathbb{R}, x \mapsto \sigma(x) - z_i$ is analytic and not the zero function by assumption, its zero set $\sigma^{-1}(\{z_i\})$ is countable by the identity theorem, Theorem H.2. Therefore, the set

$$\sigma^{-1}(\{\boldsymbol{z}\}) = \{\tilde{\boldsymbol{x}} \in \mathbb{R}^d \mid \sigma(\tilde{\boldsymbol{x}}) = \boldsymbol{z}\} = \sigma^{-1}(\{z_1\}) \times \ldots \times \sigma^{-1}(\{z_d\})$$

    is also countable. Thus, since $\boldsymbol{x}$ is nonatomic, we obtain

$$P(\sigma(\boldsymbol{x}) = \boldsymbol{z}) = P(\boldsymbol{x} \in \sigma^{-1}(\{\boldsymbol{z}\})) = \sum_{\tilde{\boldsymbol{x}} \in \sigma^{-1}(\{\boldsymbol{z}\})} P(\boldsymbol{x} = \tilde{\boldsymbol{x}}) = \sum_{\tilde{\boldsymbol{x}} \in \sigma^{-1}(\{\boldsymbol{z}\})} 0 = 0 \,. \qquad \square$$

For proving independence from nonatomic random variables, we need some preparation. In the proof of the independence result for the case with biases (Lemma H.8), a Vandermonde matrix appeared. In the case of networks without bias, we will not be able to use Lemma H.5 to control the power series coefficients of $\sigma$, which requires us to treat Vandermonde matrices with more general exponents.

**Lemma H.12** (Random Vandermonde-type matrices). *For $n \geq 1$, let $x_1, \ldots, x_n$ be independent nonatomic $\mathbb{R}$-valued random variables and let $k_1, \ldots, k_n$ be distinct non-negative integers. Then, the random Vandermonde-type matrix*

$$\boldsymbol{V} := \boldsymbol{V}_{k_1,\ldots,k_n}(x_1, \ldots, x_n) := \begin{pmatrix} x_1^{k_1} & \cdots & x_n^{k_1} \\ \vdots & \ddots & \vdots \\ x_1^{k_n} & \cdots & x_n^{k_n} \end{pmatrix}$$

*is invertible with probability one.*

*Proof.* By swapping rows of $\boldsymbol{V}$, we can assume without loss of generality that $0 \leq k_1 < k_2 < \ldots < k_n$. We prove the statement by induction on $n$. For $n = 1$, $\boldsymbol{V}$ is invertible whenever $x_1 \neq 0$, and this happens with probability one. Now assume that the statement holds for $n - 1 \geq 1$. For $i \in [n]$, define the submatrices

$$\widehat{\boldsymbol{V}}_i := \boldsymbol{V}_{k_1,\ldots,k_{n-1}}(x_1, \ldots, x_{i-1}, x_{i+1}, \ldots, x_n) \,.$$

Since the statement holds for $n - 1$, $\widehat{\boldsymbol{V}}_n$ is invertible with probability one. Now, fix any such $x_1, \ldots, x_{n-1}$ where $\widehat{\boldsymbol{V}}_n$ is invertible. Especially, $C := \det(\widehat{\boldsymbol{V}}_n)$ is a non-zero constant. We will show that $\boldsymbol{V}$ is then invertible almost surely over $x_n$. For this, we compute the determinant of $\boldsymbol{V}$ using the Laplace expansion with respect to the last row of $\boldsymbol{V}$ as

$$f(x_n) := \det(\boldsymbol{V}) = \sum_{i=1}^{n}(-1)^{i+n} x_i^{k_n} \det(\widehat{\boldsymbol{V}}_i) = C x_n^{k_n} + \sum_{i=1}^{n-1}(-1)^{i+n} x_i^{k_n} \det(\widehat{\boldsymbol{V}}_i) \,.$$

By the Leibniz formula, for $i \in [n - 1]$, $\det(\widehat{\boldsymbol{V}}_i)$ is a polynomial in $x_n$ of degree $\leq k_{n-1} < k$. Hence, $f$ is a nonzero polynomial in $x_n$ of degree $k_n$ and has at most $k_n$ zeros. Since any finite set is a null set with respect to a nonatomic distribution, it follows that $\det(\boldsymbol{V}) \neq 0$ almost surely over $x_n$. Since the assumptions on $x_1, \ldots, x_{n-1}$ are also satisfied almost surely, the statement follows. $\qquad \square$

As in the case of networks with biases, we will prove the independence result by first reducing it to the case $d = 1$. Since we also want to perform this reduction for random Fourier features in Proposition I.1, we state the reduction to the $d = 1$ case as a separate result, Lemma H.14, and define the $d = 1$ case in the following definition.

**Definition H.13** (Non-degenerate). Let $p, q \geq 1$. We call a function $f : \mathbb{R}^q \to \mathbb{R}^q$ *non-degenerate* if it is analytic and for arbitrary independent $\mathbb{R}$-valued nonatomic random variables $x_1, \ldots, x_p$, there almost surely exists $\boldsymbol{w} = \boldsymbol{w}_{x_1,\ldots,x_p} \in \mathbb{R}^q$ with

$$\det \begin{pmatrix} f(\boldsymbol{w}x_1)^\top \\ \vdots \\ f(\boldsymbol{w}x_p)^\top \end{pmatrix} \neq 0 \,. \qquad \blacktriangleleft$$

**Lemma H.14.** *Let $f : \mathbb{R}^q \to \mathbb{R}^p$ be non-degenerate, let $\boldsymbol{W} \in \mathbb{R}^{q \times d}$ be a random variable with a Lebesgue density and let $\boldsymbol{x}_1, \ldots, \boldsymbol{x}_p \in \mathbb{R}^d$ be independent nonatomic random variables. Then,*

$$\det \begin{pmatrix} f(\boldsymbol{W}\boldsymbol{x}_1)^\top \\ \vdots \\ f(\boldsymbol{W}\boldsymbol{x}_p)^\top \end{pmatrix} \neq 0$$

*almost surely over $\boldsymbol{W}$ and $\boldsymbol{x}_1, \ldots, \boldsymbol{x}_p$.*

*Proof.* By Lemma H.10, there exists $\boldsymbol{u} \in \mathbb{R}^d$ such that for all $i \in [p]$, $x_i := \boldsymbol{u}^\top \boldsymbol{x}_i \in \mathbb{R}$ is nonatomic. Obviously, $x_1, \ldots, x_n$ are independent. Fix $x_1, \ldots, x_n$ such that $\boldsymbol{w} = \boldsymbol{w}_{x_1,\ldots,x_p} \in \mathbb{R}^q$ as in Definition H.13 exists, which is true with probability one since $f$ is non-degenerate. Then, for $\widetilde{\boldsymbol{W}} := \boldsymbol{w}\boldsymbol{u}^\top$, we have

$$g(\widetilde{\boldsymbol{W}}) := \det \begin{pmatrix} f(\widetilde{\boldsymbol{W}}\boldsymbol{x}_1)^\top \\ \vdots \\ f(\widetilde{\boldsymbol{W}}\boldsymbol{x}_p)^\top \end{pmatrix} = \det \begin{pmatrix} f(\boldsymbol{w}x_1)^\top \\ \vdots \\ f(\boldsymbol{w}x_p)^\top \end{pmatrix} \neq 0 \,.$$

Since $g$ is a non-zero analytic function, Theorem H.3 shows that $g$ is only zero on a Lebesgue null set, and this null set is also a null set with respect to the distribution of $\boldsymbol{W}$ since $\boldsymbol{W}$ has a Lebesgue density. Hence, $g(\boldsymbol{W}) \neq 0$ almost surely over $\boldsymbol{W}$. $\qquad \square$

The following lemma proves the $d = 1$ version of the independence result, which can then be upgraded to the general case using Lemma H.14.

**Lemma H.15** (Independence from nonatomic random variables). *Let $p \geq 1$. Let $\sigma : \mathbb{R} \to \mathbb{R}$ be analytic and not a polynomial with less than $p$ nonzero coefficients. Then, the elementwise application function $f : \mathbb{R}^p \to \mathbb{R}^p, \boldsymbol{x} \mapsto (\sigma(x_1), \ldots, \sigma(x_p))^\top$ is non-degenerate in the sense of Definition H.13.*

*Proof.* Let $x_1, \ldots, x_p$ be independent scalar nonatomic random variables.

**Step 1: Power series coefficients.** Since $\sigma$ is analytic, there exists $\varepsilon > 0$ and coefficients $(a_k)_{k \geq 0}$ such that

$$\sigma(z) = \sum_{k=0}^\infty a_k z^k$$

for $z \in \mathbb{R}$ with $|z| < \varepsilon$. Let $K := \{k \geq 0 \mid a_k \neq 0\}$. Then, $|K| \geq p$: Assume that $|K| \leq p - 1$, then the polynomial

$$h : \mathbb{R} \to \mathbb{R}, z \mapsto \sum_{k \in K} a_k z^k$$

equals $\sigma$ on $(-\varepsilon, \varepsilon)$. Hence, the function $g := \sigma - h$ is zero on $(-\varepsilon, \varepsilon)$. By Theorem H.2, $g$ is the zero function and hence $\sigma = h$ is a polynomial with less than $p$ nonzero coefficients, which we assumed not to be the case.

**Step 2: Condition on the $x_i$.** By Step 1, we can choose indices $k_1 < k_2 < \ldots < k_p$ with $k_1, \ldots, k_p \in K$. Then, by Lemma H.12 and the Leibniz formula for the determinant, we have

$$D_x := \sum_{\pi \in S_p} \mathrm{sgn}(\pi) x_{\pi(1)}^{k_1} \cdot \ldots \cdot x_{\pi(p)}^{k_p} = \det \begin{pmatrix} x_1^{k_1} & \ldots & x_p^{k_1} \\ \vdots & \ddots & \vdots \\ x_1^{k_p} & \ldots & x_p^{k_p} \end{pmatrix} \neq 0 \tag{11}$$

with probability one.

**Step 3: Determinant power series.** Now, fix a realization of $x_1, \ldots, x_p$ such that (11) holds. For $\boldsymbol{w} \in \mathbb{R}^p$ with sufficiently small $\|\boldsymbol{w}\|_\infty$, we can write

$$\begin{aligned} g(\boldsymbol{w}) &:= \det \begin{pmatrix} f(\boldsymbol{w} x_1)^\top \\ \vdots \\ f(\boldsymbol{w} x_p)^\top \end{pmatrix} \\ &= \det \begin{pmatrix} \sigma(w_1 x_1) & \ldots & \sigma(w_p x_1) \\ \vdots & \ddots & \vdots \\ \sigma(w_1 x_p) & \ldots & \sigma(w_p x_p) \end{pmatrix} \\ &= \sum_{\pi \in S_p} \mathrm{sgn}(\pi) \prod_{i=1}^p \sum_{k=0}^\infty a_k (w_i x_{\pi(i)})^k \\ &= \sum_{k_1, \ldots, k_p \geq 0} \sum_{\pi \in S_p} \mathrm{sgn}(\pi) \prod_{i=1}^p a_{k_i} w_i^{k_i} x_{\pi(i)}^{k_i} \\ &= \sum_{k_1, \ldots, k_p \geq 0} \left( \prod_{i=1}^p a_{k_i} \right) \left( \sum_{\pi \in S_p} \mathrm{sgn}(\pi) \prod_{i=1}^p x_{\pi(i)}^{k_i} \right) w_1^{k_1} \cdot \ldots \cdot w_p^{k_p} . \end{aligned} \tag{12}$$

Now, consider the special values $k_1, \ldots, k_p$ chosen in Step 2. Since $k_i \in K$, we have $a_{k_i} \neq 0$. The coefficient of the multivariate monomial $w_1^{k_1} \cdot \ldots \cdot w_p^{k_p}$ in Eq. (12) is

$$\left( \prod_{i=1}^p a_{k_i} \right) \left( \sum_{\pi \in S_p} \mathrm{sgn}(\pi) \prod_{i=1}^p x_{\pi(i)}^{k_i} \right) \overset{(11)}{=} a_{k_1} \cdot \ldots \cdot a_{k_p} \cdot D_x \neq 0 .$$

If $g$ was the zero function, all derivatives of $g$ would be zero and therefore the coefficients of all monomials would be zero, which is not the case. Hence, there exists $\boldsymbol{w} \in \mathbb{R}^p$ with $g(\boldsymbol{w}) \neq 0$. This shows that $f$ is non-degenerate. $\qquad\square$

## H.4 RANDOM NETWORKS: CONCLUSION

In the following, we will prove Theorem 10 and discuss some possible extensions and limitations.

**Theorem 10** (Random neural networks). *Let $d, p, L \geq 1$, let $\sigma : \mathbb{R} \to \mathbb{R}$ be analytic and let the layer sizes be $d_0 = d$, $d_1, \ldots, d_{L-1} \geq 1$ and $d_L = p$. Let $\boldsymbol{W}^{(l)} \in \mathbb{R}^{d_{l+1} \times d_l}$ for $l \in \{0, \ldots, L-1\}$ be random variables and consider the two cases where*

*(a) $\sigma$ is not a polynomial with less than $p$ nonzero coefficients, $\boldsymbol{\theta} := (\boldsymbol{W}^{(0)}, \ldots, \boldsymbol{W}^{(L-1)})$ and the random feature map $\phi_{\boldsymbol{\theta}} : \mathbb{R}^d \to \mathbb{R}^p$ is recursively defined by*

$$\phi(\boldsymbol{x}^{(0)}) := \boldsymbol{x}^{(L)}, \quad \boldsymbol{x}^{(l+1)} := \sigma(\boldsymbol{W}^{(l)} \boldsymbol{x}^{(l)}) .$$

*(b) $\sigma$ is not a polynomial of degree $< p - 1$, $\boldsymbol{\theta} := (\boldsymbol{W}^{(0)}, \ldots, \boldsymbol{W}^{(L-1)}, \boldsymbol{b}^{(0)}, \ldots, \boldsymbol{b}^{(L-1)})$ with random variables $\boldsymbol{b}^{(l)} \in \mathbb{R}^{d_{l+1}}$ for $l \in \{0, \ldots, L-1\}$, and the random feature map $\phi_{\boldsymbol{\theta}} : \mathbb{R}^d \to \mathbb{R}^p$ is recursively defined by*

$$\phi(\boldsymbol{x}^{(0)}) := \boldsymbol{x}^{(L)}, \quad \boldsymbol{x}^{(l+1)} := \sigma(\boldsymbol{W}^{(l)} \boldsymbol{x}^{(l)} + \boldsymbol{b}^{(l)}) .$$

*In both cases, if $\boldsymbol{\theta}$ has a Lebesgue density and $\boldsymbol{x}$ is nonatomic, then (FRK) holds for all $n$ and almost surely over $\boldsymbol{\theta}$.*

*Proof.* By Lemma H.4, it suffices to consider the case where $\boldsymbol{\theta}$ has a standard normal distribution, since a standard normal distribution has a nonzero probability density everywhere. Especially, we can assume that all parameters in $\boldsymbol{\theta}$ are independent. By Proposition 6, we only need to prove (FRK) for $n = p$. Let $\boldsymbol{x}_1^{(0)}, \ldots, \boldsymbol{x}_p^{(0)} \sim P_X$ be i.i.d. nonatomic random variables.

(a) If $p = 1$, $\sigma$ is allowed to be a non-zero constant function. In this case, the feature map $\phi_{\boldsymbol{\theta}}$ is constant and non-zero with $p = 1$, which means that (FRK) holds. In the following, we thus assume that $\sigma$ is non-constant. Let $\boldsymbol{x}^{(0)} \sim P_X$. Since $\boldsymbol{x}^{(0)}$ is nonatomic and $\sigma$ is non-constant, an inductive application of Lemma H.11 yields that almost surely over $\boldsymbol{\theta}$, $\boldsymbol{x}^{(L-1)}$ is also nonatomic. Hence, $\boldsymbol{x}_1^{(L-1)}, \ldots, \boldsymbol{x}_p^{(L-1)}$ are independent and nonatomic almost surely over $\boldsymbol{\theta}$. But by Lemma H.15, the elementwise application of $\sigma$ is non-degenerate, and hence by Lemma H.14, we almost surely have

$$
\det \begin{pmatrix} \sigma(\boldsymbol{W}^{(L-1)} \boldsymbol{x}_1^{(L-1)})^\top \\ \vdots \\ \sigma(\boldsymbol{W}^{(L-1)} \boldsymbol{x}_p^{(L-1)})^\top \end{pmatrix} \neq 0 \,,
$$

which implies that (FRK) holds for $n = p$ almost surely over $\boldsymbol{\theta}$.

(b) If $p = 1$ and $\sigma$ is a polynomial of degree $p - 1 = 0$, this means that $\sigma$ is a non-zero constant function. Like in case (a), this implies that $\phi_{\boldsymbol{\theta}}$ is constant and non-zero with $p = 1$, which means that (FRK) holds. In the following, we thus assume again that $\sigma$ is non-constant, i.e. not a polynomial of degree less than $\max\{1, p-1\}$. Since the distribution of the $\boldsymbol{x}_i^{(0)} := \boldsymbol{x}_i$ is non-atomic, they are distinct almost surely. By Theorem H.9, $\boldsymbol{x}_1^{(L)}, \ldots, \boldsymbol{x}_p^{(L)}$ are linearly independent almost surely over $\boldsymbol{\theta}$, which proves (FRK) for $n = p$ almost surely over $\boldsymbol{\theta}$. □

**Remark H.16** (Generalizations). The proof technique used in Theorem 10 is quite robust and can be further generalized. For example, it is easy to incorporate different activation functions for different neurons, and in all layers but the last layer, the activation functions only need to be analytic and non-constant, as required by the corresponding propagation lemmas. It is also possible to treat fixed but nonzero biases using a combination of Lemma H.11 (b) and using shifted activation functions $\tilde{\sigma}_i(x) = \sigma(x + b_i)$ in Lemma H.15. Also, the propagation lemmas, which are used for all layers except the last one, can be easily extended to DenseNet-like structures where the input of a layer is concatenated to the output. ◀

**Remark H.17** (Necessity of the assumptions). For analytic $\sigma$, the assumptions on not being a too simple polynomial in Theorem 10 are necessary. For this, consider the case with $L = 1$ layer and $d = 1$.

(a) Assume that $\sigma$ is a polynomial with less than $p$ nonzero coefficients, i.e. $\sigma(x) = \sum_{k \in K} a_k x^k$ for $|K| \leq p - 1$. For arbitrary weights $\boldsymbol{w} \in \mathbb{R}^{p \times 1}$ and data points $x_1, \ldots, x_p \in \mathbb{R}$, we obtain the feature matrix $\phi_{\boldsymbol{w}}(\boldsymbol{X}) \in \mathbb{R}^{p \times p}$ with

$$
\phi_{\boldsymbol{w}}(\boldsymbol{X})_{ij} = \sigma(w_j x_i) = \sum_{k \in K} a_k w_j^k x_i^k \,,
$$

which means that $\phi_{\boldsymbol{w}}(\boldsymbol{X})$ is the sum of the $|K| \leq p - 1$ matrices $(a_k w_j^k x_i^k)_{i,j \in [p]}$, which have at most rank 1. Hence, $\phi_{\boldsymbol{w}}(\boldsymbol{X})$ has at most rank $p - 1$ and is therefore not invertible.

(b) Assume that $\sigma$ is a polynomial of degree less than $p - 1$, i.e. $\sigma = \sum_{k=0}^{p-2} a_k x^k$. For arbitrary weights $\boldsymbol{w} \in \mathbb{R}^{p \times 1}$, biases $\boldsymbol{b} \in \mathbb{R}^p$ and data points $x_1, \ldots, x_p \in \mathbb{R}$, we obtain the feature matrix $\phi_{\boldsymbol{w}, \boldsymbol{b}}(\boldsymbol{X}) \in \mathbb{R}^{p \times p}$ with

$$
\begin{aligned}
\phi_{\boldsymbol{w}, \boldsymbol{b}}(\boldsymbol{X})_{ij} = \sigma(w_j x_i + b_j) &= \sum_{k=0}^{p-2} a_k (w_j x_i + b_j)^k \\
&= \sum_{k=0}^{p-2} \sum_{l=0}^{k} a_k \binom{k}{l} b_j^{k-l} w_j^l x_i^l \\
&= \sum_{l=0}^{p-2} \left( \sum_{k=l}^{p-2} a_k \binom{k}{l} b_j^{k-l} w_j^l \right) x_i^l
\end{aligned}
$$

$$= \sum_{l=0}^{p-2} u_j^{(l)} x_i^l \ ,$$

where $u_j^{(l)} \left( \sum_{k=l}^{p-2} a_k \binom{k}{l} b_j^{k-l} w_j^l \right)$ does not depend on $i$. Hence, $\phi_{\boldsymbol{w},\boldsymbol{b}}(\boldsymbol{X})$ is the sum of the $p-1$ matrices $(u_j^{(l)} x_i^l)_{i,j\in[p]}$, each of which has rank at most 1. Hence, $\phi_{\boldsymbol{w},\boldsymbol{b}}(\boldsymbol{X})$ has at most rank $p-1$ and is therefore not invertible. ◄

## I   RANDOM FOURIER FEATURES

In a celebrated paper, Rahimi & Recht (2008) propose to approximate a shift-invariant positive definite kernel $k(\boldsymbol{x}, \boldsymbol{x}') = \overline{k}(\boldsymbol{x} - \boldsymbol{x}')$ with a potentially infinite-dimensional feature map by a random finite-dimensional feature map, yielding so-called *random Fourier features*. If $\overline{k}$ is (up to scaling) the Fourier transform of a probability distribution $P_k$ on $\mathbb{R}^d$, two versions of random Fourier features are proposed:

(1) One version uses $\phi_{\boldsymbol{W},\boldsymbol{b}}(\boldsymbol{x}) = \sqrt{2}\cos(\boldsymbol{W}\boldsymbol{x} + \boldsymbol{b})$, where the rows of $\boldsymbol{W} \in \mathbb{R}^{p \times d}$ are independently sampled from $P_k$ and the entries of $\boldsymbol{b} \in \mathbb{R}^p$ are independently sampled from the uniform distribution on $[0, 2\pi]$. This feature map is covered by Theorem 10 and hence, if $P_k$ has a Lebesgue density and $\boldsymbol{x}$ is nonatomic, (FRK) is satisfied for all $n$. For example, if $k$ is a Gaussian kernel, $P_k$ is a Gaussian distribution and therefore has a Lebesgue density.

(2) The other version uses

$$\phi_{\boldsymbol{W}}(\boldsymbol{x}) = \begin{pmatrix} \sin(\boldsymbol{W}\boldsymbol{x}) \\ \cos(\boldsymbol{W}\boldsymbol{x}) \end{pmatrix}$$

with the same distribution over $\boldsymbol{W}$. It is not covered by Theorem 10 because of the different "activation functions" and the "weight sharing" between these activation functions. In the following proposition, we show that the proof of Theorem 10 can be adjusted to this setting and the conclusions still hold.

**Proposition I.1.** *For $\boldsymbol{x} \in \mathbb{R}^d$, $\boldsymbol{W} \in \mathbb{R}^{q \times d}$ and $p := 2q$, define*

$$\phi_{\boldsymbol{W}}(\boldsymbol{x}) := \begin{pmatrix} \sin(\boldsymbol{W}\boldsymbol{x}) \\ \cos(\boldsymbol{W}\boldsymbol{x}) \end{pmatrix} \in \mathbb{R}^p \ .$$

*If $\boldsymbol{W}$ has a Lebesgue density and $\boldsymbol{x}$ is nonatomic, then (FRK) holds for all $n$ almost surely over $\boldsymbol{W}$.*

*Proof.* **Step 1: Reduction.** According to Proposition 6, it suffices to consider the case $n = p$. By Lemma H.14, it is then sufficient to prove that the function

$$f : \mathbb{R}^q \to \mathbb{R}^{2q}, \boldsymbol{x} \mapsto (\sin(\boldsymbol{x}), \cos(\boldsymbol{x}))$$

is non-degenerate in the sense of Definition H.13.

**Step 2: Condition on the $x_i$.** We will proceed similar to Lemma H.15. Let $x_1, \ldots, x_p$ be independent scalar nonatomic random variables. For $i \in [q]$, choose $k_i := 2i - 1$ and $k_{q+i} := 2i - 2$. Then, $k_1, \ldots, k_p$ are distinct non-negative integers, and by Lemma H.12 and the Leibniz formula for the determinant, we have

$$D_x := \sum_{\pi \in S_p} \text{sgn}(\pi) x_{\pi(1)}^{k_1} \cdot \ldots \cdot x_{\pi(p)}^{k_p} = \det \begin{pmatrix} x_1^{k_1} & \cdots & x_p^{k_1} \\ \vdots & \ddots & \vdots \\ x_1^{k_p} & \cdots & x_p^{k_p} \end{pmatrix} \neq 0$$

with probability one.

**Step 3: Non-degeneracy.** Now, suppose that we are indeed in the case where $D_x \neq 0$. Take the power series of sin and cos as

$$\sin(x) = \sum_{k=0}^{\infty} (-1)^k \frac{x^{2k+1}}{(2k+1)!} =: \sum_{k=0}^{\infty} a_k x^k$$

$$\cos(x) = \sum_{k=0}^{\infty} (-1)^k \frac{x^{2k}}{(2k)!} =: \sum_{k=0}^{\infty} b_k x^k .$$

Similar to the proof of Lemma H.15, we can compute

$$
\begin{aligned}
g(\boldsymbol{w}) &:= \det \begin{pmatrix} f(\boldsymbol{w}x_1)^\top \\ \vdots \\ f(\boldsymbol{w}x_p)^\top \end{pmatrix} \\
&= \sum_{k_1,\ldots,k_{2q} \geq 0} \sum_{\pi \in S_{2q}} \mathrm{sgn}(\pi) a_{k_1} \cdots a_{k_q} b_{k_{q+1}} \cdots b_{k_{2q}} w_1^{k_1+k_{q+1}} \cdots w_q^{k_q+k_{2q}} x_{\pi(1)}^{k_1} \cdots x_{\pi(2q)}^{k_{2q}} \\
&= \sum_{k_1,\ldots,k_{2q} \geq 0} a_{k_1} \cdots a_{k_q} b_{k_{q+1}} \cdots b_{k_{2q}} \left( \sum_{\pi \in S_{2q}} x_{\pi(1)}^{k_1} \cdots x_{\pi(2q)}^{k_{2q}} \right) w_1^{k_1+k_{q+1}} \cdots w_q^{k_q+k_{2q}} \quad (13)
\end{aligned}
$$

Define the set $K := \{(k_1, \ldots, k_{2q}) \in \mathbb{N}_0^{2q} \mid \text{for all } i \in [q], k_i + k_{q+i} = 2i-1\}$. Then, the coefficient $c$ of the monomial $w_1^1 w_2^3 \cdot \ldots \cdot w_q^{2q-1}$ in (13) can be written as

$$c := \sum_{(k_1,\ldots,k_{2q}) \in K} c_{k_1,\ldots,k_{2q}} ,$$

$$c_{k_1,\ldots,k_{2q}} := a_{k_1} \cdots a_{k_q} b_{k_{q+1}} \cdots b_{k_{2q}} \left( \sum_{\pi \in S_{2q}} x_{\pi(1)}^{k_1} \cdots x_{\pi(2q)}^{k_{2q}} \right) .$$

Now, consider $(k_1, \ldots, k_{2q}) \in K$. Note that $a_k \neq 0$ iff $k$ is odd and $b_k \neq 0$ iff $k$ is even. For the choice $k_i := 2i - 1, k_{q+i} := 2i - 2$ for $i \in [q]$ from Step 2, we have $(k_1, \ldots, k_{2q}) \in K$ and

$$c_{k_1,\ldots,k_{2q}} = a_{k_1} \cdots a_{k_q} b_{k_{q+1}} \cdots b_{k_{2q}} D_x \neq 0 .$$

In the following, we will show that $c_{k_1,\ldots,k_{2q}} = 0$ for all other $(k_1, \ldots, k_{2q}) \in K$, which implies $c \neq 0$ and therefore yields that $g$ is not the zero function, which is what we want to show. If $k_i = k_j$ for some $i \neq j$, we have

$$
\sum_{\pi \in S_{2q}} x_{\pi(1)}^{k_1} \cdots x_{\pi(2q)}^{k_{2q}} = \det \begin{pmatrix} x_1^{k_1} & \cdots & x_p^{k_1} \\ \vdots & \ddots & \vdots \\ x_1^{k_p} & \cdots & x_p^{k_p} \end{pmatrix} = 0 ,
$$

since the $i$-th and $j$-th rows of the matrix are equal, and hence $c_{k_1,\ldots,k_{2q}} = 0$. Now, suppose that $(k_1, \ldots, k_{2q}) \in K$ with $c_{k_1,\ldots,k_{2q}} \neq 0$. By induction, it is easy to show that $\{k_i, k_{q+i}\} = \{2i - 1, 2i - 2\}$ for all $i \in [q]$. But since

$$a_{k_1} \cdots a_{k_q} b_{k_{q+1}} \cdots b_{k_{2q}} \neq 0$$

and $a_k = 0$ for even $k$, we need to have $k_i = 2i - 1, k_{q+i} = 2i - 2$ for all $i \in [q]$. This shows the claim. $\qquad\square$

## J   PROOFS FOR SECTION 6

In this section, we first prove the analytic formulas from Section 6 before discussing the case of low input dimension $d$.

**Theorem 11.** *Let $P_Z = \mathcal{U}(\mathbb{S}^{p-1})$. Then, $P_Z$ satisfies the assumptions (MOM), (COV) and (FRK) for all $n$ with $\boldsymbol{\Sigma} = \frac{1}{p}\boldsymbol{I}_p$. Moreover, for $n \geq p = 1$ or $p \geq n \geq 1$, we can compute*

$$
\mathbb{E}_{\boldsymbol{Z}} \,\mathrm{tr}((\boldsymbol{Z}^+)^\top \boldsymbol{\Sigma} \boldsymbol{Z}^+) = \begin{cases} \frac{1}{n} & \text{if } n \geq p = 1, \\ \frac{1}{p} & \text{if } p \geq n = 1, \\ \infty & \text{if } 2 \leq n \leq p \leq n + 1, \\ \frac{n}{p-1-n} \cdot \frac{p-2}{p} & \text{if } 2 \leq n + 2 \leq p. \end{cases}
$$

*Proof.* **Step 1: Verify (MOM).** Let $\boldsymbol{x}_i \sim \mathcal{N}(0, \boldsymbol{I}_p)$ for $i \in [n]$ be independent. Then, $\boldsymbol{z}_i := \frac{\boldsymbol{x}_i}{\|\boldsymbol{x}_i\|_2} \sim \mathcal{U}(\mathbb{S}^{p-1})$. Since $\mathbb{E}\|\boldsymbol{z}_i\|_2^2 = \mathbb{E}1 = 1$, (MOM) is satisfied and thus, $\boldsymbol{\Sigma}$ is well-defined.

**Step 2: Compute $\boldsymbol{\Sigma}$.** We can use rotational invariance as follows: Let $\boldsymbol{V} \in \mathbb{R}^{p \times p}$ be an arbitrary fixed orthogonal matrix. Then, $\boldsymbol{V}\boldsymbol{x}_i \sim \mathcal{N}(0, \boldsymbol{V}\boldsymbol{V}^\top) = \mathcal{N}(0, \boldsymbol{I}_p)$ and hence $\boldsymbol{V}\boldsymbol{z}_i = \frac{\boldsymbol{V}\boldsymbol{x}_i}{\|\boldsymbol{x}_i\|_2} = \frac{\boldsymbol{V}\boldsymbol{x}_i}{\|\boldsymbol{V}\boldsymbol{x}_i\|_2} \sim \mathcal{U}(\mathbb{S}^{p-1})$. Therefore,

$$\boldsymbol{\Sigma} = \mathbb{E}\boldsymbol{z}_i\boldsymbol{z}_i^\top = \mathbb{E}\boldsymbol{V}\boldsymbol{z}_i\boldsymbol{z}_i^\top\boldsymbol{V}^\top = \boldsymbol{V}\boldsymbol{\Sigma}\boldsymbol{V}^\top \ . \tag{14}$$

If $0 \neq \boldsymbol{v} \in \mathbb{R}^p$ is an eigenvector of $\boldsymbol{\Sigma}$ with eigenvalue $\lambda$, then $\boldsymbol{V}\boldsymbol{v}$ must by Eq. (14) also be an eigenvector of $\boldsymbol{\Sigma}$ with eigenvalue $\lambda$. But since $\boldsymbol{V}$ is an arbitrary orthogonal matrix, this means that $\boldsymbol{V}\boldsymbol{v}$ is an arbitrary rotation of $\boldsymbol{v}$. From this it is easy to conclude that $\boldsymbol{\Sigma} = \lambda\boldsymbol{I}_p$, and from

$$p\lambda = \operatorname{tr}(\boldsymbol{\Sigma}) = \mathbb{E}\operatorname{tr}(\boldsymbol{z}_i\boldsymbol{z}_i^\top) = \mathbb{E}\boldsymbol{z}_i^\top\boldsymbol{z}_i = \mathbb{E}1 = 1 \ ,$$

it follows that $\boldsymbol{\Sigma} = \frac{1}{p}\boldsymbol{I}_p$. Hence, (COV) is satisfied and $\boldsymbol{w}_i = \sqrt{p}\boldsymbol{z}_i$.

**Step 3: Verify (FRK) for all $n$.** By Proposition 6, it is sufficient to verify (FRK) for $n = p$. Therefore, let $n = p$. It is obvious from Proposition 6 with $\phi = \operatorname{id}$ that $\mathcal{N}(0, \boldsymbol{I}_p)$ satisfies (FRK). Hence, $\boldsymbol{X}$ almost surely has full rank. But then, since $\|\boldsymbol{x}_i\|_2 > 0$ almost surely,

$$\boldsymbol{Z} = \operatorname{diag}\left(\frac{1}{\|\boldsymbol{x}_1\|}, \ldots, \frac{1}{\|\boldsymbol{x}_n\|}\right)\boldsymbol{X}$$

almost surely has full rank as well, which proves (FRK).

**Step 4.1: Computation for $n \geq p = 1$.** In the underparameterized case $n \geq p = 1$, we can compute

$$\mathbb{E}_{\boldsymbol{Z}}\operatorname{tr}((\boldsymbol{Z}^+)^\top\boldsymbol{\Sigma}\boldsymbol{Z}^+) = \mathbb{E}_{\boldsymbol{Z}}\operatorname{tr}((\boldsymbol{W}^\top\boldsymbol{W})^{-1}) = \mathbb{E}_{\boldsymbol{Z}}\frac{1}{\sum_{i=1}^n w_i^2} = \mathbb{E}_{\boldsymbol{Z}}\frac{1}{n} = \frac{1}{n} \ .$$

**Step 4.2: Computation for $p \geq n = 1$.** In the overparameterized case $p \geq n = 1$, we can compute

$$\mathbb{E}_{\boldsymbol{Z}}\operatorname{tr}((\boldsymbol{Z}^+)^\top\boldsymbol{\Sigma}\boldsymbol{Z}^+) = \mathbb{E}_{\boldsymbol{Z}}\operatorname{tr}((\boldsymbol{W}\boldsymbol{W}^\top)^{-1}) = \mathbb{E}_{\boldsymbol{Z}}\frac{1}{\boldsymbol{w}_1^\top\boldsymbol{w}_1} = \mathbb{E}_{\boldsymbol{Z}}\frac{1}{p} = \frac{1}{p} \ ,$$

where we used that since $\boldsymbol{\Sigma} = \frac{1}{p}\boldsymbol{I}_p$, $\boldsymbol{w}_1^\top\boldsymbol{w}_1 = \|\boldsymbol{w}_1\|_2^2 = \|\sqrt{p}\boldsymbol{z}_1\|_2^2 = p$.

**Step 4.3: Computation for $p \geq n \geq 2$.** Now, let $p \geq n \geq 2$. Since $\boldsymbol{\Sigma} = \frac{1}{p}\boldsymbol{I}_p$, we have

$$\mathbb{E}_{\boldsymbol{Z}}\operatorname{tr}((\boldsymbol{Z}^+)^\top\boldsymbol{\Sigma}\boldsymbol{Z}^+) = \mathbb{E}_{\boldsymbol{Z}}\operatorname{tr}((\boldsymbol{W}\boldsymbol{W}^\top)^{-1})$$

by Theorem 3. Using that the $\boldsymbol{w}_i$ are i.i.d., we obtain from Lemma G.2 that $\mathbb{E}((\boldsymbol{W}\boldsymbol{W}^\top)^{-1}) = n\mathbb{E}\operatorname{dist}(\boldsymbol{w}_1, \mathcal{W}_{-1})^{-2}$, where $\mathcal{W}_{-1}$ is the space spanned by $\boldsymbol{w}_2, \ldots, \boldsymbol{w}_n$. Define the subspace $\mathcal{U}_n := \{\boldsymbol{z} \in \mathbb{R}^p \mid z_n = z_{n+1} = \ldots = z_p = 0\}$. By (FRK), we almost surely have $\dim(\mathcal{W}_{-1}) = n - 1$. Thus, there is an orthogonal matrix $\boldsymbol{U}_{-1}$ depending only on $\mathcal{W}_{-1}$ that rotates $\mathcal{W}_{-1}$ to $\mathcal{U}_n$:

$$\mathcal{U}_n = \boldsymbol{U}_{-1}\mathcal{W}_{-1} \ .$$

Because $\boldsymbol{w}_1$ is stochastically independent from $\mathcal{W}_{-1}$ and $\boldsymbol{U}_{-1}$ and its distribution is rotationally symmetric, we have the distributional equivalence (using the $\boldsymbol{z}_i$ and $\boldsymbol{x}_i$ from Step 1)

$$\operatorname{dist}(\boldsymbol{w}_1, \mathcal{W}_{-1})^2 = \operatorname{dist}(\boldsymbol{U}_{-1}\boldsymbol{w}_1, \boldsymbol{U}_{-1}\mathcal{W}_{-1})^2 \overset{\text{distrib.}}{=} \operatorname{dist}(\boldsymbol{w}_1, \mathcal{U}_n)^2 = p(z_{1,n}^2 + \ldots + z_{1,p}^2)$$

$$= p\frac{x_{1,n}^2 + \ldots + x_{1,p}^2}{\|\boldsymbol{x}_1\|_2^2} = p\frac{A}{A + B} \ ,$$

where $A := x_{1,n}^2 + x_{1,n+1}^2 + \ldots + x_{1,p}^2$ has a $\chi_{p+1-n}^2$ distribution and $B := x_{1,1}^2 + \ldots + x_{1,n-1}^2$ has a $\chi_{n-1}^2$ distribution. Hence,

$$\mathbb{E}((\boldsymbol{W}\boldsymbol{W}^\top)^{-1}) = n\mathbb{E}\operatorname{dist}(\boldsymbol{w}_1, \mathcal{W}_{-1})^{-2} = \frac{n}{p}\left(1 + \mathbb{E}\frac{B}{A}\right) \ .$$

Since $p \geq n \geq 2$, $n - 1$ and $p + 1 - n$ are positive. Since $A$ and $B$ are independent, $\frac{B/(n-1)}{A/(p+1-n)}$ follows a Variance-Ratio $F$-distribution with parameters $n - 1$ and $p + 1 - n$, whose mean is known (see e.g. Chapter 20 in Forbes et al., 2011):

$$\mathbb{E}\frac{B}{A} = \frac{n-1}{p+1-n}\mathbb{E}\frac{B/(n-1)}{A/(p+1-n)} = \begin{cases} \infty & \text{, if } p+1-n \leq 2, \\ \frac{n-1}{p+1-n}\frac{p+1-n}{(p+1-n)-2} = \frac{n-1}{p-1-n} & \text{, if } p+1-n > 2. \end{cases} \quad (15)$$

The infinite expectation for $p + 1 - n \leq 2$ is not explicitly specified in Forbes et al. (2011), but it is easy to obtain from the p.d.f. of the $F$-distribution: The p.d.f. $f(x)$ of the $F(a, b)$-distribution for $x \geq 0$ is

$$f(x) = C_{a,b}\frac{x^{(a-2)/2}}{(1+(a/b)x)^{(a+b)/2}} = \Theta(x^{-b/2-1}) \qquad (x \to \infty)$$

for some constant $C_{a,b}$ (cf. Chapter 20 in Forbes et al., 2011), and the expected value is therefore

$$\int_0^\infty x f(x)\,\mathrm{d}x = \int_0^\infty \Theta(x^{-b/2})\,\mathrm{d}x\ ,$$

which is infinite for $p+1-n = b \leq 2$. For $n \in \{p, p-1\}$, we therefore obtain $\mathbb{E}((\boldsymbol{W}\boldsymbol{W}^\top)^{-1}) = \infty$. For $n \leq p - 2$, we compute

$$\mathbb{E}((\boldsymbol{W}\boldsymbol{W}^\top)^{-1}) = \frac{n}{p}\left(1 + \frac{n-1}{p-1-n}\right) = \frac{n}{p}\cdot\frac{p-2}{p-1-n}\ . \qquad \square$$

In the following, we will prove Theorem 12 using the same proof idea as for Theorem 11. The formulas in Theorem 12 have in principle already been computed by Breiman & Freedman (1983) for $p \leq n - 2$ and by Belkin et al. (2019b) for general $p$. However, our proof circumvents a technical problem in the proof of Belkin et al. (2019b): Consider for example the case $p \leq n$. Belkin et al. (2019b) mention that $(\boldsymbol{W}^\top\boldsymbol{W})^{-1}$ has an inverse Wishart distribution, which for $p \leq n - 2$ has expectation $\frac{1}{n-1-p}\boldsymbol{I}_p$, and then use $\mathbb{E}\operatorname{tr}((\boldsymbol{W}^\top\boldsymbol{W})^{-1}) = \operatorname{tr}(\mathbb{E}(\boldsymbol{W}^\top\boldsymbol{W})^{-1})$. However, for $p \geq n - 1$, the latter expectation is not specified in common literature[7] on the inverse Wishart distribution (Mardia et al., 1979; Press, 2005; von Rosen, 1988), presumably because it is $\infty$ for diagonal elements but is not well-defined for off-diagonal matrix elements.

**Theorem 12.** *Let $P_Z = \mathcal{N}(0, \boldsymbol{I}_p)$. Then, $P_Z$ satisfies the assumptions (MOM), (COV) and (FRK) for all $n$ with $\boldsymbol{\Sigma} = \boldsymbol{I}_p$. Moreover, for $n, p \geq 1$,*

$$\mathbb{E}_{\boldsymbol{Z}}\operatorname{tr}((\boldsymbol{Z}^+)^\top\boldsymbol{\Sigma}\boldsymbol{Z}^+) = \begin{cases} \frac{n}{p-1-n} & \text{if } p \geq n+2, \\ \infty & \text{if } p \in \{n-1, n, n+1\}, \\ \frac{p}{n-1-p} & \text{if } p \leq n-2. \end{cases}$$

*Proof.* **Step 1: Assumptions.** Verifying (MOM), (COV) and $\boldsymbol{\Sigma} = \boldsymbol{I}_p$ is trivial and (FRK) for all $n$ follows from Proposition 6 with $\boldsymbol{x} = \boldsymbol{z}$ and $\phi = \mathrm{id}$.

**Step 2: Overparameterized case.** For the expectation, we first follow Step 4.3 in the proof of Theorem 11 in the overparameterized case $p \geq n \geq 1$, the main difference being that instead of $\boldsymbol{w}_i = \sqrt{p}\frac{\boldsymbol{x}_i}{\|\boldsymbol{x}_i\|_2}$, we now have $\boldsymbol{w}_i = \boldsymbol{x}_i$, which translates to the simpler equation

$$\operatorname{dist}(\boldsymbol{w}_1, \mathcal{W}_{-1})^2 \overset{\text{distrib.}}{=} A$$

with $A \sim \chi^2_{p+1-n}$. Let $B \sim \chi^2_1$ be independent of $A$, then we can compute similar to Eq. (15)

$$\mathbb{E}\operatorname{tr}((\boldsymbol{W}\boldsymbol{W}^\top)^{-1}) = n\mathbb{E}\operatorname{dist}(\boldsymbol{w}_1, \mathcal{W}_{-1})^{-2} = n\mathbb{E}\frac{1}{A} = n\left(\mathbb{E}B\right)\left(\mathbb{E}\frac{1}{A}\right) = n\mathbb{E}\frac{B}{A}$$

$$= \frac{n}{p+1-n}\mathbb{E}\frac{B/1}{A/(p+1-n)}$$

$$= \begin{cases} \infty & \text{if } p+1-n \leq 2, \\ \frac{n}{p+1-n}\frac{p+1-n}{(p+1-n)-2} = \frac{n}{p-1-n} & \text{if } p+1-n > 2. \end{cases}$$

---

[7]Belkin et al. (2019b) do not cite any source on the inverse Wishart distribution.

This proves the over-parameterized case.

**Step 3: Underparameterized case.** Since the rows $\boldsymbol{w}_i$ of $\boldsymbol{W} \in \mathbb{R}^{n \times p}$ are independent and follow a $\mathcal{N}(0, \boldsymbol{I}_p)$ distribution, the rows of $\boldsymbol{W}^\top \in \mathbb{R}^{p \times n}$ are independent and follow a $\mathcal{N}(0, \boldsymbol{I}_n)$ distribution. Therefore, the underparameterized case $p \leq n$ follows from the overparameterized case $n \leq p$ by switching the roles of $n$ and $p$. $\qquad \square$

**Remark J.1.** An alternative (and presumably similar) way to prove Theorem 12 is to use that the diagonal elements of a matrix with an inverse Wishart distribution follow an inverse Gamma distribution as specified in Example 5.2.2 in Press (2005). $\qquad \blacktriangleleft$

The next proposition shows that a small input dimension $d$ does not necessarily provide a limitation:

**Proposition J.2.** *Let $p, d \geq 1$. Then, there exists a probability distribution $P_X$ on $\mathbb{R}^d$ (with bounded support) and a continuous feature map $\phi : \mathbb{R}^d \to \mathbb{R}^p$ such that for $\boldsymbol{x} \sim P_X$, $\phi(\boldsymbol{x}) \sim \mathcal{U}(\mathbb{S}^{p-1})$.*

*Proof.* For $p = 1$, the result is trivial, we will therefore assume $p \geq 2$. We will prove the result for any $d$ by a reduction to the case $d = 1$, although substantially simpler constructions are possible for $d \geq p - 1$. First, introduce the spaces

$$
\begin{aligned}
\mathbb{S}_+^{p-1} &:= \{\boldsymbol{z} \in \mathbb{S}^{p-1} \mid z_p \geq 0\} \subseteq \mathbb{R}^p \\
\mathbb{B}^{p-1} &:= \{\boldsymbol{z} \in \mathbb{R}^{p-1} \mid \|\boldsymbol{z}\|_2 \leq 1\} \\
\mathcal{X} &:= [0, 3] \times \{0\}^{p-1} \subseteq \mathbb{R}^p .
\end{aligned}
$$

**Step 1: Space-filling curve on the sphere.** In this step, we show that there exists a continuous surjective map $\phi : \mathcal{X} \to \mathbb{S}^{p-1}$. First of all, let $f_1 : [0, 1] \to [0, 1]^{p-1}$ be continuous and surjective, e.g. a Hilbert or Peano curve (see e.g. Sagan, 2012). We define the following maps:

$$
f_2 : [0, 1]^{p-1} \to \mathbb{B}^{p-1}, \boldsymbol{u} \mapsto \begin{cases} \boldsymbol{0} & \text{if } \boldsymbol{u} = \boldsymbol{0} \\ \frac{\|\boldsymbol{u}\|_\infty}{\|\boldsymbol{u}\|_2} \boldsymbol{u} & \text{if } \boldsymbol{u} \neq \boldsymbol{0} , \end{cases}
$$

$$
f_3 : \mathbb{B}^{p-1} \to \mathbb{S}_+^{p-1}, \boldsymbol{v} \mapsto \left( \boldsymbol{v}, \sqrt{1 - \|\boldsymbol{v}\|_2^2} \right) .
$$

It is not hard to verify that $f_2$ and $f_3$ are continuous and surjective as well. For example, $f_2$ is continuous in $\boldsymbol{0}$ since $\|\boldsymbol{u}\|_\infty \leq \|\boldsymbol{u}\|_2$ for all $\boldsymbol{u} \in \mathbb{R}^{p-1}$. Thus, the map $f := f_3 \circ f_2 \circ f_1 : [0, 1] \to \mathbb{S}_+^{p-1}$ is continuous and surjective. Define the map

$$
\tau : \mathbb{R}^p \to \mathbb{R}^p, \boldsymbol{z} \mapsto (z_1, \ldots, z_{p-1}, -z_p) .
$$

By the previous considerations, it is not hard to verify that the map

$$
g : [0, 3] \to \mathbb{S}^{p-1}, x \mapsto \begin{cases} f(x) & \text{if } x \in [0, 1] \\ \frac{x-1}{2-1} f(1) + \left( 1 - \frac{x-1}{2-1} \right) \tau(f(1)) & \text{if } x \in [1, 2] \\ \tau(f(3 - x)) & \text{if } x \in [2, 3] \end{cases}
$$

is continuous and surjective as well. We can therefore define the continuous and surjective map $\phi : \mathcal{X} \to \mathbb{S}^{p-1}, \boldsymbol{x} \mapsto g(x_1)$.

**Step 2: Existence of a pull-back measure.** We consider the Borel $\sigma$-algebras $\mathcal{B}(\mathcal{X}), \mathcal{B}(\mathbb{S}^{p-1})$ on $\mathcal{X}$ and $\mathbb{S}^{p-1}$. The uniform distribution $P_Z = \mathcal{U}(\mathbb{S}^{p-1})$ on the sphere is defined with respect to $\mathcal{B}(\mathbb{S}^{p-1})$ and is therefore a Borel measure. Since $\phi$ is continuous, it is Borel measurable. Moreover, since $\mathcal{X}$ and $\mathbb{S}^{p-1}$ are complete separable metric spaces, they are also Souslin spaces, cf. Section 6.6 in Bogachev (2007). Since $\phi$ is surjective, Theorem 9.1.5 in Bogachev (2007) guarantees the existence of a measure $P_X$ such that if $\boldsymbol{x} \sim P_X$, then $\phi(\boldsymbol{x}) \sim \mathcal{U}(\mathbb{S}^{p-1})$. Since $P_X(\mathcal{X}) = P_Z(\mathbb{S}^{p-1}) = 1$, $P_X$ is a probability measure.

**Step 3: Continuation.** We can arbitrarily extend the mapping $\phi : \mathcal{X} \to \mathbb{S}^{p-1}$ to a continuous mapping $\phi : \mathbb{R}^d \to \mathbb{R}^p$. Moreover, the domain $\mathcal{X}$ of $P_X$ can be extended to $\mathbb{R}^d$ via $P_X(A) := P_X(A \cap \mathcal{X})$, the support of $P_X$ is still bounded, and we still have $\phi(\boldsymbol{x}) \sim \mathcal{U}(\mathbb{S}^{p-1})$ if $\boldsymbol{x} \sim P_X$. $\qquad \square$

**Remark J.3.** The proof of Proposition J.2 could be slightly shorter if we required $\phi(\boldsymbol{x}) \sim \mathcal{U}(\mathbb{S}_+^{p-1})$ instead of $\phi(\boldsymbol{x}) \sim \mathcal{U}(\mathbb{S}^{p-1})$. This would be of similar interest since the uniform distribution $\mathcal{U}(\mathbb{S}_+^{p-1})$ on the "half-sphere" leads to the same $\mathbb{E}_{\boldsymbol{Z}}\operatorname{tr}((\boldsymbol{Z}^+)^\top\boldsymbol{\Sigma}(\boldsymbol{Z}^+))$ as the uniform distribution $\mathcal{U}(\mathbb{S}^{p-1})$ on the full sphere: If $\boldsymbol{z}_i \sim \mathcal{U}(\mathbb{S}_+^{p-1})$ and $\varepsilon_i \sim \mathcal{U}(\{-1,1\})$ are stochastically independent, then $\tilde{\boldsymbol{z}}_i := \varepsilon_i \boldsymbol{z}_i \sim \mathcal{U}(\mathbb{S}^{p-1})$. Therefore, $\boldsymbol{\Sigma} = \tilde{\boldsymbol{\Sigma}}$, $\tilde{\boldsymbol{Z}} = \operatorname{diag}(\varepsilon_1,\dots,\varepsilon_n)\boldsymbol{Z}$, and if $\boldsymbol{U}\boldsymbol{D}\boldsymbol{V}^\top$ is a SVD of $\boldsymbol{Z}$, then $(\operatorname{diag}(\varepsilon_1,\dots,\varepsilon_n)\boldsymbol{U})\boldsymbol{D}\boldsymbol{V}^\top$ is a SVD of $\tilde{\boldsymbol{Z}}$. Therefore, $\boldsymbol{Z}$ and $\tilde{\boldsymbol{Z}}$ have the same singular values, hence $\boldsymbol{W}$ and $\tilde{\boldsymbol{W}}$ have the same singular values, hence $\operatorname{tr}((\boldsymbol{W}\boldsymbol{W}^\top)^{-1}) = \operatorname{tr}((\tilde{\boldsymbol{W}}\tilde{\boldsymbol{W}}^\top)^{-1})$ for $p \geq n$ and $\operatorname{tr}((\boldsymbol{W}^\top\boldsymbol{W})^{-1}) = \operatorname{tr}((\tilde{\boldsymbol{W}}^\top\tilde{\boldsymbol{W}})^{-1})$ for $p \leq n$. ◀

**Remark J.4.** One might ask whether it is possible in Proposition J.2 to choose $P_X$ as a "nice" distribution, like a uniform distribution on a cube or a Gaussian distribution. The answer to this question is affirmative if there exists an area-preserving space-filling curve $\phi : [0, \operatorname{volume}(\mathbb{S}^{p-1})] \to \mathbb{S}^{p-1}$. For $p = 3$, such a construction is informally described by Purser et al. (2009) and it seems plausible that such a construction is possible for all $p$. ◀

## K   RELATION TO RIDGELESS KERNEL REGRESSION

In this section, we want to discuss the relation between this paper and recent work on ridgeless kernel regression. To this end, we need to introduce some terminology on representations of kernels with finite-dimensional feature maps.

**Definition K.1.** Let $k : \mathbb{R}^d \times \mathbb{R}^d \to \mathbb{R}$ be a kernel, let $p$ be an integer with $1 \leq p < \infty$ and let $\phi : \mathbb{R}^d \to \mathbb{R}^p$ be a (measurable) function. Then, $(p, \phi)$ is called a $P_X$-representation of $k$ if

- $k(\boldsymbol{x}, \tilde{\boldsymbol{x}}) = \phi(\boldsymbol{x})^\top\phi(\tilde{\boldsymbol{x}})$ almost surely for independent $\boldsymbol{x}, \tilde{\boldsymbol{x}} \sim P_X$, and
- $k(\boldsymbol{x}, \boldsymbol{x}) = \phi(\boldsymbol{x})^\top\phi(\boldsymbol{x})$ almost surely for $\boldsymbol{x} \sim P_X$.

If $k$ has a $P_X$-representation, then we define

$$p_k := \min_{(p,\phi) \text{ is a } P_X\text{-representation of } k} p \,,$$

i.e. $p_k$ is the smallest $p$ for which a $P_X$-representation exists. ◀

Usually, $p_k$ corresponds to the dimension of the RKHS associated with the restriction of $k$ to the support of $P_X$, but since the feature map $\phi$ only needs to represent the kernel $P_X$-almost surely, $p_k$ may be smaller for pathological kernels. The following lemma states that Theorem 3, if applicable, should be applied to ridgeless kernel regression with $p = p_k$:

**Lemma K.2.** *Let $k$ be a kernel on $\mathbb{R}^d$ with $P_X(k(\boldsymbol{x}, \boldsymbol{x}) \neq 0) > 0$. Let $(p, \phi)$ be a $P_X$-representation of $k$.*

- *(a) Then, (COV) in Theorem 3 is satisfied iff $p = p_k$.*
- *(b) The assumptions of Theorem 3 are satisfied for a $P_X$-representation $(p_k, \phi)$ of $k$ if and only if they are satisfied for all such representations.*
- *(c) If the assumptions of Theorem 3 are satisfied for any $P_X$-representation of $k$, then the lower bound from Theorem 3 also holds for ridgeless kernel regression with $p = p_k$.*

*Proof.* In the notation of Section 3, the definition of $P_X$-representation implies that $k(\boldsymbol{X}, \boldsymbol{X}) = \phi(\boldsymbol{X})\phi(\boldsymbol{X})^\top$ and $k(\boldsymbol{x}, \boldsymbol{X}) = \phi(\boldsymbol{x})^\top\phi(\boldsymbol{X})^\top$ almost surely.

- (a) If (COV) is not satisfied and $p \geq 2$, it is possible to construct a $P_X$-representation with smaller $p$ using the construction from Remark 7, hence $p > p_k$. If $p = 1$, (COV) is satisfied due to the assumption on $k$.
  Conversely, assume $p > p_k$ and let $(p_k, \tilde{\phi})$ be another $P_X$-representation of $k$. Set $n = p$. Then, we almost surely have

  $$\operatorname{rank}\phi(\boldsymbol{X}) = \operatorname{rank}(\phi(\boldsymbol{X})\phi(\boldsymbol{X})^\top) = \operatorname{rank}(k(\boldsymbol{X}, \boldsymbol{X})) = \operatorname{rank}(\tilde{\phi}(\boldsymbol{X})\tilde{\phi}(\boldsymbol{X})^\top) \leq p_k$$
  $$< p \,,$$

  hence $\phi(\boldsymbol{X})$ has full rank with probability zero. Since the rows $\phi(\boldsymbol{x}_i)$ of $\phi(\boldsymbol{X})$ are i.i.d., this means that there must be a proper linear subspace $U$ of $\mathbb{R}^p$ such that $\phi(\boldsymbol{x}_i) \in U$ with probability one. But then, according to Proposition 6, (COV) is not satisfied.

(b) Let $(p_k, \phi)$ and $(p_k, \tilde{\phi})$ be two $P_X$-representations of $k$ such that $\boldsymbol{z} = \phi(\boldsymbol{x})$ satisfies the assumptions of Theorem 3. We need to show that $\tilde{\boldsymbol{z}} = \tilde{\phi}(\boldsymbol{x})$ also satisfies the assumptions of Theorem 3: First of all, (INT) and (NOI) hold since they are independent of the feature map. Moreover, (COV) holds by (a). We find that (MOM) holds due to

$$\mathbb{E}\|\tilde{\boldsymbol{z}}\|_2^2 = \mathbb{E}\tilde{\phi}(\boldsymbol{x})^\top \tilde{\phi}(\boldsymbol{x}) = \mathbb{E}k(\boldsymbol{x}, \boldsymbol{x}) = \mathbb{E}\phi(\boldsymbol{x})^\top \phi(\boldsymbol{x}) = \mathbb{E}\|\boldsymbol{z}\|_2^2 < \infty$$

and (FRK) holds since, almost surely,

$$\mathrm{rank}\,\tilde{\phi}(\boldsymbol{X}) = \mathrm{rank}(\tilde{\phi}(\boldsymbol{X})\tilde{\phi}(\boldsymbol{X})^\top) = \mathrm{rank}(k(\boldsymbol{X}, \boldsymbol{X})) = \mathrm{rank}(\phi(\boldsymbol{X})\phi(\boldsymbol{X})^\top)$$
$$= \mathrm{rank}\,\phi(\boldsymbol{X}) = \min\{n, p\}\,.$$

(c) Assume that there exists a $P_X$-representation $(p, \phi)$ of $k$ that satisfies the assumptions of Theorem 3. By (a), we have $p = p_k$. By Eq. (2) in Section 3, the ridgeless kernel regression estimator and the linear regression estimator with the feature map $\phi$ are almost surely equivalent, hence they have the same $\mathcal{E}_{\mathrm{Noise}}$. □

For kernels that cannot be represented with a finite-dimensional feature space, Theorem 3 cannot be applied. In fact, any distribution-independent lower bound for ridgeless kernel regression must be zero in this case: For example, the Kronecker delta kernel given by

$$k(\boldsymbol{x}, \tilde{\boldsymbol{x}}) = \begin{cases} 1 & \text{if } \boldsymbol{x} = \tilde{\boldsymbol{x}} \\ 0 & \text{otherwise} \end{cases}$$

yields $\mathcal{E}_{\mathrm{Noise}} = 0$ for any nonatomic input distribution $P_X$. Of course, this kernel is not well-suited for learning since the learned functions are zero almost everywhere. However, there exist results for ridgeless kernel regression with specific classes of kernels. For example, Rakhlin & Zhai (2019) show that in certain settings, ridgeless kernel regression with Laplace kernels is inconsistent because $\mathcal{E}_{\mathrm{Noise}} = \Omega(1)$ as $n \to \infty$. Note that Laplace kernels in general do not allow for finite-dimensional feature map representations.

Liang & Rakhlin (2020) derive upper bounds for a certain class of kernels and input distributions with (linearly transformed) i.i.d. components. Their analysis focuses on the high-dimensional limit $d, n \to \infty$ with $0 < c \le d/n \le C < \infty$ and ignores the "effective dimension" $p_k$ of the feature space. It appears that their analysis is not impacted by Double Descent w.r.t. $p_k$ since their assumptions on the kernel imply either $p_k = \infty$ or $p_k/n \to \infty$ as $n, d \to \infty$: In particular, they consider kernels of the form

$$k(\boldsymbol{x}, \tilde{\boldsymbol{x}}) = h\left(\frac{1}{d}\langle \boldsymbol{x}, \tilde{\boldsymbol{x}}\rangle\right)$$

for a suitable smooth function $h$ that is independent of $d$. Due to the factor $\frac{1}{d}$ and the limit $d \to \infty$, the kernel behaves essentially like a quadratic kernel

$$k(\boldsymbol{x}, \tilde{\boldsymbol{x}}) \approx a_0 + a_1 \frac{1}{d}\langle \boldsymbol{x}, \tilde{\boldsymbol{x}}\rangle + a_2\left(\frac{1}{d}\langle \boldsymbol{x}, \tilde{\boldsymbol{x}}\rangle\right)^2 =: k_{\mathrm{quad}}(\boldsymbol{x}, \tilde{\boldsymbol{x}})\,,$$

where the curvature $a_2$ should be positive in order to obtain good upper bounds on $\mathcal{E}_{\mathrm{Noise}}$ (the variance term). For $a_2, a_1, a_0 > 0$, it is possible to represent this quadratic kernel with a feature map analogous to that of the polynomial kernel in Proposition 8 with feature space dimension $p = 1 + d + \frac{d(d+1)}{2}$. An argument similar to the proof of Proposition 8 shows that this feature map satisfies $\mathrm{FRK}(p)$ if $\boldsymbol{x}$ has a Lebesgue density, hence (COV) is satisfied. By Lemma K.2 (a), we have $p_{k_{\mathrm{quad}}} = p = \Theta(d^2) = \Theta(n^2)$, which shows $p_{k_{\mathrm{quad}}}/n \to \infty$ as $n, d \to \infty$.

Liang et al. (2019) consider a similar setting with $d \sim n^\alpha, \alpha \in (0, 1)$, and find that $\mathcal{E}_{\mathrm{Noise}}$ converges to zero under suitable assumptions as $d, n \to \infty$. Again, it appears that their assumptions on the kernel imply at least a strongly overparameterized regime with $p_k = \infty$ or $p_k/n \to \infty$ for $d, n \to \infty$, where our lower bound is vacuous.

## L   NOVELTY OF THE OVERPARAMETERIZED BOUND

In their Corollary 1, Muthukumar et al. (2020) provide a lower bound in the case $p \ge n$ holding with high probability for $\boldsymbol{\varepsilon}^\top(\boldsymbol{W}\boldsymbol{W}^\top)^{-1}\boldsymbol{\varepsilon}$, where $\boldsymbol{\varepsilon} \sim \mathcal{N}(0, \boldsymbol{I}_p)$ is a noise vector independent of

$\boldsymbol{W}$. Since $\mathbb{E}\boldsymbol{\varepsilon}^{\top}(\boldsymbol{W}\boldsymbol{W}^{\top})^{-1}\boldsymbol{\varepsilon} = \mathbb{E}_{\boldsymbol{Z}}\operatorname{tr}((\boldsymbol{W}\boldsymbol{W}^{\top})^{-1}\mathbb{E}_{\boldsymbol{\varepsilon}}\boldsymbol{\varepsilon}\boldsymbol{\varepsilon}^{\top}) = \mathbb{E}_{\boldsymbol{Z}}\operatorname{tr}((\boldsymbol{W}\boldsymbol{W}^{\top})^{-1})$, their lower bound yields a lower bound for $\mathbb{E}\operatorname{tr}((\boldsymbol{W}\boldsymbol{W}^{\top})^{-1})$. However, the resulting lower bound is weaker than ours and requires stronger assumptions:

(1) Assuming that the subgaussian norm $\|\boldsymbol{w}_i\|_{\psi_2} := \sup_{\boldsymbol{v}\in\mathbb{S}^{p-1}} \sup_{q\in\mathbb{N}_+} q^{-1/2}(\mathbb{E}|\boldsymbol{v}^{\top}\boldsymbol{w}_i|^q)^{1/q}$ (cf. Vershynin, 2010) is bounded by a constant $K < \infty$, they obtain a lower bound of the form $c_K\sigma^2\frac{n}{p}$ with a constant $c_K > 0$ that depends on $K$ and is only explicitly specified for the case of centered Gaussian $P_Z$. They note that $\|\boldsymbol{w}_i\|_{\psi_2} \leq K$ holds, for example, if the components $w_{i,j}$ of $\boldsymbol{w}_i$ are independent and all satisfy $\|w_{i,j}\|_{\psi_2} \leq K$. However, as discussed in Remark 1, such independence assumptions are not realistic. In contrast, our lower bound is explicit, independent of constants like $K$ and is larger: For example, at $n = p$, our lower bound is $\sigma^2 n$ and theirs is $\sigma^2 c_K$.

(2) Assuming $\|\boldsymbol{w}_i\|_2^2 \leq p$ almost surely, they obtain a lower bound of the form $c\sigma^2\frac{n}{p\log(n)}$. First of all, this lower bound converges to zero as $n = p \to \infty$. Moreover, since we always have $\mathbb{E}\|\boldsymbol{w}_i\|_2^2 = \mathbb{E}\operatorname{tr}(\boldsymbol{w}_i\boldsymbol{w}_i^{\top}) = \mathbb{E}\operatorname{tr}(\boldsymbol{I}_p) = p$, the assumption implies $\|\boldsymbol{w}_i\|_2^2 = p$ almost surely. Although we can sometimes guarantee constant $\|\boldsymbol{z}_i\|_2^2$, e.g. for certain random Fourier features, we cannot guarantee the same for $\boldsymbol{w}_i = \boldsymbol{\Sigma}^{-1/2}\boldsymbol{z}_i$ since $\boldsymbol{\Sigma}$ depends on the unknown input distribution $P_X$.

By inspecting the proof behind (1), one finds that $c_K \to 0$ as $K \to \infty$. Hence, lower bound of Muthukumar et al. (2020) might raise hope that it is possible to achieve low $\mathcal{E}_{\text{Noise}}$ by choosing features with a large (or even infinite) subgaussian norm. Our result shows that this is not possible: Essentially the only possibility to avoid a large $\mathcal{E}_{\text{Noise}}$ for ridgeless linear regression around $n \approx p$ is to violate the property (FRK) that guarantees the ability to interpolate the data in the overparameterized case, see Section 5. Otherwise, in order to achieve $\mathcal{E}_{\text{Noise}} < \varepsilon\sigma^2$, $\varepsilon \ll 1$, it is necessary to make the model either strongly underparameterized ($p < \varepsilon n$) or strongly overparameterized ($p > n/\varepsilon$).

