# OpenReview forum: "On the Universality of the Double Descent Peak in Ridgeless Regression"
_ICLR.cc/2021/Conference — ICLR 2021 Poster_

### Official Review · AnonReviewer2 · 2020-10-26
**While this paper provides relevant theoretical insight into ridgeless linear regression there are some concerns over novelty given prior work.**

**Rating:** 6
**Confidence:** 3

**Review:**

# Contributions

This work studies linear regression with feature maps (or kernel regression) without regularization in order to theoretically explore double descent phenomena seen empirically when training over-parametrized networks.

This paper provides lower bounds on the out of sample error or generalization error caused by the label noise. In the over-parametrized regime or beyond the interpolation threshold this is the primary source of error.

While this setting has been studied before this work strictly generalizes previously provided bounds especially around the interpolation threshold. They also consider analytic feature maps including random ones and thus imply results for random deep neural networks.

The main text of the paper is well organized. However, it could benefit from some more clarity in presentation of the technicalities especially in section  5 when comparing with prior work and in section 6 when providing examples.

# Concerns
1. It seems that a big part of the assumptions is that the data is generated from a full dimensional distribution. Given that in the high dimensional settings where d ~ n a key problem is to characterize the behavior of estimators when true data has low intrinsic dimension can we say anything about this regime given the results in the paper?
2. It seems that the main novelty over prior work especially Muthukumar et al. (2020) is around the interpolation threshold or slight weakening of the assumption on the noise. Since the over-parametrized regime is currently most relevant to the community is there a relevant example in this regime where the results provided in this work are strictly better (in terms of actual rates or in understand of this regime) than those from prior work?
3. Or just as importantly are there examples of proof techniques that are used here that are substantially novel over those in the prior work that may be beneficial to the community in general in understanding these over-parametrized regimes?
4. There is some recent work on ridgless kernel regression by [Liang et al. (2020)][1]. Since these two settings are fairly intertwined it would be nice to understand how results in current work compare to the results in this paper.

[1]:https://arxiv.org/abs/1808.00387

---

> ### Author Response · Authors · 2020-11-16
> **Answer to AnonReviewer2**
>
> Thank you for the positive feedback. In the meantime, we have further improved our paper, see the top-level comment. If you have specific suggestions for Section 5 and 6, we are happy to include them.
>
> Regarding your concerns:
> 1. We address this with the updated theorems for deep NNs and random Fourier features in the new version: In these cases, the input distribution now only needs to be nonatomic (i.e. every individual point has probability 0), which allows for distributions on submanifolds. We discuss this in more detail in the new revision of our paper in the paragraph before Theorem 10 and the corresponding footnote.
> 2. The main novelty over Muthukumar et al. (2020) is *not* the noise assumption although this is also an improvement. Rather, the main novelty is that our lower bound does not involve a distribution-dependent constant and that its peak height converges to infinity for $p = n \to \infty$. We discuss the differences to Muthukumar et al. (2020) at the end of Section 4 (in both the new and old version of the paper). Here is a short summary: The bound of Muthukumar et al. (2020) involves a non-explicit constant $c_K$ depending on the subgaussian norm $K$ of the whitened features. For example, for $p = n$, our lower bound is $\sigma^2 n$ and their bound is $\sigma^2 c_K$, which can become arbitrarily small for large $K$. $K$ can change depending on the feature map, hence it can also change depending on $p$, which is not desirable for a double descent analysis. We can also compare both bounds to the uniform distribution on the sphere (Theorem 11) for $p-2 \geq n \geq 2$, where $\mathcal{E}_{\mathrm{noise}} = \sigma^2 \frac{n}{p-1-n} \cdot \frac{p-2}{p}$, our lower bound is $\sigma^2 \frac{n}{p+1-n} = \sigma^2 \frac{n}{p-1-n} \cdot \frac{(p+1-n)-2}{p+1-n}$ and their lower bound is $\sigma^2 c_K \frac{n}{p}$.
> 3. The proof of our overparameterized lower bound essentially relies on a novel combination of Jensen's inequality with a Schur complement reformulation. The latter reformulation also enabled us to prove the explicit result for the sphere in Theorem 11. Moreover, perhaps our proof of the "whitening inequality" $(**)$ ($(\mathrm{II})$ in the new version) in Theorem 3 can be helpful to understand the influence of the eigenvalues of $\boldsymbol{\Sigma}$ on the noise-induced error. It should also be noted that while the lower bound is the main message of our paper, the majority of our proof efforts (ca. 15 pages) go into proving the novel results in Section 5 and 6, especially the result for random deep neural networks. These proofs also involve novel approaches that might be useful to the community.
> 4. We discuss the relation to Liang et al. (2020) and other related papers on ridgeless kernel regression in detail in Appendix K in the new version of our paper. Liang et al. (2020) derive upper bounds for a certain class of kernels and input distributions with (linearly transformed) i.i.d. components. Their analysis focuses on the high-dimensional limit $d, n \to \infty$ with $c \leq d/n \leq C$ and ignores the dimension $p$ of the feature map / RKHS. We argue in Appendix K that their analysis is not impacted by Double Descent w.r.t. $p$ since their kernels satisfy either $p=\infty$ or $p/n \to \infty$ as $d, n \to \infty$.

---

### Official Review · AnonReviewer4 · 2020-10-28
**Official Blind Review #4**

**Rating:** 6
**Confidence:** 2

**Review:**

Summary:

The paper focuses on the theoretical understanding of the so-called double descent phenomenon, which may offer insights into the practical success of deep learning methods and has been observed in both overparametrized neural networks and kernel machines.  In particular, the authors derive a nonasymptotic distribution-independent lower bound on the excess generalization error of the ridgeless linear regression under mild conditions on the input distributions and feature maps. More specifically, their analysis applies to the cases where the input distribution has a Lebesgue density and the features are induced by random deep neural networks with analytic activation functions,  random Fourier features, polynomial kernels, and so on. The sharpness of the lower bound has been demonstrated by some numerical experiments. The results should be of interest to the community of theoretical deep learning. Overall, I vote for accepting.


Concerns:

1. Is it possible to derive a nonasymptotic upper bound which matches the lower bound in Theorem 3?
2. In Theorem 3 and Corollary 4, which lower bound should be adopted when $p=n$, $\sigma^2n$ or $\sigma^2$?
3. In the underparameterized regime ($\gamma<1$), the derived lower bound seems to be not asymptotically sharp by looking at Theorem 2 of Hastie et al. (2019), any special reason for this?
4. The authors may consider using subsections since sections 2 and 3 are very short compared to other sections.

---

> ### Author Response · Authors · 2020-11-16
> **Answer to AnonReviewer4**
>
> Thank you for the positive feedback. In the meantime, we have further improved our paper, see the top-level comment.
>
> Regarding your concerns:
> 1. The answer depends on whether the upper bound should hold for all distributions, a larger class of distributions or a single "optimal" distribution:
>     - It seems that there exist distributions with arbitrarily high $\mathcal{E}_{\mathrm{noise}}$. Hence, there probably does not exist a uniform upper bound that holds for all distributions.
>     - It is a challenging open problem to find upper bounds for larger classes of distributions. However, such an upper bound will most likely not match our lower bound.
>     - The best values known to us for any single distribution are those for the sphere (Theorem 11). These match the lower bound in the case $p=1$ or $n=1$. We conjecture that the values for the sphere are the best possible values, which means the lower bound could still be improved in the non-asymptotic regime. In the underparameterized case, we cannot compute the values for the sphere, but the values for the Gaussian distribution are not much larger (Theorem 12).
> 2. It is possible to adopt the better lower bound $\sigma^2 n$. In the new version of the paper, both bounds yield $\sigma^2 n$.
> 3. The reason was that when adapting the proof of the overparameterized case to the underparameterized case, it is not possible to exploit the independence of the samples anymore (see p. 22 in the old version). However, as mentioned above, we have a better underparameterized lower bound in the new version, and this improved lower bound is asymptotically sharp.
> 4. Thanks for your suggestion. Using subsections is indeed an option, but we feel that introducing them only for Section 2 and 3 may not be worthwile.

---

### Official Review · AnonReviewer3 · 2020-10-29
**"On the Universality of the Double Descent Peak in Ridgeless Regression" review**

**Rating:** 7
**Confidence:** 3

**Review:**

This work studies the double descent phenomenon in ridgeless regression with deterministic or random features. The work provides a lower bound on the generalization error that requires weaker assumptions than bounds given in previous work and applies to many interesting learning methods.

Strengths:
-- The generalization bound presented this work requires fewer assumptions than previous such bounds while also being stronger.
-- The work is theoretically rigorous and helps to shed light on why various learning methods perform well.
-- The authors thoroughly investigate the applicability of their bound with specific discussion of each of their assumptions.

Weaknesses:
-- While the analysis applies to a class of feedforward neural networks with analytic activation functions, some common activationswith a perfectly linear component like ReLU, however are not covered by the results in this work.

The second sentence in Remark 7 says, "If (d) in Propositions 8 does not hold..." but I don't think you mean to reference Prop. 8 here.

---

> ### Author Response · Authors · 2020-11-16
> **Answer to AnonReviewer3**
>
> Thank you for the positive feedback. In the meantime, we have further improved our paper, see the top-level comment. Thanks also for your comment on Remark 7, the reference is now corrected to Proposition 6.

---

### Official Review · AnonReviewer1 · 2020-11-02
**Well-written Paper on Generality of Double Descent Phenomenon for Unregularized Regression**

**Rating:** 7
**Confidence:** 3

**Review:**

The paper studies the phenomenon of double descent for ridgeless regression. They show that when the label noise in the regression problem is lower bounded, the test error for regression must peak at the interpolation threshold (n=p) before descending again in the over-parameterized regime and that this holds with very weak assumptions making it a universal phenomenon when we consider unregularized linear regression.
These results extend our understanding of double descent and point that under most general settings it is impossible to avoid for ridgeless linear regression.

The paper is well-written and the analysis and comparison to prior work provided appears thorough. I have not verified all the proofs in the appendix.

-------
Thank you to the authors for their response and update. I have read the response and am keeping my current rating for the paper.

---

> ### Author Response · Authors · 2020-11-16
> **Answer to AnonReviewer1**
>
> Thank you for the positive feedback. In the meantime, we have further improved our paper, see the top-level comment.

---

### Author Response · Authors · 2020-11-16
**New revision with updated lower bound and deep NN theorem**

Dear reviewers,

besides other smaller updates, we have made some major improvements to our paper:
1. We have been informed that an adaptation of an argument by [Mourtada (2019)](https://arxiv.org/abs/1912.10754) yields a better lower bound $\sigma^2 \frac{p}{n+1-p}$ instead of $\sigma^2 \frac{p}{n}$ in the underparameterized case. Despite requiring a different proof strategy than our novel overparameterized lower bound, both bounds are now identical with opposite roles of $n$ and $p$. We have incorporated the better underparameterized lower bound into our theorems and plots. **With this update, the underparameterized lower bound is now also asymptotically sharp (cf. Section 6).**
2. We have been informed that a property similar to (FRK) has been proven for certain deep neural networks by Lemma 4.4 in [Nguyen and Hein (2017)](https://arxiv.org/abs/1704.08045) and, in contrast to the results mentioned in our Appendix E, this proof appears to be correct. In our new revision, we mention this result and we have improved our Theorem for deep NNs (and also the result for random Fourier features). **Our new version strictly improves on both our old version and Nguyen and Hein (2017):**
    - In contrast to our old result, the new result does not make any restrictions on the hidden layer sizes and it only requires the input distribution $P_X$ to be nonatomic (instead of requiring a Lebesgue density). Especially, this allows for distributions $P_X$ that are concentrated on a submanifold, as long as no single input value $\boldsymbol{x}$ occurs with positive probability.
    - In contrast to Nguyen and Hein (2017), our result also applies to networks without biases and it applies to more activation functions. Activation functions supported by our Theorem but not by Nguyen and Hein (2017) include GELU, SiLU/Swish, Mish, RBF, sin and cos.
    While both our result and the result by Nguyen and Hein (2017) are based on the identity theorem for analytic functions, we need to use fundamentally different arguments to cover a larger class of activation functions. We are happy to elaborate on the differences if you are interested.
3. We have included experiments on random Fourier features in Appendix C.

---

### Decision · Program_Chairs · 2021-01-07
**Final Decision**

**Decision:**

Accept (Poster)

**Comment:**

This paper shows that the double descent phenomenon of ridgeless regression appears under considerably general settings of the input distributions by showing a lower bound of the excess risk. The analysis covers various types of input distributions including deterministic and random feature maps and its asymptotic sharpness is also shown.

One reviewer raised a concern about its novelty compared with existing work, but the authors properly clarified the novelty in the rebuttal and updated version of the manuscript. Although there were some other minor concerns, the reviewers all agree that this paper gives a valuable theoretical result supporting universality of double descent phenomenon. I also concur with this assessment. I think this paper is a solid theoretical paper giving an informative result as a piece of researches in double descent. Thus, I would recommend acceptance of this paper.